# Emotionotopy in the human right temporo-parietal cortex

Giada Lettieri [1,2], Giacomo Handjaras [1,2], Emiliano Ricciardi[1], Andrea Leo [1], Paolo Papale [1], Monica Betta[1], Pietro Pietrini[1,3] & Luca Cecchetti [1,3]*

Humans use emotions to decipher complex cascades of internal events. However, which mechanisms link descriptions of affective states to brain activity is unclear, with evidence supporting either local or distributed processing. A biologically favorable alternative is provided by the notion of gradient, which postulates the isomorphism between functional representations of stimulus features and cortical distance. Here, we use fMRI activity evoked by an emotionally charged movie and continuous ratings of the perceived emotion intensity to reveal the topographic organization of affective states. Results show that three orthogonal and spatially overlapping gradients encode the polarity, complexity and intensity of emotional experiences in right temporo-parietal territories. The spatial arrangement of these gradients allows the brain to map a variety of affective states within a single patch of cortex. As this organization resembles how sensory regions represent psychophysical properties (e.g., retinotopy), we propose emotionotopy as a principle of emotion coding.

---

[1] MoMiLab Research Unit, IMT School for Advanced Studies Lucca, Lucca, Italy. [2]These authors contributed equally: Giada Lettieri, Giacomo Handjaras. [3]These authors jointly supervised this work: Pietro Pietrini, Luca Cecchetti. *email: luca.cecchetti@imtlucca.it

To understand our own emotions, as well as those of others, is crucial for human social interactions. Also, witnessing facts and events of others' life sometimes prompts inner reactions related to the beliefs, intentions and desires of actors. Through years, the relevance and pervasiveness of these aspects motivated the quest for models that optimally associate behavioral responses to emotional experiences.

In this regard, seminal works pointed toward the existence of discrete basic emotions, characterized by distinctive and culturally stable facial expressions[1], patterns of autonomous nervous system activity[2,3] and bodily sensations[4]. Happiness, surprise, fear, sadness, anger and disgust represent the most frequently identified set of basic emotions[5], though alternative models propose that other emotions, such as pride or contempt, should be included for their social and biological relevance[6]. To prove the neurobiological validity of these models, neuroscientists investigated whether basic emotions are consistently associated with specific patterns of brain responses across subjects. Findings show that activity in amygdala, medial prefrontal, anterior cingulate, insular, middle/inferior frontal, and posterior superior temporal cortex, is associated to the perceived intensity of emotions and supports their recognition[7-10]. However, this perspective has been challenged by other studies, which failed to demonstrate significant associations between single emotions and activity within distinct cortical areas or networks[11-13].

An alternative theory proposes that behavioral and physiological characteristics of emotions would be more appropriately described along a number of continuous cardinal dimensions[14,15], generally one governing pleasure versus displeasure (i.e., valence) and another one the strength of the experience (i.e., arousal). While these two dimensions have been reliably and consistently described, other models propose that additional dimensions, such as dominance or unpredictability, are needed to adequately explain affective states[16,17]. Neuroimaging studies also demonstrated that stimuli varying in valence and arousal elicit specific and reliable brain responses[18,19], which have been recently employed to decode emotional experiences[20]. Activity recorded in insula, amygdala, ventral striatum, anterior cingulate, ventromedial prefrontal and posterior territories of the superior temporal cortex is associated to transitions between positive and negative valence and fluctuations in arousal[21,22].

Of note, other than in ventromedial prefrontal regions, studies using either discrete emotion categories[9,10,12] or emotion dimensions[22-25] have shown responses in the posterior portion of the superior temporal cortex, extending to temporo-parietal territories. Furthermore, these temporo-parietal regions are fundamental for social cognition, as they support empathic processing[26,27] and the attribution of intentions, beliefs and emotions to others[28,29].

However, despite this large body of evidence, it remains to be determined whether emotional experiences are better described through discrete basic emotions or emotion dimensions. Moreover, regardless of the adopted model, it is still debated how emotion features are spatially encoded in the brain[8,11,13,30-32]. As a matter of fact, while findings support the role of distinct regions[7], others indicate the recruitment of distributed networks in relation to specific affective states[33].

An alternative and biologically favorable perspective may be provided by the notion of gradient. Gradients have been proven a fundamental organizing principle through which the brain efficiently represents and integrates stimuli coming from the external world. For instance, the location of a stimulus in the visual field is easily described through two orthogonal spatially overlapping gradients in primary visual cortex: rostrocaudal for eccentricity and dorsoventral for polar angle[34]. Thus, using functional magnetic resonance imaging (fMRI) and retinotopic mapping, one can easily predict the location of a stimulus in the visual field considering the spatial arrangement of recruited voxels with respect to these orthogonal gradients. Crucially, recent investigations revealed that gradients support the representation of higher-order information as well[35-37], with features as animacy or numerosity being topographically arranged onto the cortical mantle[35,38,39].

Following this view, we hypothesize that affective states are encoded in a gradient-like manner in the human brain. Specifically, different affective states would be mapped onto the cortical mantle through spatially overlapping gradients, which would code either the intensity of discrete emotions (e.g., weak to strong sadness) or, alternatively, the smooth transitions along cardinal dimensions (e.g., negative to positive valence). In either case, the pattern of brain activity could be used to predict the current affective state as function of cortical topography.

Here, we test this hypothesis using moment-by-moment ratings of the perceived intensity of emotions elicited by an emotionally charged movie. To unveil cortical regions involved in emotion processing, behavioral ratings are used as predictors of fMRI activity in an independent sample of subjects exposed to the same movie. The correspondence between functional characteristics and the relative spatial arrangement of distinct patches of cortex is then used to test the topography of affective states. Results show that three orthogonal and spatially overlapping gradients encode the polarity, complexity and intensity of emotional experiences in right temporo-parietal cortex. As this organization resembles how primary sensory regions represent psychophysical properties of stimuli (e.g., retinotopy), we propose emotionotopy as a principle of emotion coding in the human brain.

## Results

**Emotion ratings.** A group of Italian native speakers continuously rated the perceived intensity of six basic emotions[5] (i.e., happiness, surprise, fear, sadness, anger and disgust) while watching an edited version of the Forrest Gump movie (R. Zemeckis, Paramount Pictures, 1994). We first assessed how much each basic emotion contributed to the behavioral ratings and found that happiness and sadness explained 28% and 36% of the total variance, respectively. Altogether, fear (18%), surprise (8%), anger (7%) and disgust (3%) explained the remaining one-third of the total variance. We also evaluated the agreement in ratings of the six basic emotions (Fig. 1a), and found that happiness (mean Spearman's $\rho = 0.476 \pm 0.102$ standard deviation, range 0.202–0.717), fear ($\rho = 0.522 \pm 0.134$, range 0.243–0.793), sadness ($\rho = 0.509 \pm 0.084$, range 0.253–0.670), and anger ($\rho = 0.390 \pm 0.072$, range 0.199–0.627) were consistent across all subjects, whereas surprise ($\rho = 0.236 \pm 0.099$, range 0.010–0.436) and disgust ($\rho = 0.269 \pm 0.115$, range 0.010–0.549) were not. Nonetheless, ratings for these latter emotions were on average significantly different from a null distribution of randomly assigned emotion ratings ($p$-value < 0.05; permutation test).

To reveal emotion dimensions, we averaged across subjects the ratings of the six basic emotions, measured their collinearity (Fig. 1b) and performed principal component (PC) analysis (Fig. 1c). The first component reflected a measure of polarity (PC$_1$: 45% explained variance) as positive and negative emotions demonstrated opposite loadings. The second component was interpreted as a measure of complexity (PC$_2$: 24% explained variance) of the perceived affective state, ranging from a positive pole where happiness and sadness together denoted inner conflict and ambivalence, to a negative pole mainly representing fearful events. The third component was a measure of intensity (PC$_3$: 16% explained variance), since all the six basic emotions showed

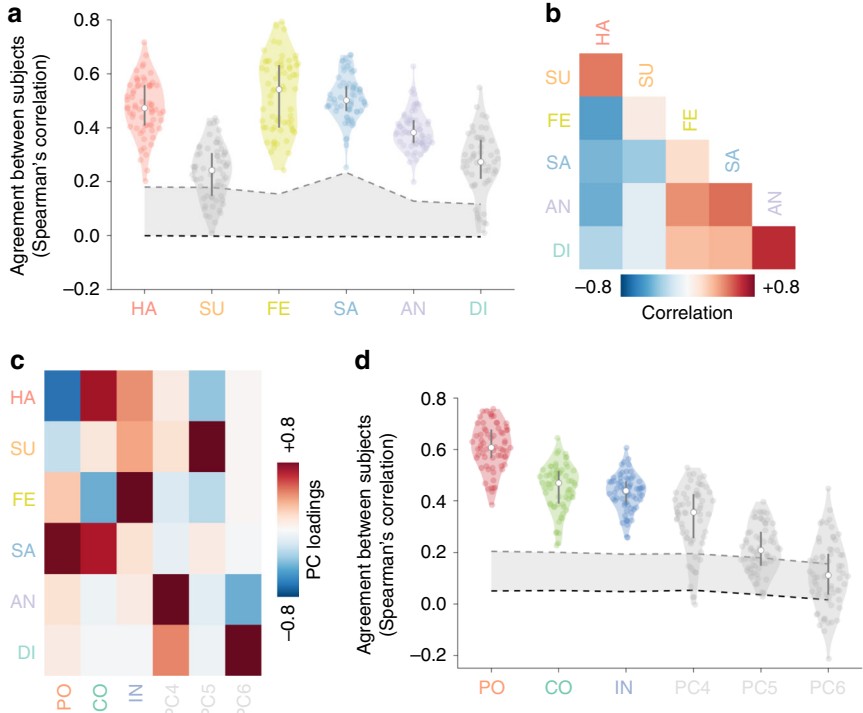

**Fig. 1 Emotion ratings. a** Violin plots show the agreement between subjects (Spearman's $\rho$ coefficient) of the six basic emotions. White circular markers indicate mean correlation across subjects and black bars denote 25th and 75th percentile of the distribution ($n = 66$ subject pairings). Gray shaded area represents the null distribution of behavioral ratings and dashed lines the mean and 95th percentile of the null distribution. **b** Correlation matrix showing Spearman's $\rho$ values for pairings of basic emotions. **c** Principal component analysis: loadings of the six principal components. Explained variance was 45% for polarity, 24% for complexity and 16% for intensity. **d** Violin plots show the agreement between subjects (Spearman's $\rho$ coefficient) of the six principal components. White circular markers indicate mean correlation across subjects and black bars denote 25th and 75th percentile of the distribution ($n = 66$ subject pairings). Gray shaded area represents the null distribution of behavioral ratings and dashed lines the mean and 95th percentile of the null distribution. HA happiness, SU surprise, FE fear, SA sadness, AN anger, DI disgust, PC principal component, PO polarity, CO complexity, IN intensity.

positive loadings (Fig. 1c). Altogether, the first three components explained ~85% of the total variance. We further assessed the stability of the PCs and found that only these first three components (polarity: $\rho = 0.610 \pm 0.089$, range 0.384–0.757; complexity: $\rho = 0.453 \pm 0.089$, range 0.227–0.645; intensity: $\rho = 0.431 \pm 0.071$, range 0.258–0.606), hereinafter emotion dimensions, were consistent across all subjects (Fig. 1d). The fourth PC described movie segments during which participants experienced anger and disgust at the same time (PC$_4$: 8% explained variance, $\rho = 0.329 \pm 0.128$, range $-0.003$–0.529), whereas the fifth PC was mainly related to surprise (PC$_5$: 6% explained variance, $\rho = 0.214 \pm 0.090$, range 0.028–0.397). Notably, these two PCs were not consistent across all subjects, even though their scores were on average significantly different from a null distribution ($p$-value < 0.05; permutation test). Scores of the sixth PC were not significantly consistent across subjects (PC$_6$: 1% explained variance, $p$-value > 0.05; permutation test).

**Richness of the reported emotional experience.** In our behavioral experiment, participants were allowed to report the perceived intensity of more than one emotion at a time. Thus, the final number of elicited emotional states might be greater than the original six emotion categories. To measure the richness of affective states reported by our participants, we performed dimensionality reduction and clustering analyses on group-averaged behavioral ratings. Results revealed the existence of 15 distinct affective states throughout the movie (Fig. 2), indicating that Forrest Gump evoked complex and multifaceted experiences, which cannot be reduced to the original six emotion categories.

**Brain regions encoding emotion ratings.** Emotion ratings obtained from the behavioral experiment explained brain activity in independent subjects exposed to the same movie (studyforrest project http://studyforrest.org[40]; $q < 0.01$ false discovery rate - FDR - corrected and cluster size >10; voxelwise encoding permutation test; Fig. 3a and Supplementary Table 1). Notably, the association between emotion ratings and brain activity was right-lateralized and the peak was found in the right posterior superior temporal sulcus/temporo-parietal junction (pSTS/TPJ), an important region for social cognition[12,22,26,28,29] ($R^2 = 0.07 \pm$ standard error = 0.009; center of gravity—CoG: $x = 61$, $y = -40$, $z = 19$; noise ceiling lower bound 0.13, upper bound 0.23; Fig. 3b and Supplementary Fig. 1). The peak of association was also located in proximity (11 mm displacement) of the reverse inference peak for the term TPJ (CoG: $x = 58$, $y = -50$, $z = 16$), as reported in the NeuroSynth database (http://neurosynth.org) (Fig. 3b).

**Emotion gradients in right temporo-parietal cortex.** We tested the existence of either basic emotion or emotion dimension gradients in a spherical region of interest (ROI) located at the reverse inference peak for the term TPJ. This analysis was conducted on behavioral ratings consistent across all subjects: happiness, sadness, fear and anger for basic emotions and polarity, complexity and intensity for emotion dimensions.

Using $\beta$ coefficients obtained from the encoding analysis, we observed that, within right TPJ, voxels appeared to encode happiness in an anterior to posterior arrangement, fear and sadness in an inferior to superior manner, while anger showed a patchier organization (Fig. 3c). With respect to emotion

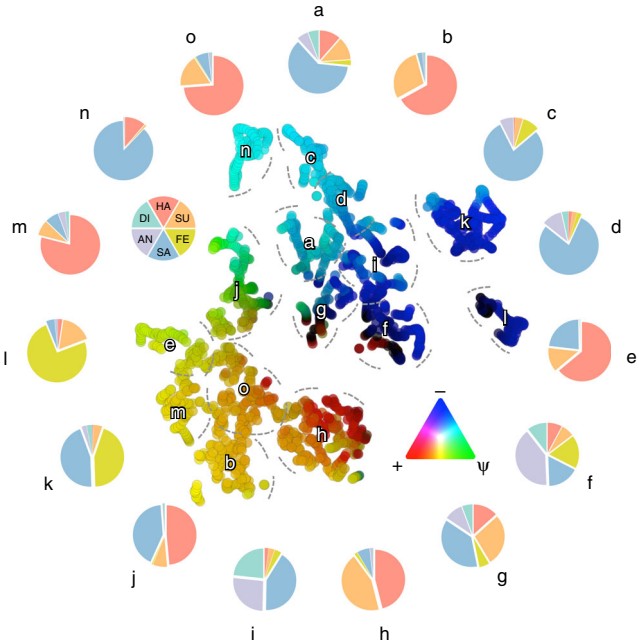

**Fig. 2 Richness of the emotional experience.** Results of the dimensionality reduction (t-SNE) and clustering analyses (k-means) on the group-averaged behavioral ratings showing the existence of 15 distinct affective states throughout the movie. Each element represents a specific timepoint in the movie and the distance between elements depends on the statistical similarity of emotion ratings. Element color reflects the scores of the polarity and complexity dimensions: positive (+) and negative (−) events (i.e., polarity) are associated, respectively, to the red and blue channels, whereas complexity (Ψ) scores modulate the green channel. Pie charts show the relative contribution of the six basic emotions to each of the 15 identified clusters. Combinations of distinct emotions likely express secondary affective states, as ambivalence (i.e., cluster j depicting movie scenes in which happiness and sadness are simultaneously experienced) or resentment (i.e., cluster i representing movie segments in which a mixture of sadness, anger and disgust is perceived). Of note, this evidence is also supported by single-subject reports, in which the 38% (SE: ±2.3%) of timepoints were associated to a single emotion, the 29% (SE: ±3.5%) to two basic emotions and the 6% (SE: ±1.4%) to the concurrent experience of three distinct emotions. HA happiness, SU surprise, FE fear, SA sadness, AN anger, DI disgust.

dimensions, voxels seemed to encode polarity and intensity in a more inferior to superior fashion, whereas complexity in a more posterior to anterior direction (Fig. 3d).

To prove the existence and precisely characterize the orientation of these gradients, we tested the association between physical distance and functional characteristics of right TPJ voxels (Supplementary Fig. 2). Results demonstrated that within a 15-mm radius sphere, the relative spatial arrangement and functional features of right TPJ were significantly and maximally correlated, either considering the basic emotion model ($\rho = 0.352$, $p$-value = 0.004; permutation test; 95% confidence interval—CI: 0.346–0.357) or the emotion dimension one ($\rho = 0.399$, $p$-value < 0.001; permutation test; 95% CI: 0.393–0.404). For alternative definitions of the right TPJ region, see Supplementary Table 2.

Crucially, when focusing on each emotion dimension, results revealed the existence of three orthogonal and spatially over-lapping gradients: polarity ($\rho = 0.241$, $p$-value = 0.041; permutation test; 95% CI: 0.235–0.247), complexity ($\rho = 0.271$, $p$-value = 0.013; permutation test; 95% CI: 0.265–0.277) and intensity ($\rho = 0.229$, $p$-value = 0.049; permutation test; 95% CI: 0.223–0.235; Fig. 4 and Supplementary Table 3). On the contrary, happiness

($\rho = 0.275$, $p$-value = 0.013; permutation test; 95% CI: 0.269–0.281), but not other basic emotions, retained a gradient-like organization (fear: $\rho = 0.197$, $p$-value = 0.091; sadness: $\rho = 0.182$, $p$-value = 0.160; anger: $\rho = 0.141$, $p$-value = 0.379; permutation test; Supplementary Table 3). Of note, the peculiar arrangement of group-level emotion dimension gradients (Fig. 4) was also identified using single-subject fMRI data (Supplementary Fig. 3 and Supplementary Table 4).

As any orthogonal rotation applied to the emotion dimensions would result into different gradients, we measured to what extent rotated solutions explained the topography of right TPJ. Therefore, we tested the correspondence between anatomical distance and the fitting of ~70,000 rotated versions of polarity, complexity and intensity (see Supplementary Methods for a comprehensive description). Results showed that the original unrotated emotion dimensions represented the optimal solution to explain the gradient-like organization of right temporo-parietal cortex (Supplementary Fig. 4).

Further, we performed a data-driven searchlight analysis to test whether right TPJ was the only region significantly encoding all the three emotion dimension gradients (please refer to Supplementary Methods for details). Results obtained from the meta-analytic definition of right TPJ were confirmed using this alternative approach ($q < 0.05$ FDR corrected and cluster size >10; voxelwise encoding permutation test; CoG: $x = 58$, $y = -53$, $z = 21$; Supplementary Fig. 5), as no other region encoded the combination of polarity, complexity and intensity in a topographic manner.

Moreover, we conducted three separate searchlight analyses to characterize the spatial arrangement of single emotion dimension gradients (see Supplementary Information). Polarity, complexity and intensity maps revealed specific topographies: regions as the right precentral sulcus represented the three emotion dimensions in distinct—yet adjoining—subregions, whereas the right occipito-temporal sulcus encoded overlapping gradients of complexity and intensity (Supplementary Fig. 6).

When we explored whether the left hemisphere homologous of TPJ (CoG: $x = -59$, $y = -56$, $z = 19$) showed a similar gradient-like organization, we did not find significant associations between spatial and functional characteristics either for the basic emotion model ($\rho = 0.208$, $p$-value = 0.356; permutation test) or the emotion dimension one ($\rho = 0.251$, $p$-value = 0.144; permutation test; Supplementary Table 2). Specifically, neither any of the emotion dimensions (polarity: $\rho = 0.132$, $p$-value = 0.354; complexity: $\rho = 0.157$, $p$-value = 0.222; intensity: $\rho = 0.149$, $p$-value = 0.257; permutation test) nor any of the basic emotions showed a gradient-like organization in left TPJ (happiness: $\rho = 0.158$, $p$-value = 0.216; fear: $\rho = 0.142$, $p$-value = 0.293; sadness: $\rho = 0.156$, $p$-value = 0.213; anger: $\rho = 0.073$, $p$-value = 0.733; permutation test; Supplementary Table 3).

Lastly, as spatial smoothness of functional data and cortical folding may affect the estimation of gradients, we performed additional analyses considering the unfiltered version of group-average brain activity and obtaining a measure of the anatomical distance respectful of cortical topology. Results showed that the topographic arrangement of emotion dimensions in right temporo-parietal territories was not affected by smoothing (Supplementary Fig. 7) and respected the cortical folding (polarity: $\rho = 0.248$, $p$-value = 0.026, CI: 0.238–0.257; complexity: $\rho = 0.314$, $p$-value = 0.001, CI: 0.304–0.323; intensity: $\rho = 0.249$, $p$-value = 0.013, CI: 0.239–0.258; permutation test). For details about this procedure and a comprehensive description of the results, please refer to Supplementary Information.

To summarize, polarity, complexity and intensity dimensions were highly consistent across individuals, explained the majority of the variance in behavioral ratings (85%) and were mapped in a

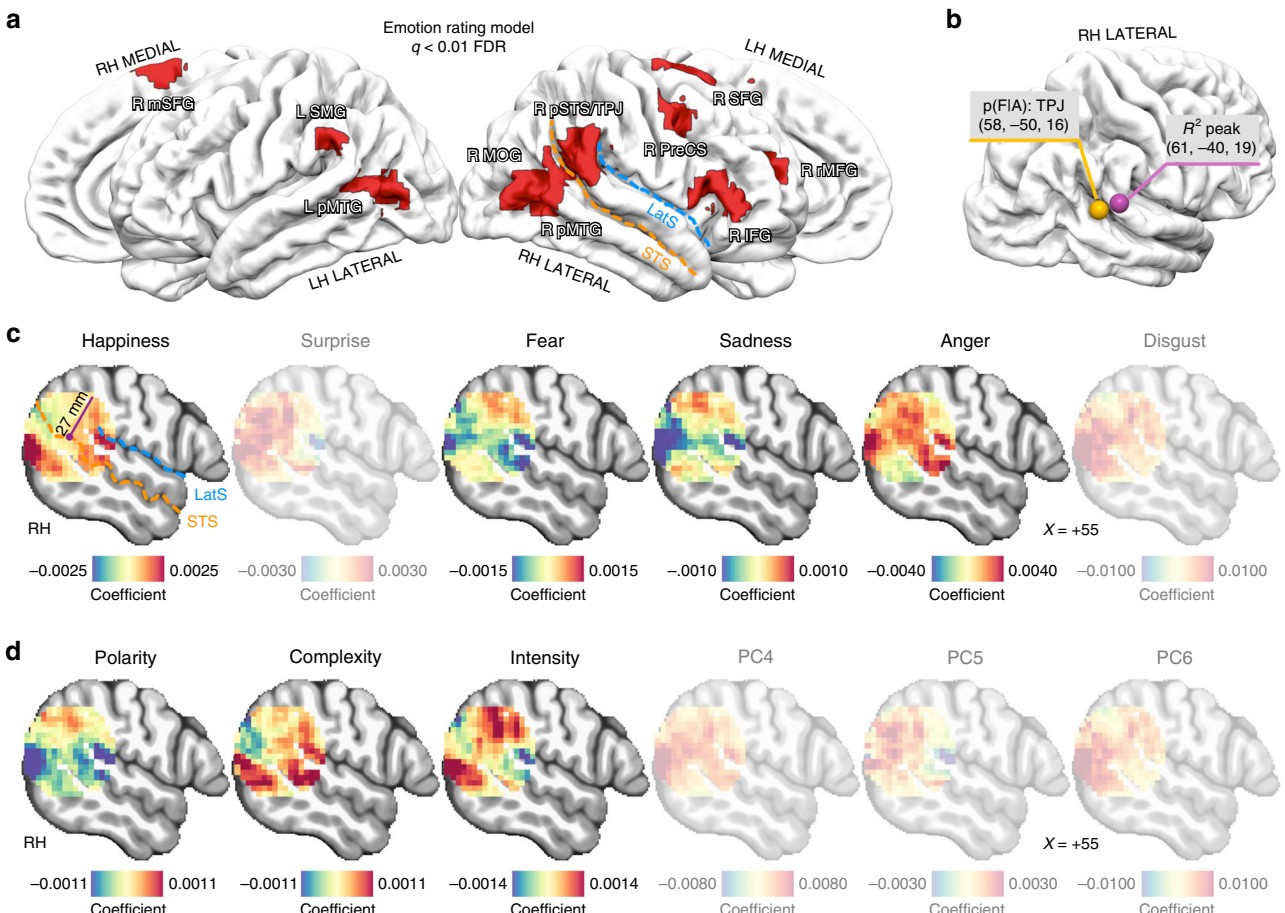

**Fig. 3 Encoding of emotion ratings. a** Brain regions encoding emotion ratings corrected for multiple comparisons using the False Discovery Rate method ($q < 0.01$; voxelwise encoding permutation test; $n = 3595$ timepoints). **b** Peak of association between emotion ratings and brain activity (purple sphere) and reverse inference peak for the term TPJ as reported in the NeuroSynth database (yellow sphere). Coordinates represent the center of gravity in MNI152 space. **c** $\beta$ coefficients associated to basic emotions in a spherical region of interest (27 mm radius) located at the reverse inference peak for the term TPJ. Maps for emotions not consistent across all the subjects (i.e., surprise and disgust) are faded. **d** $\beta$ coefficients associated to emotion dimensions in a spherical region of interest (27 mm radius) located at the reverse inference peak for the term TPJ. Maps for components not consistent across all the subjects (i.e., PC4, PC5 and PC6) are faded. IFG inferior frontal gyrus, rMFG rostral middle frontal gyrus, mSFG medial superior frontal gyrus, preCS precentral sulcus, pSTS/TPJ posterior part of the superior temporal sulcus/temporo-parietal junction, MOG middle occipital gyrus, pMTG posterior middle temporal gyrus, SMG supramarginal gyrus, LatS lateral sulcus, STS superior temporal sulcus.

gradient-like manner in right (but not left) TPJ. Happiness (28% of the total variance in behavioral ratings) was the only basic emotion to be consistent across subjects and to be represented in right TPJ. Importantly, though, happiness and complexity demonstrated high similarity both in behavioral ratings ($\rho = 0.552$) and in brain activity patterns ($\rho = 0.878$). Taken together, these pieces of evidence indicate the existence of emotion dimension gradients in right temporo-parietal cortex, rather than the presence of discrete emotion topographies.

**Emotion dimension gradients and portrayed emotions.** In movie watching, actions and dialogues are not usually directed toward the observer and the reported subjective experience is very likely influenced by character emotions, intentions and beliefs. Therefore, we tested whether the gradient-like organization of right TPJ can be explained considering portrayed emotions of movie characters. We took advantage of publicly available tagging data of Forrest Gump[41], in which participants indicated the portrayed emotion of each character and whether it was directed toward the character itself (self-directed; e.g., Forrest feeling sad) or toward another one (other-directed; e.g., Forrest feeling happy

for Jenny). These reports constituted two third-person descriptions, which we used as models of the attribution of affective states to others (please refer to Supplementary Information for details).

On average, subjective ratings shared the $11.4\% \pm 8.6\%$ (standard deviation) of the variance with the self-directed emotion attribution model and the $35.3\% \pm 16.1\%$ with the other-directed model, indicating that first-person experience and characters' emotions are not completely independent. Moreover, in line with previous studies highlighting the role of right TPJ in the attribution of mental states to others[28,29,42], the other-directed—but not the self-directed—emotion attribution model significantly explained activity within this region (Supplementary Fig. 8). However, none of the first six components obtained from the other-directed emotion attribution model (i.e., 87% of the explained variance) retained a topographic organization in right TPJ (Supplementary Table 5). In addition, we used canonical correlation analysis to transform the other-directed model into the space defined by subjective emotion ratings and tested whether starting from a third-person complex description of portrayed emotions, one can fully reconstruct the brain topography of emotion dimensions. Only the first aligned

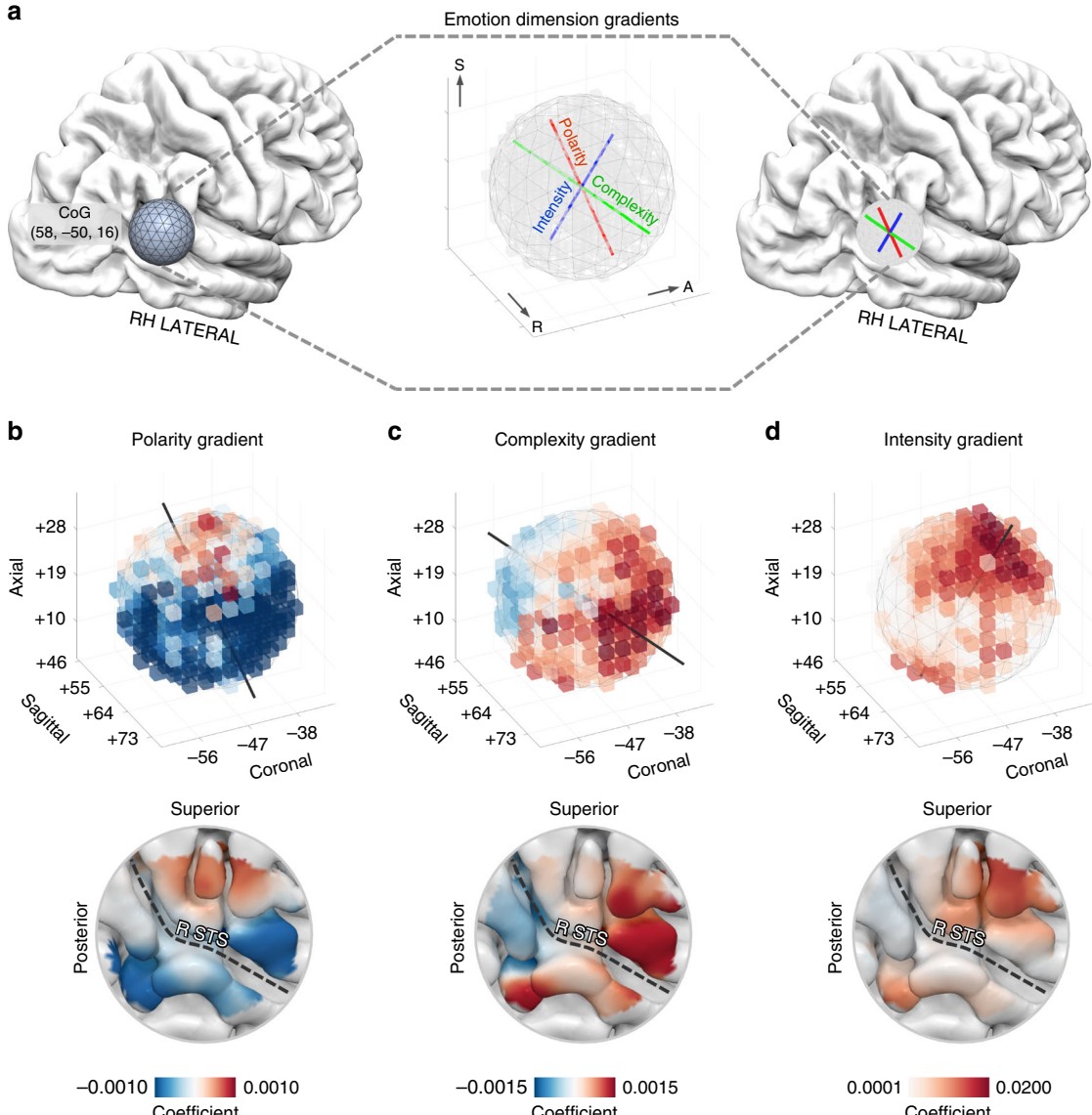

**Fig. 4 Emotion gradients in right TPJ. a** We revealed three orthogonal and spatially overlapping emotion dimension gradients (polarity, complexity and intensity) within a region of interest located at the reverse inference peak for the term TPJ (15 mm radius sphere). Symmetry axis of the region of interest represents the main direction of the three gradients. **b** $\beta$ coefficients of the polarity dimension are mapped in an inferior to superior direction. **c** $\beta$ coefficients of the complexity dimension are mapped in a posterior to anterior direction. **d** $\beta$ coefficients of the intensity dimension are mapped in an inferior to superior direction. For single-subjects results, please refer to Supplementary Fig. 3. Lowermost row depicts the arrangement of the emotion dimension gradients in surface space. CoG center of gravity, R STS right superior temporal sulcus.

component was mapped in a topographic manner within right TPJ (reconstructed polarity: $\rho = 0.221$, $p$-value $= 0.036$; reconstructed complexity: $\rho = 0.150$, $p$-value $= 0.384$; reconstructed intensity: $\rho = 0.207$, $p$-value $= 0.092$; permutation test). Overall, these results suggest that right TPJ topography is better explained by subjective reports, rather than by information coded in portrayed emotions. At the same time, they may not provide the clearest support for the interpretation that emotion dimension gradients exclusively map first-person experiences. First, in social interactions, one's affective state is often influenced by facts and events of others' life. In our study, we observe a positive correlation between first-person reports and portrayed emotions (e.g., highest sadness score when Forrest holds dying Bubba) and the lack of complete orthogonality between models prevents the precise distinction of the two. Second, real-time subjective ratings and accurate descriptions of characters' emotions are better

captured using different experimental paradigms. Indeed, our emotion ratings were continuously recorded during movie watching, whereas for portrayed emotions, individuals tagged movie scenes in a random order, choosing among a wide array of labels and were allowed to watch each excerpt more than once. In light of all this, further studies are needed to clarify whether emotion dimension gradients exclusively encode first-person experience.

**Characterization of emotion dimension gradients**. To detail how right TPJ gradients encode perceived affective states, we have reconstructed fMRI activity for movie segments connoted by either positive or negative polarity, as well as higher or lower complexity and intensity. The orientation of the three emotion dimension gradients was represented by the symmetry axis of our

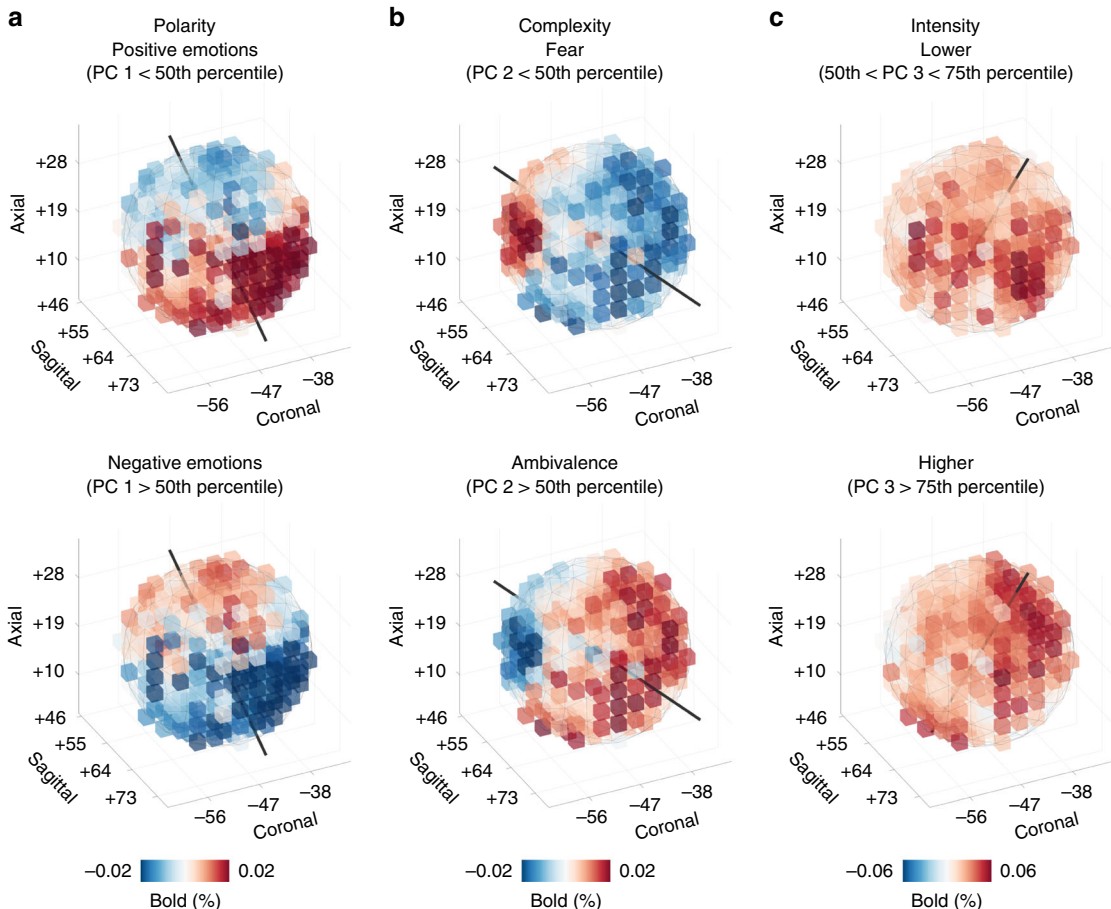

**Fig. 5 Characterization of emotion dimension gradients in right TPJ. a** Right TPJ hemodynamic activity related to the scores below and above the 50th percentile for polarity. **b** Right TPJ hemodynamic activity related to the scores below and above the 50th percentile for complexity. **c** Since intensity is not bipolar as the other two components (i.e., scores ranged from ~0 to positive values only), for this dimension we mapped the average TPJ activity above the 75th percentile and within 50th and 75th percentile. PC principal component.

ROI. For polarity, events connoted by positive emotions increased activity in ventrorostral territories, lying close to the superior temporal sulcus, whilst highly negative events augmented hemodynamic activity in dorsocaudal portions of right TPJ, extending to the posterior banks of Jensen sulcus (Fig. 5a, d).

Events connoted by higher complexity (e.g., concurrent presence of happiness and sadness) were associated to signal increments in rostrolateral territories of right TPJ, whereas those rated as having lower complexity (e.g., fearful events) increased hemodynamic activity in its caudal and medial part, encompassing the ascending ramus of the superior temporal sulcus (Fig. 5b, d). Higher levels of intensity were related to increased activity in rostrodorsal and ventrocaudal territories, reaching the ascending ramus of the lateral sulcus and posterior portions of the middle temporal gyrus, respectively. On the contrary, low-intensity events augmented hemodynamic activity in a central belt region of right TPJ, located along the superior temporal sulcus (Figs. 5c, 3d). Noteworthy, the orthogonal arrangement of polarity and complexity and the fact that intensity was represented both superiorly and inferiorly to the superior temporal sulcus determined that the variety of emotional states elicited by the Forrest Gump movie (see Fig. 2) could be mapped within a single patch of cortex.

Moreover, in sensory areas, topographies result from the maximal response of neurons to a graded stimulus feature. To parallel right TPJ emotion dimension gradients with those observed in primary sensory regions, we investigated whether

distinct populations of voxels were selective for specific polarity, complexity and intensity scores. Thus, we employed the population receptive field method[43] to estimate the tuning curve of right TPJ voxels for each emotion dimension. The maps of voxel selectivity demonstrated the existence of four populations of voxels tuned to specific polarity values, which encoded highly and mildly positive or negative events, respectively (Fig. 6a). Also, two distinct populations of voxels were tuned to maximally respond during cognitively mediated affective states (i.e., highly and mildly positive complexity values), and two other populations were selective for emotions characterized by higher and lower levels of automatic responses (i.e., highly and mildly negative complexity values; Fig. 6b). Lastly, for the intensity dimension, two specific populations of voxels were engaged depending on the strength of the emotional experience (Fig. 6c). This further evidence favored the parallel between emotion and sensory gradients.

## Discussion

Previous studies reported that activity of individual brain regions codes distinct emotion features[7], whereas others suggested that a distributed network of cortical areas conjointly interacts to represent affective states[33]. However, the possibility that gradients may encode the emotional experience as function of either basic emotions, or emotion dimensions, has never been explored. The topological isomorphism between feature space and cortical

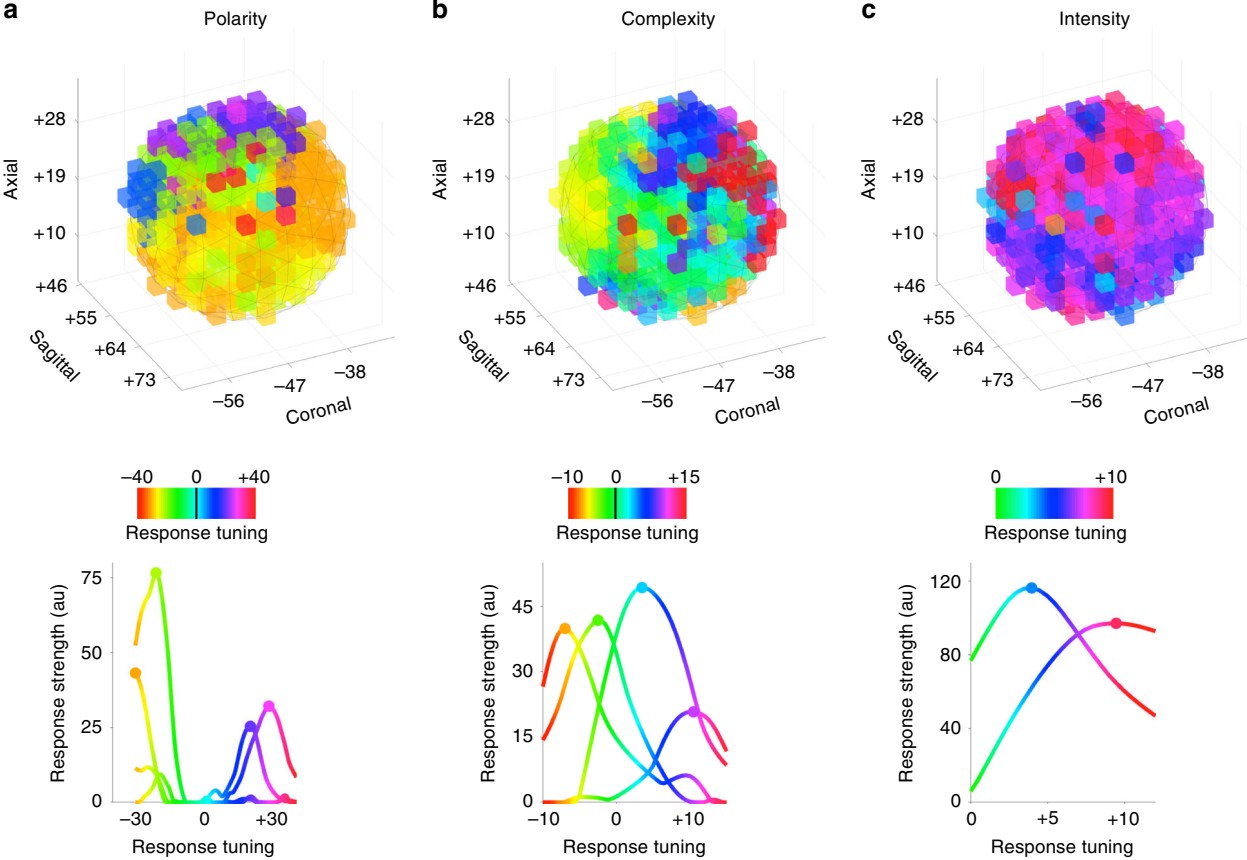

**Fig. 6 Population receptive field estimates in right TPJ.** Response selectivity maps of right TPJ voxels for **a** polarity, **b** complexity and **c** intensity. Preferred responses of distinct populations of voxels were obtained using non-negative matrix factorization (Supplementary Fig. 9). Components explaining at least 5% of the variance were plotted as a tuning curve (lowermost row) after averaging all the possible tuning width values for each emotion dimension score. The maps of voxel selectivity were consistent with the topography obtained from the original gradient estimation for the three emotion dimensions (polarity: $\rho = 0.547$, $p$-value = 0.001; complexity: $\rho = 0.560$, $p$-value < 0.001 and intensity: $\rho = 0.596$, $p$-value < 0.001; permutation test; $n = 428$ voxels).

distances has been successfully adopted to relate psychophysical characteristics of stimuli to patterns of activity in sensory regions[34]. Nonetheless, this biologically advantageous mechanism has been proven to lie at the basis of the cortical representation of higher-level features as well[35,38,39]. Thus, we tested whether different affective states could be mapped onto the cortical mantle through spatially overlapping gradients.

We demonstrated that the topography of right TPJ, a crucial cortical hub for social cognition[12,22,26,28,29], is described by emotion dimensions, rather than by single basic emotions. Indeed, within this region, we discovered three orthogonal and spatially overlapping gradients encoding the polarity, complexity and intensity of the emotional experience. The peculiar arrangement of these gradients allows a gamut of emotional experiences to be represented in a single patch of cortex, including affective states perceived as pleasant, unpleasant or ambivalent, connoted by calmness or excitement and mediated by primitive reactions or mentalization. Therefore, TPJ organization resembles the one observed in primary sensory areas, where stimulus properties are topographically arranged onto the cortical mantle, as in the case of eccentricity and polar angle in primary visual cortex (V1), frequency in primary auditory region and body parts in primary somatosensory area. In this regard, the evidence that emotion dimensions are encoded in a gradient-like manner supports a biologically plausible mechanism for the coding of affective states, which we named emotionotopy. Indeed, as in vision precise portions of V1 map distinct locations of the visual field, specific regions of temporo-parietal cortex code

unique emotional experiences. This emerged also from the analysis of response tuning, showing how within each emotional hemifield of polarity and complexity, populations of voxels code specific levels of emotional experience.

As for polar angle and eccentricity in V1, right TPJ emotion dimension gradients are lower-dimensional descriptions of the underlying neural activity. The retinotopic representation of azimuth and elevation in V1 overlaps with local maps of ocular dominance and orientation tuning. Therefore, multiple neural codes exist at different spatial scales and the ability to capture either global or local representations relates to the resolution of the imaging technique. Our data provide evidence for a lower-dimensional, yet biologically favorable, neural code to represent emotions in temporo-parietal regions. Considering the parallel with the organization of sensory areas, we believe that the topography of right TPJ does not prevent the existence of other neural codes, especially considering the coexistence of global and local representations and the multifaceted nature of this region.

Furthermore, the fact that affective reports explained the activity of other cortical modules is not necessarily in contrast with the topographic organization of TPJ. In fact, as in vision a rich and complex percept relies on both primary visual cortex to extract fundamental features and other regions to process specific stimulus properties (e.g., V5 for motion), so in emotion processing TPJ may represent a hub embedded in a distributed network of regions carrying out distinct computations.

Here, we employed a naturalistic continuous stimulation paradigm to foster emotional contagion and empathic

reactions[22]. Indeed, we found that within a 60-s time window, emotion transitions represented in the Forrest Gump movie are similar to those experienced in real life[44] and are predicted by a mental model of emotion co-occurrence[45] (see Supplementary Information). This supports the ecological validity of our stimulus and emphasizes that movies can be successfully adopted to nurture emotional resonance[46–48], also in the fMRI setting.

In movie watching, actions and dialogues generally are not directed toward the observer. Thus, the emotional experience results from narrative choices aimed at fostering empathic responses and emotional contagion[49] as well as from perspective-taking and mentalizing processes[50,51]. That character intentions and beliefs shape subjective experience in a bystander may also explain the high between-subject agreement in reports of experienced emotions[46–48]. This is in line with the consistency of behavioral ratings of happiness, fear, sadness and anger in our data. Noteworthy, surprise and disgust were not consistent across all participants and, even though this may appear in contradiction to the supposed[52] universalism of basic emotions, our stimulus was not built to reflect the well-established definition of six emotions. For instance, some of our subjects reported that movie scenes rated as disgusting were mainly associated to situations that required interpretation of the context (e.g., the principal of the school using his power to obtain sexual favors), rather than to repulsive images. This cognitive interpretation of disgust was apparently not present in all subjects, with some of them relying more on its well-established definition for their ratings. Also, the use of six distinct emotion categories allowed to compare basic emotion and emotion dimension models starting from the same data[15,17]. Moreover, while the definition of basic emotions is common across individuals, ratings based on emotion dimensions require participants to be acquainted with the meaning of psychological constructs (e.g., dominance[16]).

Nonetheless, single basic emotions provide a coarse description of subjective experiences, as affective states could emerge from psychological processes not directly reducible to single emotions[53]. Our rating model, though, does account for this possibility, as subjects were allowed to report more than one emotion at a time. This resulted in the identification of 15 distinct affective states (Fig. 2), a number compatible with previous studies[29,54]. Also, despite divergences in literature[55,56], when subjects are free to detail their personal experience—as in our case—they report a complex blend of apparently conflicting emotions as well (e.g., happiness and sadness together).

With respect to emotion dimensions, the components we identified were deliberately interpreted not following any known model. Yet, polarity mainly relates to positive against negative emotions as in valence[14], whereas intensity is unipolar and mimics arousal[14]. We considered the second component as a measure of complexity of the emotional state. Indeed, this dimension contrasts events in the movie rated as fearful, an emotion with a fast and automatic response[57], against scenes characterized by ambivalence, where cognitive processes play a significant role in generating mixed emotions[58]. Even though this component does not pertain to classical emotion dimension theories, complexity may be related to the involvement of Theory of Mind[28] in emotion perception[59]. In addition, a recent study on mental representation of emotions[45] described the "human mind" component as a cardinal dimension of the affective space. This dimension maps states "[…] purely mental and human specific vs. bodily and shared with animals", which is in line with our interpretation of complexity.

We collected behavioral emotion ratings to explain brain activity in independent subjects. In line with previous studies[10,12], results highlighted a set of regions located mainly in the right hemisphere (Fig. 3 and Supplementary Table 1). Interestingly, the

peak of association between emotion ratings and brain activity was located in right pSTS/TPJ. This area plays a central role in the attribution of mental states to others, as demonstrated by functional neuroimaging[28,29], noninvasive transcranial stimulation[60] and lesion studies[61]. In addition, this region spans across the posterior portion of the superior temporal sulcus, which is implicated in emotion perception[10,12,22,62,63]. In line with this, we showed that activity in right TPJ is significantly explained by the process of emotion attribution to others and by subjective emotional ratings. This evidence fits well with the involvement of right TPJ in the representation of subjective emotional experience[10,22,63], in empathic processes[26,27] and in the attribution of beliefs and emotions to others.[28,29,42]

In addition, in the current study, ratings of the emotional experience elicited by an American movie in Italian participants explained brain activity in German subjects. This suggests that the topographic representation of emotion dimensions exists regardless of linguistic or micro-cultural differences. Yet, the mapping of distinct emotional states within right TPJ gradients may depend on the background of each individual.

Our study presents the following limitations: first, the effect size we report for the relationship between emotion ratings and brain activity appears to be relatively small (i.e., 7% of explained variance in right TPJ). However, (1) brain regions significantly encoding emotions are selected after rigorous correction for multiple comparisons; (2) the magnitude of the effect is in line with recent fMRI literature on the coding of emotions in the brain[29] and the evaluation of the noise ceiling suggests that our emotion dimension model explains between 30% (i.e., upper bound) and 54% (i.e., lower bound) of right TPJ activity; (3) we used a parsimonious encoding model, in which only six predictors explained 3595 samplings of brain activity.

Second, although using a larger set of emotion categories the same polarity, complexity and intensity components still emerged (see Supplementary Information for details), we cannot exclude that our emotion dimensions are specific for the present stimulus. Therefore, alternative movies should be employed to test the generalizability of the topographic organization of polarity, complexity and intensity within right TPJ.

Third, our findings suggest that emotion dimension gradients are better explained considering subjective reports of the affective experience, rather than by portrayed emotions. However, the significant association between subjective ratings and characters' emotions, as well as differences in rating scales and choice of emotion categories, limit the possibility to draw clear conclusions about the encoding of subjective experiences, rather than emotion attribution processes, in right TPJ topography.

In summary, our results showed that moment-by-moment ratings of perceived emotions explain brain activity recorded in independent subjects. Most importantly, we demonstrated the existence of orthogonal and spatially overlapping right temporo-parietal gradients encoding emotion dimensions, a mechanism that we named emotionotopy.

## Methods

**Behavioral experiment**. In the present study, we took advantage of a high-quality publicly available dataset, part of the studyforrest project[40] (http://studyforrest.org), to demonstrate the existence of a gradient-like organization in brain regions coding emotion ratings. Particularly, we used moment-by-moment scores of the perceived intensity of six basic emotions elicited by an emotionally charged movie (Forrest Gump; R. Zemeckis, Paramount Pictures, 1994), as predictors of fMRI activity in an independent sample. We then tested the correspondence between the fitting of the emotion rating model in TPJ voxels and their relative spatial arrangement to reveal the existence of orthogonal spatially overlapping gradients.

To obtain moment-by-moment emotion ratings during the Forrest Gump movie, we enrolled 12 healthy Italian native speakers (5F; mean age 26.6 years, range 24–34). None of them reported to have watched the movie in 1 year period prior to the experiment. All subjects signed an informed consent to participate in

the study, had the right to withdraw at any time and received a small monetary compensation for their participation. The study was conducted in accordance with the Declaration of Helsinki and was approved by the local IRB (CEAVNO: Comitato Etico Area Vasta Nord Ovest; Protocol No. 1485/2017).

We started from the Italian dubbed version of the movie, edited following the exact same description reported in the studyforrest project (eight movie segments ranging from a duration of 11 to 18 min). The movie was presented in a setting free from distractions using a 24″ monitor with a resolution of 1920 × 1080 pixels connected to a MacBook™ Pro running Psychtoolbox[64] v3.0.14. Participants wore headphones in a noiseless environment (Sennheiser™ HD201; 21–18,000 Hz; Maximum SPL 108 dB) and were instructed to continuously rate the subjective perceived intensity (on a scale ranging from 0 to 100) of six basic emotions[5] throughout the entire movie: happiness, surprise, fear, sadness, anger and disgust. Specific buttons mapped the increase and decrease in intensity of each emotion and subjects were instructed to represent their inner experience by freely adjusting or maintaining the level of intensity. Participants were allowed to report more than one emotion at the same time and ratings were continuously recorded with a 10-Hz sampling rate. Subjects were presented with the same eight movie segments employed in the fMRI study one after the other, for an overall duration of 120 min. Further, before starting the actual emotion rating experiment, all participants performed a 20-min training session to familiarize with the experimental procedure. Specifically, they had to reproduce various levels of intensity for random combinations of emotions that appeared on the screen every 10 s.

For each subject, we recorded six timeseries representing the moment-by-moment perceived intensity of basic emotions. First, we downsampled timeseries to match the fMRI temporal resolution (2 s) and, afterward, we introduced a lag of 2 s to account for the delay in hemodynamic activity. The resulting timeseries were then temporally smoothed using a moving average procedure (10 s window). This method allowed us to further account for the uncertainty of the temporal relationship between the actual onset of emotions and the time required to report the emotional state.

To verify the consistency in the occurrence of affective states while watching the Forrest Gump movie, we computed the Spearman's $\rho$ correlation coefficient across subjects for each of the six ratings (Fig. 1b). Statistical significance of the agreement was assessed by generating a null distribution of random ratings using the IAAFT procedure (Iterative Amplitude Adjusted Fourier Transform[65]; Chaotic System Toolbox), which provided surrogate data with the same spectral density and temporal autocorrelation of the averaged ratings across subjects (1000 surrogates).

Preprocessed and temporally smoothed single-subject emotion ratings were averaged to obtain six group-level timeseries representing the basic emotion model. After measuring the Spearman's $\rho$ between pairings of basic emotions (Fig. 1b), we performed PC analysis and identified six orthogonal components, which constituted the emotion dimension model (Fig. 1c).

To verify the consistency across subjects of the PCs, we computed the agreement of the six components by means of a leave-one-subject-out cross validation procedure (Fig. 1d). Specifically, for each iteration, we performed PC analysis on the left-out subject behavioral ratings and on the averaged ratings of all the other participants. The six components obtained from each left-out subject were rotated (Procrustes analysis, reflection and orthogonal rotation only) to match those derived from all the other participants. This procedure generated for each iteration (i.e., for each of the left-out subjects) six components, which were then compared across individuals using Spearman's $\rho$, similarly to what has been done for the six basic emotions. To assess the statistical significance, we created a null distribution of PCs from the generated surrogate data of the behavioral ratings, as described above (1000 surrogates).

Although subjects were asked to report their inner experience using six emotion categories, their ratings were not limited to binary choices. Indeed, at each timepoint raters could specify the perceived intensity of more than one emotion, leading to the definition of more complex affective states as compared to the basic ones. To further highlight this aspect, we performed dimensionality reduction and clustering analyses on emotion timeseries. Starting from emotion ratings averaged across participants, we selected timepoints characterized by the highest intensity (i.e., by summing the six basic emotions and setting the threshold to the 50th percentile) and applied Barnes–Hut t-distributed stochastic neighbor embedding[56,66] (t-SNE; perplexity = 30; theta = 0.05). The algorithm measures the distances between timepoints in the six-dimensional space defined by the basic emotions as joint probabilities according to a Gaussian distribution. These distances are projected onto a two-dimensional embedding space using a Student's t probability distribution and by minimizing the Kullback–Leibler divergence. To further describe the variety of affective states elicited by the movie, we then applied k-means clustering analysis to the projection of timepoints in the t-SNE manifold and determined the number of clusters using the silhouette criterion[67].

**fMRI experiment.** We selected data from the phase II of the studyforrest project, in which 15 German mother tongue subjects watched an edited version of the Forrest Gump movie during the fMRI acquisition. Participants underwent two 1-h sessions of fMRI scanning (3T, TR 2 s, TE 30 ms, FA 90°, 3 mm ISO, FoV 240 mm, 3599 tps), with an overall duration of the experiment of 2 h across eight runs. Subjects were instructed to inhibit any movement and simply enjoy the movie (for further details[40]). We included in our study all participants that underwent the

fMRI acquisition and had the complete recordings of the physiological parameters (i.e., cardiac trace) throughout the scanning time (14 subjects; 6F; mean age 29.4 years, range 20–40 years). For the fMRI pre-processing pipeline, please refer to Supplementary Information.

**Encoding analysis.** Voxel-wise encoding[68,69] was performed using a multiple linear regression approach to measure the association between brain activity and emotion ratings, constituted by the six PCs. Of note, performing a least-square linear regression using either the six PCs or the six basic emotion ratings yields the same overall fitting (i.e., full model $R^2$), even though the coefficient of each column could vary among the two predictor sets.

To reduce the computational effort, we limited the regression procedure to gray matter voxels only (~44 k with an isotropic voxel resolution of 3 mm). We assessed the statistical significance of the $R^2$ fitting of the model for each voxel using a permutation approach, by generating 10,000 null encoding models. Null models were obtained by measuring the association between brain activity and surrogate data having the same spectral density and temporal autocorrelation of the original six PCs. This procedure provided a null distribution of $R^2$ coefficients, against which the actual association was tested. The resulting $p$-values were corrected for multiple comparisons using the FDR[70] method ($q < 0.01$; Fig. 3a, Supplementary Fig. 1 and Supplementary Table 1). $R^2$ standard error was calculated through a bootstrapping procedure (1000 iterations). Moreover, we conducted a noise-ceiling analysis for right TPJ data, similarly to what has been done by Ejaz and colleagues[71] (please see Supplementary Methods).

**Emotion gradients in right TPJ.** We tested the existence of emotion gradients by measuring the topographic arrangement of the multiple regression coefficients[72] in regions lying close to the peak of fitting for the encoding procedure (i.e., right pSTS/TPJ). To avoid any circularity in the analysis[73], we first delineated an ROI in the right pSTS/TPJ territories using an unbiased procedure based on the Neuro-Synth[74] database v0.6 (i.e., reverse inference probability associated to the term TPJ). Specifically, we started from the peak of the TPJ NeuroSynth reverse inference meta-analytic map to draw a series of cortical ROIs, with a radius ranging from 9 to 27 mm. Afterward, to identify the radius showing the highest significant association, for each spherical ROI we tested the relationship between anatomical and functional distance[75] (Supplementary Table 2). This procedure was performed using either multiple regression coefficients obtained from the three emotion dimensions or from the four basic emotions stable across all subjects. As depicted in Supplementary Fig. 2, we built for each radius two dissimilarity matrices: one using the Euclidean distance of voxel coordinates, and the other one using the Euclidean distance of the fitting coefficients (i.e., $\beta$ values) of either the three emotion dimensions or the four basic emotions. The rationale behind the existence of a gradient-like organization is that voxels with similar functional behavior (i.e., lower functional distance) would also be spatially arranged close to each other on the cortex[75] (i.e., lower physical distance). The functional and anatomical dissimilarity matrices were compared using the Spearman's $\rho$ coefficient. To properly assess the significance of the anatomo-functional association, we built an ad hoc procedure that maintained the same spatial autocorrelation structure of TPJ in the null distribution. Specifically, we generated 1000 IAAFT-based null models for the emotion dimension and the basic emotion data, respectively. These null models represented the predictors in a multiple regression analysis and generated a set of null $\beta$ regression coefficients. Starting from these coefficients, we built a set of functional dissimilarity matrices that have been correlated to the anatomical distance and provided 1000 null Spearman's $\rho$ coefficients, against which the actual anatomo-functional relationship was tested. Confidence intervals (CI; 2.5 and 97.5 percentile) for the obtained correlation values were calculated employing a bootstrap procedure (1000 iterations). We also tested the existence of gradients in other brain regions encoding emotion ratings using a data-driven searchlight analysis. Results and details of this procedure are reported in Supplementary Information.

To estimate the significance of right TPJ gradients, we used null models built on emotion ratings, leaving untouched the spatial and temporal structure of brain activity. However, as spatial smoothness may still affect the estimation of gradients, we tested right TPJ topography using the group-average unfiltered data. In brief, all the steps described in the fMRI data pre-processing section (see Supplementary Methods) were applied, with the only exception of spatial filtering. Following this procedure, the estimated smoothness of the right TPJ region was $4.5 \times 4.2 \times 3.6$ mm (3dFWHMx tool). Using these data and the same procedure described above, we measured the significance of emotion gradients. Results are detailed in Supplementary Table 6 and Supplementary Fig. 7.

The Euclidean metric does not take into account cortical folding. Indeed, because of the morphological characteristics of TPJ, which include a substantial portion of STS sulcal walls, the estimation of emotion gradients would benefit from the use of a metric respectful of cortical topology. For this reason, we ran the Freesurfer recon-all analysis pipeline[76] on the standard space template[77] used as reference for the nonlinear alignment of single-subject data. We then transformed the obtained files in AFNI-compatible format (@SUMA_Make_Spec_FS). This procedure provided a reconstruction of the cortical ribbon (i.e., the space between pial surface and gray-to-white matter boundary), which has been used to measure the anatomical distance. In this regard, we particularly employed the Dijkstra algorithm as it represents a computationally efficient method to estimate cortical distance based on folding[78,79].

The single-subject unsmoothed timeseries were then transformed into the standard space, averaged across individuals and projected onto the cortical surface (AFNI 3dVol2Surf, map function: average, 15 steps). Afterward, we performed a multiple linear regression analysis using PCs derived from emotion ratings as predictors of the unsmoothed functional data. This analysis was carried out within a cortical patch that well approximated the size of the 3D-sphere used in the original volumetric pipeline and centered at the closest cortical point with respect to the Neurosynth TPJ peak. Thus, for each regressor of interest, we obtained unsmoothed $\beta$ values projected onto the cortical mantle. We then tested the existence of a gradient-like organization for each predictor, using the Dijkstra algorithm and the same procedure described above. Results are detailed in Supplementary Table 6 and Fig. 4.

**Right temporo-parietal gradients and portrayed emotions**. We tested whether the gradient-like organization of right TPJ reflects portrayed emotions. Thus, we took advantage of publicly available emotion tagging data of the same movie, provided by an independent group[41]. Differently from our behavioral task, raters were asked to indicate the portrayed emotion of each character (e.g., Forrest Gump, Jenny) in 205 movie segments (average duration ~35 s) presented in random order and labeled over the course of ~3 weeks. As also suggested by the authors[41], this particular procedure minimizes carry-over effects and help observers to exclusively focus on indicators of portrayed emotions. Importantly, in their study, the authors respected the narrative of movie scenes (e.g., Forrest in front of Jenny's grave is a single cut with a duration of ~131 s), so that the raters could have a clear understanding of what was shown on the screen. In addition, the possibility to tag emotions independently in each movie segment and to watch each scene more than once, allowed subjects to choose among a larger number of emotion categories[80] ($N = 22$), as compared to our set of emotions. Moreover, each observer was instructed to report with a binary label whether the portrayed emotion was directed toward the character itself (self-directed; e.g., Forrest feeling sad) or toward another character (other-directed; e.g., Forrest feeling happy for Jenny). These two descriptions served as third-person emotion attribution models and underwent the exact same processing steps (i.e., 2 s lagging and temporal smoothing), which have been applied to our subjective emotion rating model. As the two third-person emotion attribution models included the four basic emotions found to be consistent across observers in our experiment (i.e., happiness, fear, sadness and anger), we have been able to directly assess the correlation for these ratings using Spearman's $\rho$.

We then measured the extent to which the two third-person emotion attribution models explained brain activity in right TPJ following the method described in the "Encoding analysis" section. As these two descriptions are higher in dimensionality as compared to our subjective emotion rating model, we assessed the significance of fitting using three different procedures: (A) matching the dimensionality across models by selecting the first six PCs only; (B) matching the emotion categories in ratings, by performing PCA on the four basic emotions shared across models (i.e., happiness, fear, sadness and anger); (C) using the full model regardless of the dimensionality (i.e., six components for our subjective emotion rating model and 22 for each of the two emotion attribution models). In addition, to allow a direct and unbiased comparison between $R^2$ values obtained from different models, we performed cross-validation using a half-run split method (Supplementary Fig. 8).

Lastly, we tested whether right TPJ gradients encode emotion attribution models. Specifically, we evaluated two different scenarios: (1) the existence of right TPJ gradients encoding the 22 components of each emotion attribution model; (2) the possibility to identify emotion gradients following the multidimensional alignment[81] (i.e., canonical correlation analysis) of the 22-dimensional emotion attribution space to the six-dimensional space defined by subjective ratings. These alternative procedures relate to two different questions: (1) whether the process of emotion attribution is associated to emotion gradients in right TPJ and (2) whether starting from a third-person complex description of portrayed emotions, one can reconstruct the subjective report of our raters. Results for these two procedures are detailed in Supplementary Table 5.

**Characterization of emotion gradients in right TPJ**. Once the optimal ROI radius was identified, we tested the gradient-like organization of right TPJ for each individual emotion dimension and basic emotion (Supplementary Table 3), using the same procedure described above. We calculated the numerical gradient of each voxel using $\beta$ values. This numerical gradient estimates the partial derivatives in each spatial dimension ($x$, $y$, $z$) and voxel, and can be interpreted as a vector field pointing in the physical direction of increasing $\beta$ values. Afterward, to characterize the main direction of each gradient, rather than calculating its divergence (i.e., Laplacian of the original data[82,83]), we computed the sum of all vectors in the field (Fig. 4 and Supplementary Fig. 2). This procedure is particularly useful to reveal the principal direction of linear gradients and provides the opportunity to represent this direction as the orientation of the symmetry axis of the selected ROI. The above-mentioned procedure was also adopted to assess the reliability of the emotion gradients in each subject. Results and details of this procedure are reported in Supplementary Information. Furthermore, since gradients built on $\beta$ coefficients could reflect positive or negative changes in hemodynamic signal depending on the sign of the predictor, we represented the average TPJ activity during movie scenes characterized by specific affective states (Fig. 5).

We investigated whether distinct populations of voxels are selective for specific affective states. To this aim, we employed the population receptive field method[43] (pRF) and estimated the tuning curve of right TPJ voxels for each predictor found to be topographically encoded within this region. We modeled the tuning curve of each voxel as a Gaussian distribution, in which $\mu$ represented the preferred score of the predictor and $\sigma$ the width of the response. The optimal combination of tuning parameters was selected among ~5k plausible values of $\mu$ (5th–95th percentile of the scores of each predictor—0.5 step) and $\sigma$ (ranging from 1 to 12—0.25 step), sampled on a regular grid. Each emotion timeseries was then filtered using these ~5k Gaussian distributions and fitted in brain activity through a linear regression approach. This produced $t$-values (i.e., $\beta$/SE $\beta$) expressing the goodness of fit of $\mu$ and $\sigma$ combinations, for each right TPJ voxel. The principal tuning of voxels was then obtained by selecting the combination characterized by the highest $t$-value across the ~5k samples (Fig. 6).

To estimate the similarity between tunings (i.e., $\mu$ parameters) obtained from the pRF approach and our original results (i.e., $\beta$ coefficients of the gradient estimation), we computed Spearman's $\rho$ across right TPJ voxels. The significance of such an association was tested against a null distribution of $\beta$ coefficients obtained through the IAAFT procedure ($N = 1000$).

Lastly, we further characterized the prototypical responses of populations of voxels as function of affective states. To do so, we used the non-negative matrix factorization[84] and decomposed the multivariate pRF data (i.e., voxels $t$-values for each $\mu$ and $\sigma$) into an approximated matrix of lower rank (i.e., 10, retaining at least 90% of the total variance). This method allows parts-based representations, as the tuning of right TPJ voxels is computed as a linear summation of non-negative basis responses. The results of this procedure are summarized in Supplementary Fig. 9.

All the analyses were performed using MATLAB R2016b (MathWorks Inc., Natick, MA, USA).

**Reporting summary**. Further information on research design is available in the Nature Research Reporting Summary linked to this article.

## Data availability
The data that support the findings of this study are available from OpenScience foundation: Emotionotopy (https://osf.io/tzpdf). Raw fMRI data are available atstudyforrest.org (http://studyforrest.org). Real life experience-sampling dataset is available at https://osf.io/zrdpa. Portrayed emotions dataset is available at F1000 repository (https://f1000research.com/articles/4–92/v1). A reporting summary for this article is available as a Supplementary Information file.

## Code Availability
The code and the preprocessed data are publicly available at OpenScience foundation: Emotionotopy (https://osf.io/tzpdf).

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

## Acknowledgements

We would like to thank all the people behind the studyforrest project, especially Michael Hanke, Annika Labs and colleagues for sharing the Forrest Gump portrayed emotions data and Mark A. Thornton and Diana I. Tamir for making available the real life emotion transitions data. We also thank Maria Luisa Catoni for her valuable suggestions. M.B. was in part supported by the PRIN (Research projects of national interest) 2015—Italian Ministry of Education, University and Research (MIUR) Prot. 2015AR52F9 granted to P.P.

## Author contributions

G.L., G.H. and L.C. conceived the study, designed the behavioral experiment, developed the code, performed behavioral and fMRI data analysis. G.L., G.H., L.C. and P.P. interpreted the obtained results and drafted the manuscript. A.L., Pa.P. and M.B. contributed to the fMRI data analysis. L.C., E.R. and P.P. critically revised the manuscript. All the authors approved the final version of the manuscript.

## Competing interests

The authors declare no competing interests.
