## [Peer Review File · Nature Communications]

Reviewers' Comments:

Reviewer #1:

Remarks to the Author:

The manuscript by Lettieri and colleagues investigates the topographic representation of emotional states, describing three emotion dimensions (polarity, complexity, intensity) within the right temporoparietal junction (TPJ). The study combines behavioral data with the openly available fMRI data from the studyforrest project to characterize the representation of emotion dimensions. The motivation and results are clearly presented, and additional supplementary analyses on the individual-level provide a valuable addition to the current study. However, I have the following methodological suggestions, which the authors may wish to consider:

- The 'cortical mantle' is described as the space of the gradients described in the current study. However, the analyses appear to be conducted entirely in volumetric space. Have the authors considered running their analyses in surface-space to ensure that the affective dimensions map onto meaningful spatial representations of cortical space? The morphological characteristics of TPJ, which include the superior temporal sulcus, may introduce substantial artifacts into the analysis and results, especially due to smoothing over adjacent sulcal walls as well as the functional distinctions across sulci. (The results presented in Supplementary Figure 7 suggest that the sulcal position and orientation may in fact play a critical role in the main results.) Considering the examples provided by the authors of other topographic organizational patterns in the cortex treat the cortical surface as a two-dimensional sheet, I would urge the authors to consider the same approach for their analysis here.

- A searchlight analysis is conducted to investigate whether the three combined dimensions are also represented within other portions of the cortex, however, have the same analyses been applied to investigate whether individual dimensions are represented elsewhere? While the mappings may differ, there would likely be repeated topographies throughout the cortex, even if TPJ is the only region to encode all three dimensions. Further investigation of the presence of these dimensions (in individual or combined forms) across the cortical surface would provide greater insight into the neural representation of emotional states.

Reviewer #2:

Remarks to the Author:

Lettieri et al. collected emotion ratings from 12 subjects watching clips of Forrest Gump that had been overdubbed with Italian. Subjects were instructed to continuously report on their first-person subjective experience of six emotions (happiness, surprise, fear, sadness, anger and disgust) during the viewing. These ratings were decomposed into principal components, 3 of which were found to be individually reliable. Voxel activity in an independent set of 14 subjects watching the same video clips (this time overdubbed with German) was then modeled using these six orthogonal components. Activation gradients were computed based on the coefficients from the multiple regression. The authors report that activity in the RTPJ reflects continuous gradients of encoding strength of the three reliable principal components (which they call 'polarity', 'complexity' and 'intensity'). These three gradients are off-axis relative to each other, potentially allowing the activation of a cortical location in the RTPJ to represent a vector in the feature space spanned by the principal components of behavioral ratings. These results are potentially interesting, and compatible with previous similar work.

However, the authors claim that the RTPJ encodes people's first-person subjective experience of emotion using this basis set. This claim goes far beyond what can be established from their data.

First, though the authors claim to be studying first person emotional experience, they did not control for third-person emotion attributions. The RTPJ plays a central role in the attribution of mental states to other agents. The authors used stimuli of emotionally engaging narratives about other people that are bound to evoke strong emotion attributions to the portrayed characters. If the emotions attributed to the characters positively correlate with the emotions experienced by the raters, it is entirely plausible that the signals modeled in the RTPJ reflect third party emotion attributions rather than the first person subjective experience of emotion, which is highly consistent with prior work on the RTPJ representation of others' emotions. The manuscript's introduction does not make a strong case that we should expect the RTPJ to encode first person subjective emotional experience based on prior work and the study does not adequately demonstrate that first person representation is the correct interpretation of the results.

To claim that these results specifically reflect dimensions of first-person emotional experience, the authors would need to demonstrate that the results are not due to the fMRI subjects' emotion attributions to the characters in the film. The authors could make this claim if behavioral ratings of the film characters' emotions departed substantially from the first person emotion ratings already collected, and if similar neural results cannot be obtained based on the 3rd person emotion ratings. The authors could have also used stimuli that evoked emotions sans engaging narratives of other agents. As the study stands, unfortunately, the first and third person alternatives are completely confounded. Coupled with extensive prior evidence that the RTPJ is specifically recruited to differentiate first- from third-person experience, the authors have not made a strong case that the neural correlates reported reflect the psychological phenomena they claim.

Second, the authors smooth the BOLD time series with a 6mm FWHM Gaussian kernel. Is it possible that the gradients are artifacts of smoothing activation peaks? The gradients of visual cortex that the authors liken their results to are mapped using voxels that exhibit a maximal response to a graded feature of the stimulus (e.g. voxels that respond maximally to stimulus features at a particular eccentricity or polar angle). The gradients used in this study estimate "the partial derivatives in each spatial dimension (x, y, z) and voxel, and can be interpreted as a vector field pointing in the physical direction of increasing β values." It seems that smoothing a peak activation would necessarily produce a gradient towards the mode, and that if the peak is near the boundary of the meta-analytic ROI it would appear to be a gradient across the region. This seems consistent with the activation maps (except the 'intensity' component has multiple peaks near the RTPJ). I presume that the authors considered this and I am simply not understanding how they made the gradient analysis robust to it. Thus, the manuscript would benefit from further elaboration on how the gradient estimation used here compares with the estimation of cortical gradients in sensory cortices and how this study's analysis protects against the case outlined above.

More generally, the voxel size of fMRI means that the underlying signal is necessarily smoothed, and the dimensions detected are very likely smoother and lower dimensional than the underlying neural code.

The authors report a peak variance explained in the RTPJ of $\rho^2 = 0.07$ and compare that to other research on emotion representation in the RTPJ (Skerry & Saxe 2015), which reported correlations of similar strength. However, Kendall's τ is typically smaller than the Spearman's ρ coefficient for the same data so the proportion of variance explained in this study might be still be substantially less than the comparison study (which also used 1.5x as many subjects). More generally, this number suggests that there is substantial variance within this region that is not yet explained by the authors' theory.

The authors state that, "...the orthogonal arrangement of polarity and complexity in right TPJ and the

fact that intensity was represented both superiorly and inferiorly to the superior temporal sulcus determined that all the possible combinations of emotional states elicited by the 'Forrest Gump' movie could be mapped within this region." The authors should be much more careful about asserting that the emotions they measured represent the full space of emotions, especially considering that the 3 components that the authors use do not even capture the disgust and surprise ratings in their data and other work they cite (Cowen & Keltner 2017) report high dimensional emotion representations.

Reviewer #3:

Remarks to the Author:

Lettieri, Handjaras, et al. present an investigation of the topographic organization of emotion representation across the cortex. They draw upon a rich open fMRI data set to examine how emotional experiences are encoded in activity across the temporoparietal junction (TPJ). They find that three overlapping yet orthogonal gradients encode the polarity, intensity, and complexity of participants emotions. The research topic is of theoretical interest to a wide range of psychological and neural scientists. The methods are sophisticated, and the paper is well and clearly written. Thus I believe this work could make a substantial contribution to the literature. However, below I raise a number of points which I believe the manuscript would benefit from addressing:

1) The authors derived three emotions dimensions by applying a PCA to ratings of six basic emotions across the movie Forrest Gump. Although this is a straightforward way to address this problem, it raises several concerns:

a. The PCs produced by this procedure inevitably depend to some extent on the particular emotions the authors choose to have rated. Although there is a theoretical justification for the six states in question – as “basic emotions” – these states omit a wide range of important emotional states, such as social/secondary emotions like pride and envy. It would be helpful to know whether the same dimensions emerge when a broader set of states are rated.

b. Related to (a), the PCs extracted from the movie ratings also depend on the qualities of the stimulus itself. Forrest Gump is well known for being both an emotionally evocative movie, and one with highly varied content, which makes it a prudent choice in the present context. However, I doubt that it or indeed any individual movie could come close to covering the full range of human emotion. Moreover, the temporal structure of emotion may differ considerably between movies in general, as opposed to real life. Presumably this is part of why we are often willing to pay money to watch a movie, but would probably not pay so much to watch a random slice of someone’s actual life. Perhaps the authors could compare temporal dynamics observed in their rating data to available experience-sampling data sets to assess how well their stimulus reflects real life experience?

c. By conducting the PCA on ratings of the movie itself, these dimensions are in some sense overfitted to this particular stimulus. For example, if factor structure/loadings were derived from ratings of separate movies (or even non-movie stimuli), and then applied to the present data, I imagine that they would explain less of the variance in both the basic emotion ratings and the fMRI data. The authors should note this caveat, perhaps in relation to their more general discussion of how much variance their model captures.

d. In PCA, once the number of components has been specified, any rotation of the retained components will explain the same total variance. How can we know that the rotation the authors consider is the “canonical” rotation of these dimensions? A recent preprint (<https://psyarxiv.com/6dvn3/>) makes this point at length in a fairly similar context: topographic maps of facial expressions of emotion across the FFA. The authors might try testing rotations of their components to see whether they produce better or worse gradients across the TPJ. Indeed, the search for neural gradients might suggest an interesting way to establish which rotations are canonical, which would be a valuable methodological contribution in itself.

2) The region the authors consider as the "TPJ" is very large – much larger than this region typically appears in the literature. It includes substantial portions of parietal and occipital cortex well outside of what would usually be called the TPJ (e.g., as defined using a false-belief localizer). I do not think this is necessarily problematic from an analytic point of view, but I do think it may give casual readers the wrong impression of the spatial extent of the observed patterns. I think the authors should acknowledge this discrepancy more explicitly, and make it clear from the beginning (i.e. in the title or the abstract) just how extensive these gradients appear. However, they can emphasize at the same time that this result generalizes across a range of spatial scales (as demonstrated in the results reported in supplementary table 2).

3) The TPJ is also a region which is more typically implicated in understanding others' thoughts and feelings (i.e., theory of mind) than in the actual experience of emotion. As the authors point out, one way in which movies elicit emotions in people is through empathy with the characters. However, these facts together suggest a possible confound: the emotions that participants rated may be highly associated with the emotions they perceive the characters to experience. Such a confound would complicate the interpretation of the present results: do the TPJ emotion gradients encode one's own emotional experience, or the perceived emotional experience of others? Either result would be interesting, but it is important to know which account is better supported. One straightforward way to address this would be to ask additional movie viewers to rate the characters' emotions, rather than their own. This would allow the authors to measure the extent of this potential confound, and potentially statistically control for it.

4) The authors tested Italian rater's emotional experiences in German speaking participants' brains, while each watched an American movie. The success of the encoding model across these linguistic and cultural boundaries is impressive and might be emphasized further. It might be interesting to discuss how other/larger cross-cultural differences might qualify the conclusions of this investigation. Ample evidence demonstrates that emotional experience and expression differ substantially across cultures – how might such differences potentially be reflected in the organization of cortex?

5) The authors raise the low R2 of their model as a potential limitation. Given that they have data from multiple participants watching/rating the same movie, it seems as if they have the necessary data to compute the reliability of both emotion dimensions and neural activity. These reliabilities could then be used to perform a noise-ceiling/disattenuation analysis. Knowing how much reliable variance is out there to explain would help to contextualize whether the observed variance-explained is really low or high.

Reviewer #1 (Remarks to the Author):

The manuscript by Lettieri and colleagues investigates the topographic representation of emotional states, describing three emotion dimensions (polarity, complexity, intensity) within the right temporo-parietal junction (TPJ). The study combines behavioral data with the openly available fMRI data from the studyforrest project to characterize the representation of emotion dimensions. The motivation and results are clearly presented, and additional supplementary analyses on the individual-level provide a valuable addition to the current study. However, I have the following methodological suggestions, which the authors may wish to consider:

[1] The 'cortical mantle' is described as the space of the gradients described in the current study. However, the analyses appear to be conducted entirely in volumetric space. Have the authors considered running their analyses in surface-space to ensure that the affective dimensions map onto meaningful spatial representations of cortical space? The morphological characteristics of TPJ, which include the superior temporal sulcus, may introduce substantial artifacts into the analysis and results, especially due to smoothing over adjacent sulcal walls as well as the functional distinctions across sulci. (The results presented in Supplementary Figure 7 suggest that the sulcal position and orientation may in fact play a critical role in the main results.) Considering the examples provided by the authors of other topographic organizational patterns in the cortex treat the cortical surface as a two-dimensional sheet, I would urge the authors to consider the same approach for their analysis here.

Response: We thank the Reviewer for the positive evaluation of our manuscript and we recognize the relevance of the two points raised. To address these concerns, we evaluated the significance of anatomo-functional gradients using unsmoothed fMRI data and estimates of anatomical distance based on cortical folding. These additional analyses prove that spatial smoothing does not affect the significance of right TPJ gradients (please see response to point 2 of Reviewer #2) and that the topography of *emotion dimensions* is preserved when considering cortical folding.

In brief, we applied the Freesurfer *recon-all* analysis pipeline (Reuter et al., 2012) to the standard space template (Fonov et al., 2009) used as reference for the nonlinear

alignment of single-subject data. Afterwards, the reconstruction of the cortical ribbon (i.e., the space between pial surface and gray-to-white matter boundary) was transformed in AFNI-compatible format using the `@SUMA_Make_Spec_FS` script. We then opted for the Dijkstra algorithm to obtain estimates of anatomical distance based on cortical folding (Fischl et al., 1999; Van Essen et al., 2011). The figure below clearly shows that while the Euclidean distance does not respect right TPJ topology, the Dijkstra algorithm provides an adequate measure of cortical distance within this region.

In addition, the single-subject unsmoothed timeseries were transformed into the standard space, averaged across individuals and projected onto the cortical surface (AFNI 3dVol2Surf, map function: average, 15 steps). Following the procedure adopted for volumetric data, we fitted the *emotion dimension model* in group-average cortical activity. Thus, for each of the six regressors of interest (i.e., the six PCs), we obtained unsmoothed β values projected onto the cortical mantle.

The correspondence between unfiltered functional data and cortical distance was tested considering a patch of cortex comparable in size to the original volumetric definition of right TPJ. Also, the center of this ROI was located at the closest cortical point with respect to the Neurosynth reverse inference peak for the term "TPJ".

Importantly, results support the original findings: using the cortical projection of unsmoothed functional data we found that *polarity* (Spearman's $\rho = 0.248$, $p = 0.026$; CI: 0.238-0.257), *complexity* ($\rho = 0.314$, $p = 0.001$; CI: 0.304-0.323) and *intensity* (ρ

= 0.249, $p = 0.013$; CI: 0.239-0.258) dimensions are mapped in right TPJ through orthogonal and spatially overlapping gradients.

We believe that these new findings corroborate our original results and prove that right TPJ *emotion dimension* gradients are robust to smoothing artifacts and are mapped onto meaningful spatial representations of cortical space. We have now modified Figure 4 to include the results of surface space analyses and updated the *Methods, Results* and *Supplementary Materials* of the revised version of the manuscript.

[2] *A searchlight analysis is conducted to investigate whether the three combined dimensions are also represented within other portions of the cortex, however, have the same analyses been applied to investigate whether individual dimensions are represented elsewhere? While the mappings may differ, there would likely be repeated topographies throughout the cortex, even if TPJ is the only region to encode all three dimensions. Further investigation of the presence of these dimensions (in individual or combined forms) across the cortical surface would provide greater insight into the neural representation of emotional states.*

Response: We thank the Reviewer for this suggestion and we agree that it would be of interest to search for individual *emotion dimension* topographies in regions encoding the *emotion rating model*.

To do this, we ran three separate searchlight analyses (see *Supplementary Materials* for details), measuring the topographic arrangement of *polarity*, *complexity* and *intensity*. The resulting log(p-value) maps were then combined into a comprehensive description of the distribution of gradients across the brain. Indeed, this procedure highlighted regions predominantly involved either in *polarity*, *complexity* or *intensity*, as well as in any combination of the three.

Results confirm that the area of maximum overlap for *emotion dimension* gradients is located within the right TPJ/pSTS region. This is also in line with Nummenmaa et al., 2012 paper, in which brain networks encoding *valence* and *arousal* dimensions overlapped in the bilateral pSTS/pSTG (please see green-colored regions in Figure 3 of Nummenmaa et al., 2012).

In addition, *polarity*, *complexity* and *intensity* maps revealed other interesting topographies: regions as the right preCS represented the three *emotion dimensions* in

distinct - yet adjoining - subregions, whereas the right OTS encoded overlapping gradients of *complexity* and *intensity* (Supplementary Figure 6). A representation of the distribution of *emotion dimension* gradients across brain regions encoding the *emotion rating model* is now reported in Supplementary Figure 6 and discussed in the *Results* section of the revised version of the manuscript.

Further studies are needed to explore emotion topographies highlighted here: previous studies already pointed to the existence of distinct and partially overlapping networks encoding single emotion dimensions (Nummenmaa et al., 2012), yet a comprehensive description of the functional interactions among regions of these networks is still lacking (Raz et al., 2016).

Reviewer #2 (Remarks to the Author):

Lettieri et al. collected emotion ratings from 12 subjects watching clips of Forrest Gump that had been overdubbed with Italian. Subjects were instructed to continuously report on their first-person subjective experience of six emotions (happiness, surprise, fear, sadness, anger and disgust) during the viewing. These ratings were decomposed into principal components, 3 of which were found to be individually reliable. Voxel activity in an independent set of 14 subjects watching the same video clips (this time overdubbed with German) was then modeled using these six orthogonal components. Activation gradients were computed based on the coefficients from the multiple regression. The authors report that activity in the RTPJ reflects continuous gradients of encoding strength of the three reliable principal components (which they call 'polarity', 'complexity' and 'intensity'). These three gradients are off-axis relative to each other, potentially allowing the activation of a cortical location in the RTPJ to represent a vector in the feature space spanned by the principal components of behavioral ratings. These results are potentially interesting, and compatible with previous similar work.

However, the authors claim that the RTPJ encodes people's first-person subjective experience of emotion using this basis set. This claim goes far beyond what can be established from their data.

[1] First, though the authors claim to be studying first person emotional experience, they did not control for third-person emotion attributions. The RTPJ plays a central role in the attribution of mental states to other agents. The authors used stimuli of emotionally engaging narratives about other people that are bound to evoke strong emotion attributions to the portrayed characters. If the emotions attributed to the characters positively correlate with the emotions experienced by the raters, it is entirely plausible that the signals modeled in the RTPJ reflect third party emotion attributions rather than the first person subjective experience of emotion, which is highly consistent with prior work on the RTPJ representation of others' emotions. The manuscript's introduction does not make a strong case that we should expect the RTPJ to encode first person subjective emotional experience based on prior work and

the study does not adequately demonstrate that first person representation is the correct interpretation of the results.

To claim that these results specifically reflect dimensions of first-person emotional experience, the authors would need to demonstrate that the results are not due to the fMRI subjects' emotion attributions to the characters in the film. The authors could make this claim if behavioral ratings of the film characters' emotions departed substantially from the first person emotion ratings already collected, and if similar neural results cannot be obtained based on the 3rd person emotion ratings. The authors could have also used stimuli that evoked emotions sans engaging narratives of other agents. As the study stands, unfortunately, the first and third person alternatives are completely confounded. Coupled with extensive prior evidence that the RTPJ is specifically recruited to differentiate first- from third-person experience, the authors have not made a strong case that the neural correlates reported reflect the psychological phenomena they claim.

Response: We thank the Reviewer for this constructive criticism, which we agree is relevant for the interpretation of our findings. We conducted additional analyses to support the evidence that right temporo-parietal *emotion dimension* gradients reflect the subjective experience of observers, rather than the attribution of emotional states to movie characters.

In movie watching, actions and dialogues are not generally directed toward the observer. Thus, one's own emotional experience results, on the one hand, from narrative choices aimed at fostering empathic responses and emotional contagion (Smith, 1995) and, on the other hand, from perspective-taking and mentalizing processes (Lombardo et al., 2010; Raz et al., 2013). The fact that character intentions and beliefs shape the subjective experience in a bystander may also explain the high between-subjects agreement in reports of experienced emotions (Philippot, 1993; Gross & Levenson, 1995), present also in our data. However, we believe that the subjective reports we recorded do not merely represent a process of emotion attribution.

To prove that this claim is supported by data, we tested whether the gradient-like organization of right TPJ specifically reflects the subjective experience of raters, rather than portrayed emotions. To this aim, we took advantage of publicly available emotion tagging data of the Forrest Gump movie, provided by an independent group

(Labs et al., 2015). Differently from our behavioral task, raters were asked to indicate the portrayed emotion of each character (e.g., Forrest Gump, Jenny) in 205 movie segments presented in random order and labeled over the course of approximately three weeks. As also indicated by the authors (Labs et al., 2015), this particular procedure minimized carry-over effects and helped observers to exclusively focus on indicators of portrayed emotions. In addition, the possibility to tag emotions independently in each movie segment and to watch each scene more than once, allowed subjects to choose among a larger number of emotion categories ($N = 22$; Ortony et al., 1990), as compared to our set of emotions. Moreover, each observer was instructed to report with a binary label whether the portrayed emotion was directed toward the character itself ("*self-directed*"; e.g., Forrest feeling sad) or toward another character ("*other-directed*"; e.g., Forrest feeling happy for Jenny). These labels were aggregated at group-level and resulted in a regressor of interest that, for each timepoint, offered a measure of the agreement across raters about the "*direction*" of portrayed emotions.

This additional characterization provided us with two *emotion attribution models*: the *self-directed model*, based on the inferring of another person's emotions (i.e., first-order affective Theory of Mind; Shamay-Tsoory & Aharon-Peretz, 2007) and the *other-directed model*, which considers embedded affective states (i.e., second-order affective Theory of Mind; Shamay-Tsoory & Aharon-Peretz, 2007). These two descriptions served as third-person *emotion attribution models* and underwent the exact same processing steps (i.e., 2s lagging and temporal smoothing), which have been applied to our *subjective emotion rating model*.

To address the Reviewer's concerns regarding the validity of our interpretation of the results, (1) we first tested the collinearity between our *subjective emotion rating model* and the two third-person *emotion attribution models*; (2) we then measured which of these significantly explained activity of right TPJ and, most importantly, (3) which of these models are topographically represented in this region.

As the two third-person *emotion attribution models* included the four basic emotions found to be consistent across observers in our experiment (i.e., *happiness*, *fear*, *sadness* and *anger*), we were able to directly assess the correlation for these ratings. We found that scores of our *subjective emotion rating model* positively correlated with those derived from the *other-directed emotion attribution model*. Specifically, Spearman's correlation was $\rho = 0.442$ for *happiness*, $\rho = 0.521$ for *fear*, $\rho = 0.488$ for

sadness and $\rho = 0.442$ for *anger*. The fact that ratings of these two models were positively correlated is not surprising: as already mentioned, the subjective experience of our raters likely depends on portrayed emotions as well. However, the average shared variance between our *subjective emotion rating model* and the *other-directed emotion attribution model* was 35.3% (*happiness*: 24.2%; *fear*: 56.2%; *sadness*: 39.5%; *anger*: 21.2%), indicating that the subjective emotional experience can be inferred from portrayed emotions only in part.

When we assessed the relationship between ratings of the four basic emotions obtained from our experiment and those of the *self-directed emotion attribution model*, we still observed smaller, yet positive, correlations. As a matter of fact, Spearman's correlation was $\rho = 0.284$ for *happiness*, $\rho = 0.309$ for *fear*, $\rho = 0.365$ for *sadness* and $\rho = 0.137$ for *anger*. Here, the shared variance between the two models is even lower: average is 11.4%, *happiness* 5.0%, *fear* 16.0%, *sadness* 21.1% and *anger* 3.3%. Even considering the *other-directed emotion attribution model* (i.e., the one showing the higher correlation with our model), there is a ~65% of variance in ratings which is not shared between tagging of third-person emotion attribution and reports of subjectively experienced affective states. These results made it possible to test the goodness of fit of these alternative descriptions with brain activity and, ultimately, to assess whether *portrayed emotions* are topographically encoded in right TPJ.

Using studyforrest.org fMRI data and the "*direction*" regressor of interest, Hanke and colleagues (2016) already demonstrated that right TPJ activity represents whether emotions are *self-* or *other-directed*. Of note, using a different pipeline (i.e., voxel-wise encoding of *direction* on group-average BOLD signal), we obtained similar results: the higher the BOLD of right TPJ, the more raters labeled emotions as *other-directed* (right TPJ peak R^2 : 0.04; right TPJ average R^2 : 0.02). Significant associations ($p < 0.01$ FDR corrected) between *emotion direction* and BOLD signal were also found in other brain regions of the ToM, empathy and emotion processing networks (for a complete description please refer to Supplementary Figure 16).

Further, we measured the extent to which the two *third-person emotion attribution models* explained brain activity in right TPJ. As these two descriptions are higher in dimensionality as compared to our model, we decided to assess the significance of fitting using three different procedures: (A) matching the dimensionality across models by selecting the first six principal components only; (B) matching the emotion categories in ratings, by performing PCA on the four basic emotions shared across

models (i.e., *happiness, fear, sadness and anger*); (C) using the full model regardless of the dimensionality (i.e., six components for our *subjective emotion rating model* and 22 for each of the two *emotion attribution models*). In addition, to allow a direct and unbiased comparison between R^2 values across different models in right TPJ, we performed cross-validation using a half-run split method (for further details please refer to the caption of Supplementary Figure 8).

Therefore, (A) following the procedure adopted for our subjective ratings, for each of the *emotion attribution models* we performed PCA and selected the first six components, matching the dimensionality of our description. These two six-column partitions, of the original 22-dimensional spaces, represented the vast majority of the explained variance both in the *other-directed* (87.0%) and in the *self-directed* (88.6%) *emotion attribution model*. The two six-column partitions were then fitted into brain activity of right TPJ (15mm radius sphere centered at the Neurosynth reverse inference peak) and the distributions of R^2 coefficients were tested for significance against R^2 null distributions generated from surrogate models (please refer to *Encoding Analysis* in the *Methods* section). Results showed that our *subjective emotion rating model* and the six-column partition of the *other-directed emotion attribution model* significantly explained activity of right TPJ ($p < 0.05$; Supplementary Figure 8). Conversely, the R^2 fitting of the *self-directed* model was not significantly different from the null distribution ($p = 0.269$).

The same holds using (B) ratings of the four basic emotions shared across models. Indeed, our *subjective model* and the *other-directed model* significantly explained right TPJ activity ($p < 0.05$), whereas the *self-directed model* did not ($p = 0.335$). Comparable results were also obtained (C) when testing the full models (i.e., 22 columns; *other-directed emotion attribution model* $p < 0.05$; *self-directed emotion attribution model* $p = 0.078$). Of note, our *subjective emotion rating model* and the *other-directed emotion attribution model* did not significantly differ in explaining activity of right TPJ ($p > 0.05$ for the three above-mentioned procedures).

Most importantly, to prove that the topographic arrangement of *emotion dimensions* is specific for subjective descriptions of the emotional experience, we tested the existence of right TPJ gradients using portrayed emotions. As the *other-directed* was the only *emotion attribution model* significantly explaining activity of our region of interest, we focused our analysis on this description. Specifically, we evaluated two different scenarios: (A) the existence of right TPJ gradients encoding the 22

components of the *other-directed emotion attribution model*; (B) the possibility to identify emotion gradients following the multidimensional alignment (i.e., canonical correlation analysis - CCA; Bilenko & Gallant, 2016) of the 22-dimensional emotion attribution space to the 6-dimensional space defined by subjective ratings. These alternative procedures relate to two different questions: (A) whether the process of emotion attribution is associated to emotion gradients in right TPJ and (B) whether starting from a third-person complex description of portrayed emotions, one can fully reconstruct the subjective report of our raters, as well as its brain topography. Crucially, results proved that (A) none of the first six components obtained from the *other-directed emotion attribution model* (i.e., 87% of the explained variance) retained a topographical organization in right TPJ. Only the 18th PC, explaining the 0.3% of the variance, appeared to be encoded in a gradient-like manner ($\rho = 0.290$, $p = 0.004$). In addition, (B) using CCA we transformed the 22-dimensional space defined by the *other-directed model* to match our subjective reports and found that the correlation values between the aligned components and our six original PCs were $\rho = 0.615$, $\rho = 0.535$, $\rho = 0.490$, $\rho = 0.504$, $\rho = 0.265$ and $\rho = 0.236$, respectively. Noteworthy, when fitting the aligned components into right TPJ activity, only the first PC (i.e., reconstructed *polarity*) was represented through a gradient ($\rho = 0.221$, $p = 0.036$). Gradients for reconstructed *complexity* and *intensity* did not reach statistical significance ($\rho = 0.150$, $p = 0.384$ and $\rho = 0.207$, $p = 0.092$, respectively). Results for these two procedures are detailed in Supplementary Table 5.

Taken together, these pieces of evidence provide two relevant points for the interpretation of our findings. First, other than the *subjective emotion rating model*, also the *other-directed emotion attribution model* explains right temporo-parietal activity. This is in line with previous studies pointing toward the fundamental role of this region in the attribution of mental and affective states to others (Saxe & Kanwisher, 2003; Van Overwalle, 2009; Schurz et al., 2014; Skerry & Saxe, 2015). At the same time, as highlighted by a comprehensive meta-analysis (Kober et al., 2008), right pSTS/TPJ is consistently activated when experiencing and perceiving emotions (see also Burnett & Blakemore, 2009; Nummenmaa et al., 2012). Indeed, activity of this region has been linked to the comprehension (Mano et al., 2009) and understanding of prosody (Hervé et al., 2012, 2013) of emotional narratives, to emotional contagion (Lee et al., 2007; Nummenmaa et al., 2008) and empathy (Morelli et al., 2012; Morelli & Lieberman, 2013), and to the processing of

emotionally-charged facial expressions (Srinivasan et al., 2016; Spunt & Adolphs, 2017).

In our data, the first three *emotion dimensions* obtained from the *subjective emotion rating model* (~85% of the variance) demonstrated a significant topographic organization, whereas components explaining the vast majority (~87% of the variance) of the *other-directed emotion attribution model* were not encoded through gradients. Only a low-variance component of the third-person *other-directed model* appeared to be arranged in a topographic fashion (18th PC in Supplementary Table 5). Of note, the right TPJ pattern associated to this component was also collinear with activity evoked by *polarity* ($\rho = 0.494$) and *intensity* ($\rho = 0.475$) dimensions.

Second, the information coded in the *other-directed emotion attribution model* is not sufficient to fully reconstruct the subjective emotional experience, as demonstrated by the multidimensional alignment procedure. Indeed, by fitting the aligned components, the gradient-based organization of right TPJ was revealed for reconstructed *polarity* and not for reconstructed *complexity* and *intensity*.

In summary, the fact that right TPJ activity, but not its topography, is explained by emotion attribution processes sheds new light on the role of this region in the representation of affective states. We reason that emotion attribution to movie characters modulates right TPJ activity, as this process requires mentalization. At the same time, portrayed emotions influence the affective state of the observer through empathy and emotional contagion, and the final subjective experience is mapped in the same region following the three cardinal axes represented by *emotion dimensions*. This view would reconcile previous studies demonstrating the involvement of this region in the representation of subjective emotional experience (Burnett & Blakemore, 2009; Nummenmaa et al., 2012), empathic processes (Morelli et al., 2012; Morelli & Lieberman, 2013) and the attribution of beliefs and emotions to others (Saxe & Kanwisher, 2003; Van Overwalle, 2009; Schurz et al., 2014).

We have now added all these relevant aspects in the revised version of the *Methods*, *Results* and *Discussion* sections.

[2] Second, the authors smooth the BOLD time series with a 6mm FWHM Gaussian kernel. Is it possible that the gradients are artifacts of smoothing activation peaks? The gradients of visual cortex that the authors liken their results to are mapped using voxels that exhibit a maximal response to a graded feature of the stimulus (e.g. voxels

that respond maximally to stimulus features at a particular eccentricity or polar angle). The gradients used in this study estimate “the partial derivatives in each spatial dimension (x, y, z) and voxel, and can be interpreted as a vector field pointing in the physical direction of increasing β values.” It seems that smoothing a peak activation would necessarily produce a gradient towards the mode, and that if the peak is near the boundary of the meta-analytic ROI it would appear to be a gradient across the region. This seems consistent with the activation maps (except the ‘intensity’ component has multiple peaks near the RTPJ). I presume that the authors considered this and I am simply not understanding how they made the gradient analysis robust to it. Thus, the manuscript would benefit from further elaboration on how the gradient estimation used here compares with the estimation of cortical gradients in sensory cortices and how this study’s analysis protects against the case outlined above.

Response: We agree with the Reviewer that smoothing activation maps would essentially produce gradients. However, as in our study we employed a voxelwise encoding analysis and continuous stimulation, β values represent the association between stimulus features and brain activity. Relevant papers employed a similar approach to describe the existence of topographies outside primary sensory regions (Haxby et al., 2011; Huth et al., 2012, 2016; Connolly et al., 2016; Long et al., 2018). In addition, as reported in Figure 4 of the original submission (now Figure 5 in the revised version of the manuscript), the peak location of BOLD signal for *polarity* and *complexity* dimensions depends on the reported affective state. Indeed, during positive events, brain activity is higher in inferior territories of right TPJ, whereas during negative events, greater BOLD activity is observed in superior territories. Therefore, also *polarity* and *complexity* maps demonstrate multiple peaks within this region. Despite this, we agree with the Reviewer that is not clear how we made our analyses robust to smoothing artifacts in the original submission and we are now providing more details in the revised version of *Methods* section and in the *Supplementary Materials*.

First, to estimate the significance of right TPJ gradients, we used surrogate models built on the *emotion dimensions* (i.e., predictors), leaving untouched the spatial and temporal structure of the fMRI data. This procedure ensures that the spatial smoothness of this region is identical when the association between anatomical and

functional distance is tested either using the actual predictors or the null models. Thus, the strength of a gradient (i.e., effect size) depends on the smoothness of fMRI data, as it is computed from β values, whereas the significance of its estimation is not affected by this filtering procedure.

Second, considering a defined patch of cortex, very high levels of smoothing will cancel the fine-grained spatial organization of anatomo-functional gradients, similarly to what has been shown in multi-voxel pattern analysis (Gardumi et al., 2016). For this reason, we limited the smoothness of fMRI data to 6mm FWHM. In this regard, we did not simply apply a 6mm smoothing filter to the original data, rather we adopted the newer AFNI's 3dBlurToFWHM tool, which estimates and iteratively increases the smoothness of data until a specific FWHM level is reached. We have added this relevant detail in the *fMRI data pre-processing* section of the *Supplementary Materials*.

Third, to confirm that the results we obtained are not due to smoothing artifacts, we estimated the significance of gradients using the original unsmoothed fMRI data. Crucially, also using unfiltered data we have been able to reveal the topographical arrangement of *polarity* ($\rho = 0.167$, $p = 0.033$), *complexity* ($\rho = 0.186$, $p = 0.010$) and *intensity* ($\rho = 0.184$, $p = 0.010$) dimensions in right TPJ. Also, to ensure the qualitative comparison between smoothed and unfiltered data, we mapped the principal direction of *emotion dimension* gradients using spatially unsmoothed fMRI timeseries (see Supplementary Figure 7).

Overall, we believe that these three pieces of evidence strongly support that right TPJ *emotion dimension* gradients do not depend on the application of spatial filtering to fMRI data. We discussed this point in the *Results* section of revised manuscript.

In addition, we would like to thank the Reviewer for the comment about the parallel between emotion and sensory gradients. This comment motivated us to further explore the tuning of voxel response with respect to the three *emotion dimensions*. Indeed, as in sensory cortices topographies result from the maximal response of neurons to a graded stimulus feature, we investigated whether distinct populations of voxels are selective for affective states having specific *polarity*, *complexity* and *intensity* scores. To this aim, we employed the population receptive field method (pRF; Dumoulin & Wandell, 2008) and estimated the tuning curve of right TPJ voxels for each *emotion dimension*. First, we modeled $\sim 5k$ Gaussian distributions using a wide range of plausible values of μ (5th-95th percentile of the scores of each emotion

dimension - 0.5 step) and σ (ranging from 1 to 12 - 0.25 step), sampled on a regular grid. Each *emotion dimension* timeseries was then filtered using these ~5k Gaussian distributions and fitted in brain activity using a linear regression approach. This produced, for each voxel, t-values (i.e., $\beta/SE \beta$) expressing the goodness of fit of each μ and σ combination. Maps of the principal tuning of each voxel were then obtained by selecting the combination characterized by the highest t-value across the ~5k samples (Figure 6).

To estimate the similarity between tunings (i.e., μ parameters) obtained from the pRF approach and our original results (i.e., β coefficients of the gradient estimation), we computed Spearman's ρ across right TPJ voxels. The significance of such an association was tested against a null distribution of β coefficients obtained through the IAAFT procedure ($N = 1,000$).

Results revealed that the tuning obtained from the pRF method significantly approximated the right TPJ topography for each of the three *emotion dimensions* (*polarity*: $\rho = 0.547$, $p = 0.001$; *complexity*: $\rho = 0.560$, $p < 0.001$ and *intensity*: $\rho = 0.596$, $p < 0.001$).

Lastly, we further characterized the prototypical responses of populations of voxels as function of *emotion dimension* scores. To do so, we used the non-negative matrix factorization (NNMF; Lee et al., 1999) and decomposed the multivariate pRF data (i.e., voxels t-values for each μ and σ) into an approximated matrix of lower rank (i.e., 10, retaining at least 90% of the total variance). This method allows parts-based representations (Lee et al., 1999), as the tuning voxels is computed as a linear summation of non-negative basis responses (Supplementary Figure 9).

Results demonstrated the existence of four populations of voxels tuned to specific *polarity* values, which encoded highly and mildly positive and negative events, respectively (Figure 6A). Also, two distinct populations of voxels were tuned to maximally respond during cognitively mediated affective states (i.e., highly and mildly positive *complexity* values), and two other populations were selective for emotions characterized by higher and lower levels of automatic responses (i.e., highly and mildly negative *complexity* values; Figure 6B). Lastly, for the *intensity* dimension two specific populations of voxels were engaged depending on the strength of the emotional experience (Figure 6C). Therefore, as in the retinotopic mapping each hemifield can be further decomposed into quadrants or wedges mapping few visual

degrees, within the *emotional hemifield* of *polarity* and *complexity*, populations of voxels are tuned to lower and higher *intensity* values of the emotional experience. These additional findings better characterize how the topography of *emotion dimensions* relates to gradients represented in sensory areas and provide information in favor of the parallel between emotion and sensory gradients. We have added this aspect in the *Results* and *Discussion* sections of the revised manuscript.

[3] *More generally, the voxel size of fMRI means that the underlying signal is necessarily smoothed, and the dimensions detected are very likely smoother and lower dimensional than the underlying neural code.*

Response: Even though the spatial resolution of fMRI data is limited, this in-vivo noninvasive technique provides us with the opportunity to visualize brain activity of cortical and subcortical regions with millimeter resolution. This precision and the possibility to measure the simultaneous activity of several voxels represent two crucial aspects for the study of the topographical organization of stimulus features in the brain.

Nonetheless, we agree with the Reviewer that the spatial resolution and the biophysical principles of BOLD activity (i.e., the hemodynamic nature of this signal) complicate the conclusions that can be drawn about the underlying *neural code*.

Please note that as we recognize the relevance of such conundrum, we deliberately avoided the use of "*neural*" in the first submission of our manuscript.

On these grounds, we agree that right TPJ *emotion dimension* gradients are lower-dimensional representations of the underlying neural activity, but we would also like to point out that such descriptions could actually be considered as a *neural code*.

Indeed, lower-dimensional representations of neuronal responses are not necessarily incorrect. We can once again use the example of retinotopy and primary visual cortex to better clarify this aspect. Which is the *underlying neural code* in striate cortex?

Animal studies demonstrate that within this region a fine-grain code determines whether visual information is collected from the left or right eye (i.e., columns of ocular dominance) and also that neuronal populations are tuned to specific stimulus orientation. Nevertheless, on top of this local organization, other lower-dimensional *neural codes* do exist, such as the retinotopic mapping of azimuth and elevation.

These smoother lower-dimensional gradients depend on the underlying neuronal

activity, even though they overlap with the columnar organization of ocular dominance and orientation tuning. The ability to capture either local or global representations of neural activity relates to the spatial resolution of imaging techniques. Indeed, while standard resolution fMRI has been successfully adopted to map stimulus location as function of eccentricity and polar angle (Serenó et al., 1995), specific ultra-high resolution sequences are needed to map ocular dominance (Cheng et al., 2001; Yacoub et al., 2007). Further, whether fMRI resolution is sufficient to map the selectivity to stimulus orientation is still unclear (Freeman et al., 2011; Roth et al., 2018).

Therefore, one can conclude that multiple *neural codes* exist at different scales and that the spatial precision of the technique determines which of these are revealed. In the present study, our analyses proved the existence of *emotion dimension* gradients resembling the spatial characteristics and resolution of retinotopic maps in primary visual cortex. There is abundance of evidence suggesting that gradients represent an efficient brain mechanism of information coding, which minimizes the metabolic cost (Harvey et al., 2013; Huth et al., 2016; Margulies et al., 2016; Huntenburg et al., 2018). Hence, our findings support the existence of a lower dimensional, yet biologically favorable, *neural code* of emotion processing in temporo-parietal regions.

Ultra high-resolution fMRI acquisitions may be adopted to prove the existence of other local *neural codes* within right TPJ. We believe that such a possibility is not in contrast with the *emotion dimension* gradients we found, especially considering the coexistence of multi-scale *neural codes* and the multifaceted nature of this region. We have added these aspects to the revised version of the Discussion section.

[4] The authors report a peak variance explained in the RTPJ of $\rho^2 = 0.07$ and compare that to other research on emotion representation in the RTPJ (Skerry & Saxe 2015), which reported correlations of similar strength. However, Kendall's τ is typically smaller than the Spearman's ρ coefficient for the same data so the proportion of variance explained in this study might be still be substantially less than the comparison study (which also used 1.5x as many subjects). More generally, this number suggests that there is substantial variance within this region that is not yet explained by the authors' theory.

Response: Please note that the effect size we reported is $R^2 = 0.07$, being the output of a multiple linear regression analysis (i.e., voxelwise encoding). To compare our effect size with the one reported by Skerry & Saxe, 2015 using the same metric (i.e., Kendall's $\tau \sim 0.08$), we correlated the predicted fMRI signal obtained from the encoding procedure, with the actual BOLD activity within the same voxel (i.e., at R^2 peak). The association between the two time-series was Spearman's $\rho = 0.23$ and Kendall's $\tau = 0.15$.

In addition, we acknowledge that while perceptual and motor representations (Khaligh-Razavi & Kriegeskorte, 2014; Ejaz et al., 2015) explain most of the variance in brain activity, the fitting of emotion models in associative cortical areas does not yield similar results. However, as also suggested by Reviewer #3, to better describe how much variance is actually explained by our *emotion dimension model*, we computed noise ceiling at right TPJ peak. We found that lower and upper noise ceiling bounds were $R^2 = 0.13$ (Spearman's $\rho = 0.33$ and Kendall's $\tau = 0.22$) and $R^2 = 0.23$ (Spearman's $\rho = 0.45$ and Kendall's $\tau = 0.31$), respectively. These results suggest that, within this region, our *emotion dimension model* explains between 30% (i.e., upper bound) and 54% (i.e., lower bound) of the variance.

Lastly, concerning the sample size, we are aware that Skerry & Saxe, 2015 study included a larger number of participants as compared to our research. However, considering the total amount of collected data (i.e., degrees of freedom) we are confident that the two experiments are comparable, as our final sample was composed by 3,595 timepoints acquired in 14 individuals and Skerry & Saxe, 2015 acquired 2,285 timepoints in 22 subjects. These aspects are now detailed in *Supplementary Materials* and reported in the *Discussion* section.

[5] The authors state that, "...the orthogonal arrangement of polarity and complexity in right TPJ and the fact that intensity was represented both superiorly and inferiorly to the superior temporal sulcus determined that all the possible combinations of emotional states elicited by the 'Forrest Gump' movie could be mapped within this region." The authors should be much more careful about asserting that the emotions they measured represent the full space of emotions, especially considering that the 3 components that the authors use do not even capture the disgust and surprise ratings in their data and other work they cite (Cowen & Keltner 2017) report high dimensional emotion representations.

Response: We thank the Reviewer for raising this point and we regret for not having better detailed this aspect in the original submission of our manuscript. First, we do agree with the Reviewer that it would not be possible to obtain a comprehensive representation of the space of human emotions considering states elicited by a 2h movie. In fact, when we stated that gradients could represent "[...] all the possible combinations of emotional states [...]", we specifically referred to those elicited by the Forrest Gump movie. In this regard, we would like to emphasize that even though ratings of *surprise* and *disgust* were not consistent across all the participants, we did not exclude such emotions from PCA. Therefore, the first three components do capture the *disgust* and *surprise* ratings, as can be appreciated from loadings of these two emotions on *polarity* and *intensity* (Figure 1C).

Moreover, we would also like to point out that even though subjects were asked to report their inner experience using six emotion categories, their ratings were not limited to binary choices. Indeed, at each timepoint raters could simultaneously specify the perceived intensity of more than one emotion, leading to the definition of more complex affective states as compared to the basic ones. To further highlight this aspect, we performed dimensionality reduction and clustering analyses on emotion timeseries. In brief, starting from emotion ratings averaged across participants, we selected timepoints characterized by the highest intensity (i.e., above the 50th percentile) and applied t-distributed stochastic neighbor embedding as in Cowen & Keltner, 2017 (t-SNE; perplexity = 30; theta = 0.05). The results of this analysis are presented in Figure 2: here, each element represents a specific timepoint in the movie and the distance between elements depends on the statistical similarity of emotion ratings. Element color reflects the scores of the *polarity* and *complexity* dimensions: positive and negative events (i.e., *polarity*) are associated to the red and blue channels, respectively, whereas *complexity* scores modulate the green channel. To reveal the variety of affective states elicited by the movie, we then applied k-means clustering analysis to the projection of timepoints in the t-SNE manifold and determined the number of clusters using the silhouette criterion (Rousseeuw & Kaufman, 1990). Such procedure highlighted 15 distinct clusters, each defined by timepoints expressing either a single or a mixture of basic emotions (see pie-charts in Figure 2). For instance, cluster *m* represents timepoints mainly characterized by *happiness*, whereas cluster *k* identifies movie scenes in which the reported intensity of

fear and *sadness* is similar. Other clusters, such as *e* and *j*, depict timepoints connoted by high ambivalence (thus, high *complexity*), as *happiness* and *sadness* are similarly represented.

Therefore, in our experiment the final number of elicited emotional states is greater than the six emotion categories among which subjects could choose. Interestingly, the variety of affective states elicited by the *Forrest Gump* movie is closer to the number of emotion categories identified by previous studies (e.g., 22 in Skerry and Saxe, 2015; 27 in Cowen and Keltner, 2017), whose aim was to capture a wide range of emotional states (e.g., *sexual desire*, *craving*, *terror*). Of note, this evidence is supported also by single-subject reports, in which 38% (SE: $\pm 2.3\%$) of timepoints were associated to a single emotion, 29% (SE: $\pm 3.5\%$) were connoted by two emotions and 6% (SE: $\pm 1.4\%$) by the concurrent experience of three distinct emotions.

All these aspects are now detailed in the revised version of the *Methods*, *Results* and *Discussion* sections and the sentences indicated by the Reviewer have been rephrased accordingly.

Reviewer #3 (Remarks to the Author):

Lettieri, Handjaras, et al. present an investigation of the topographic organization of emotion representation across the cortex. They draw upon a rich open fMRI data set to examine how emotional experiences are encoded in activity across the temporoparietal junction (TPJ). They find that three overlapping yet orthogonal gradients encode the polarity, intensity, and complexity of participants emotions. The research topic is of theoretical interest to a wide range of psychological and neural scientists. The methods are sophisticated, and the paper is well and clearly written. Thus I believe this work could make a substantial contribution to the literature. However, below I raise a number of points which I believe the manuscript would benefit from addressing:

Response: We thank the Reviewer for the appreciation of our work.

1) The authors derived three emotions dimensions by applying a PCA to ratings of six basic emotions across the movie Forrest Gump. Although this is a straightforward way to address this problem, it raises several concerns:

a. The PCs produced by this procedure inevitably depend to some extent on the particular emotions the authors choose to have rated. Although there is a theoretical justification for the six states in question – as “basic emotions” – these states omit a wide range of important emotional states, such as social/secondary emotions like pride and envy. It would be helpful to know whether the same dimensions emerge when a broader set of states are rated.

Response: To address the Reviewer’s concern, we employed Labs and colleagues (2015) ratings describing portrayed emotions of Forrest Gump considering embedded affective states (i.e., *other-directed* emotions; for details please see response to point 3 below, and to point 1 of Reviewer #2). In their behavioral experiment, the authors provided subjects with a wide range of emotion categories among which they could draw to describe emotions of movie characters. In addition to four basic emotions (i.e., *happiness, fear, sadness* and *anger*), these 22 emotion categories included secondary and social states (Ortony et al., 1990) as *admiration, contempt, gratitude,*

hate, love, and pride among others (for a complete description please refer to Labs et al., 2015).

To test whether the *polarity, complexity* and *intensity* dimensions could be retrieved using a larger number of emotion categories, we applied PCA to Labs data after lagging and temporally smoothing the 22 emotion timeseries, as we did for our ratings. The first six dimensions - 85% of explained variance - were selected to match the dimensionality of our emotion rating model and were transformed by rotating PC scores using the procrustes criterion. Results of this procedure are presented in Supplementary Figure 11 of the revised manuscript, in which factor loadings of *polarity, complexity* and *intensity* dimensions (panel A) can be compared with those obtained from Labs and colleagues data in the original unrotated (panel B) and rotated (panel C) version of PCs. The first three rotated components represented respectively the 20.6%, 19.8% and 16.6% of the explained variance, and were positively associated with the three *emotion dimensions* obtained from our data. Correlation for rotated PC₁ versus *polarity* was Spearman's $\rho = 0.589$, for rotated PC₂ versus *complexity* was Spearman's $\rho = 0.533$ and for rotated PC₃ versus *intensity* was Spearman's $\rho = 0.488$.

In addition, it is important to note that other than basic emotions (i.e., *happiness, fear, sadness* and *anger*), only four secondary/social affective states - i.e., *love, contempt, admiration* and *gloating* - substantially contributed to the first six components derived from Labs and colleagues data, even considering the unrotated version (panel B). This is in line with Labs et al., (2015), as the authors highlighted that the majority of emotional episodes involved the five categories of *anger, fear, happiness, love* and *sadness*, whereas other secondary/social categories available to subjects (e.g., *resentment, gratification, satisfaction*), were used infrequently or employed only by a subset of observers.

Lastly, in our experiment the final number of emotional states elicited by the Forrest Gump movie was greater ($N = 15$) than the six emotion categories among which subjects could choose (for details please see response to point 5 of Reviewer #2). A number of these affective states represented a complex mixture of basic emotions (see Figure 2 in the revised version of the manuscript) and likely expressed secondary affective states, as in the case of *ambivalence* (i.e., cluster *j* of Figure 2, where *happiness* and *sadness* were equally reported) or *resentment* (i.e., cluster *i* of Figure 2, where *sadness, anger* and *disgust* were present at the same time). Nonetheless, as the

description of clusters would be speculative, we deliberately chose to not label them using a single term.

In summary, the same *polarity*, *complexity* and *intensity* dimensions emerged even when a broader set of emotion categories was used. However, we also cannot exclude that other dimensions encoding emotions not primarily elicited by *Forrest Gump* (e.g., sexual desire, envy, terror), may be topographically represented in the brain. We are now discussing this evidence in the *Discussion* section of the revised version of the manuscript and in *Supplementary Text*.

b. Related to (a), the PCs extracted from the movie ratings also depend on the qualities of the stimulus itself. Forrest Gump is well known for being both an emotionally evocative movie, and one with highly varied content, which makes it a prudent choice in the present context. However, I doubt that it or indeed any individual movie could come close to covering the full range of human emotion. Moreover, the temporal structure of emotion may differ considerably between movies in general, as opposed to real life. Presumably this is part of why we are often willing to pay money to watch a movie, but would probably not pay so much to watch a random slice of someone's actual life. Perhaps the authors could compare temporal dynamics observed in their rating data to available experience-sampling data sets to assess how well their stimulus reflects real life experience?

Response: We agree with the Reviewer that *Forrest Gump* is an emotionally evocative movie that elicits a variety of affective states in a relatively short amount of time. Although movies have been successfully used to study emotions in the laboratory setting (Gross & Levenson, 1995; Philippot, 1993; Schaefer et al., 2010), we also recognize that they cannot be representative of all possible affective states, as those experienced in real-life. In this regard, we consider the criticism raised by the Reviewer an interesting point that provides us the opportunity to test the validity of our stimulus. Thus, we assessed whether the temporal dynamics of portrayed emotions in *Forrest Gump* reflect those reported in real-life experiences. Of note, only a few studies assessed the temporal dynamics of emotion transitions in real life and one of these, Thornton & Tamir, 2017 (hereinafter T&T) released the collected data. Thus, we took advantage of their experience-sampling dataset (i.e., Study 3) comprising ~65,000 ratings obtained from ~10,000 participants, who were

asked to report their own emotional state throughout the day, choosing among 18 categories (i.e., *alertness, amusement, awe, gratitude, hope, joy, love, pride, satisfaction, anger, anxiety, contempt, disgust, embarrassment, fear, guilt, offense and sadness*). In T&T's study, the authors used the collected reports to build an experience-based description of emotion transitions (i.e., *real life emotion transitions*). Specifically, by considering each reported emotion and the one following in time, they tested whether the co-occurrence of emotions is predicted by a mental representation of emotion transitions (for further details please refer to the original publication). We particularly selected the model based on T&T's study 3, as nine out of 18 emotion categories included in this dataset (i.e., *anger, sadness, fear, contempt, satisfaction, gratitude, hope, love and pride*) were also adopted by Labs and colleagues (Labs et al., 2015) to label portrayed emotions in *Forrest Gump*. As reported above (i.e., response to point 1 of Reviewer #2), for the emotion tagging data, Labs and colleagues asked a group of participants to indicate the portrayed emotions of each character (e.g., *Forrest Gump, Jenny*) in 205 randomly presented movie segments. The possibility to tag emotions independently in each movie segment as well as to watch each scene more than once, allowed subjects to choose among a larger number of emotion categories ($N = 22$; Ortony et al., 1990), as compared to our set of emotions.

Starting from these data, we thoroughly followed T&T methods and converted ratings into discrete outcomes (i.e., emotion present or not) for each timepoint. We then built a transition count matrix by measuring the number of transitions between all possible emotion pairings in adjacent timepoints (i.e., between t and $t+1$). This matrix was further normalized by frequency-based expectations (as in T&T's supplementary materials), obtaining the odds of each transition. The log-transformed version of this matrix (i.e., *movie emotion transitions*) was then compared to real-life data (i.e., *real life emotion transitions*, as reported in T&T's original paper) using Spearman's ρ . To assess the statistical significance of this association, we generated surrogate timeseries for the nine emotion categories through the IAAFT procedure ($N = 1,000$; see *Methods* for details). For each of the 1,000 null models, a transition count matrix was then obtained, normalized and log-transformed, similarly to what has been done for emotion transitions present in the movie. The obtained matrices were correlated with

T&T's real life data, generating a null distribution against which the actual association between *movie-based* and *real life emotion transition models* was tested.

Results showed that emotion transitions obtained from movie and real life data were significantly associated (Spearman's $\rho = 0.646$; $p = 0.001$; Supplementary Figure 12). In addition, as this analysis explored the consistency between *movie-based* and *real life emotion transitions* in a short time window (2s), we also evaluated whether this relationship exists at different time scales. Therefore, we built a number of *movie-based* models, each measuring the likelihood of emotion transitions between timepoint t and timepoint $t+n$ in the future, with a maximum delay of 120 seconds (60 timepoints). These models were then correlated with real life data and statistical significance was assessed using the procedure described above. Results are reported in panel D of Supplementary Figure 12 and show that the *real life model* predicts emotion transitions in the movie up to 58 seconds.

Of note, *happiness* is one of the emotion categories most present in Forrest Gump tagging data, yet it has not been used in reports collected for study 3 of T&T's paper. Hence, we decided to include this emotion in the *movie-based model*, using *joy*, *awe* or *amusement* as its counterpart in the *real life model*. This allowed us to estimate the robustness of the association between movie and real life data considering different facets of the basic emotion *happiness*.

Interestingly, using *joy*, *awe* or *amusement* as proxies of *happiness*, the association between the *movie* and the *real life emotion transitions* is significant (*joy*: Spearman's $\rho = 0.702$; $p = 0.001$; *awe*: Spearman's $\rho = 0.702$; $p = 0.001$; *amusement*: Spearman's $\rho = 0.686$; $p = 0.001$) and real life data predict emotion transitions in the movie up to 64 seconds in the future.

Altogether, these analyses show that within a ~60 seconds time window our stimulus reflects emotion transitions similar to those experienced in real life and predicted by a mental model of emotion co-occurrence. We believe that these new findings provide a relevant contribution to our manuscript by substantiating the ecological validity of our stimulus.

c. By conducting the PCA on ratings of the movie itself, these dimensions are in some sense overfitted to this particular stimulus. For example, if factor structure/loadings were derived from ratings of separate movies (or even non-movie stimuli), and then

applied to the present data, I imagine that they would explain less of the variance in both the basic emotion ratings and the fMRI data. The authors should note this caveat, perhaps in relation to their more general discussion of how much variance their model captures.

Response: We agree with the Reviewer that the collected emotion ratings are specific for the present movie. Thus, reconstructed factor loadings presumably explain less variance when applied to brain activity evoked by other stimuli. Also, because of specific narrative choices, it is likely that different affective states are more represented in other movies (e.g., *terror*, *envy*) and even training on two hours of fMRI acquisition, may not be sufficient to compensate the different distribution of emotion categories between the training and test set.

However, we recognize that the robust estimation of the effect size of our *emotion dimension* model is an important point (also raised by Reviewer #2). In addition, the use of cross-validation provided us with the opportunity to measure how other, more complex, emotion models explain brain activity in an unbiased manner, as the estimation of cross-validated R^2 coefficients is not affected by model dimensionality. Therefore, to assess the fitting of our model we applied a half-split procedure to each fMRI run and randomly selected one of the two halves as the training data for the estimation of β coefficients. We then measured the goodness of fit of our model by multiplying the predictors of the remaining half with estimated β coefficients, thus reconstructing the predicted fMRI signal. The latter was then correlated with the actual fMRI activity, obtaining the final cross-validated R^2 coefficient.

To avoid possible confounds introduced by selecting the first or the second part of each run as training/test dataset, we repeated the same procedure 200 times (i.e., bootstrapping), each one randomly assigning the first or second half to the training/test set. Results for the cross-validated R^2 are reported in Supplementary Figure 8.

In addition, following the Reviewer's suggestion we better discussed the ability of our model to explain brain activity, also considering the results obtained from the cross-validation and the noise ceiling procedures (for details please refer to point 5 below). These aspects are detailed in the revised version of the *Discussion* section and in the Supplementary Text.

d. In PCA, once the number of components has been specified, any rotation of the retained components will explain the same total variance. How can we know that the rotation the authors consider is the “canonical” rotation of these dimensions? A recent preprint (<https://psyarxiv.com/6dvn3/>) makes this point at length in a fairly similar context: topographic maps of facial expressions of emotion across the FFA. The authors might try testing rotations of their components to see whether they produce better or worse gradients across the TPJ. Indeed, the search for neural gradients might suggest an interesting way to establish which rotations are canonical, which would be a valuable methodological contribution in itself.

Response: We thank the Reviewer for raising this interesting point. Indeed, we agree that testing the correspondence between anatomo-functional gradients and PC rotations could reveal which stimulus features are actually encoded onto the cortical mantle (Huth et al., 2016).

We carefully read the suggested preprint and conducted additional analyses to address this relevant point. In the original submission, we limited our analyses to the unrotated version of principal components as the assessment of all possible rotations increases exponentially with the number of components, becoming computationally intractable even with few dimensions (e.g., four dimensions and $\pm 45^\circ$ rotations produce ~ 68 million solutions). However, to test whether the unrotated version of PCs represents the canonical description of emotion dimensions, we developed a novel approach that isolates the best-approximated solution.

First, we restricted our analysis to the three emotion dimensions consistent across subjects (i.e., *polarity*, *complexity* and *intensity*), which also showed a gradient-like organization in right TPJ. Second, we performed only orthogonal rotations because of two reasons: (1) as indicated by the Reviewer, any orthogonal rotation of the original components will explain the same total variance; (2) the computation of gradient directions requires the accurate estimate of β coefficients obtained from a multiple linear regression analysis. This approach is however not robust if predictors are collinear, which may be the case when oblique rotations are applied.

Therefore, we first estimated all the possible elemental rotations along the axes defined by the three emotion dimensions (i.e., x: *polarity*, y: *complexity* and z: *intensity*). We explored rotations between $\pm 45^\circ$ with 1° step, as this range ensured

univocal solutions that would not produce the shifting of PC labels. As a matter of fact, considering a convenient bi-dimensional example, we can assert that 60° orthogonal rotations for PC_1 and PC_2 would produce solutions in which PC_1 approximates the unrotated version of PC_2 and PC_2 resembles the 180° -rotated (i.e., flipped) version of PC_1 . Such a solution, though, would be identical to a 30° rotation, except for the PC sign. In line with this, rotations of $\pm 90^\circ$ would simply shift PC labels (e.g., rotated *complexity* would become now unrotated *intensity*), whereas $\pm 180^\circ$ rotations would result in sign flipping. The latter case would lead to brain activity estimates (i.e., β values) being the topographically mirrored version of those obtained using the unrotated dimensions and, thus, to ρ values of the same magnitude for the association between anatomical and functional distance.

As all the possible rotations between $\pm 45^\circ$ produce $\sim 750k$ solutions - which is already computationally intense -, we uniformly sampled 70k rotations from the original space. Further, the intuitive mapping of gradient magnitude (i.e., Spearman's ρ between anatomical and functional distance) in the manifold defined by the rotated solutions is non trivial and a specific figure has been dedicated to illustrate the method we propose (Supplementary Figure 4A).

In brief, we represented gradient intensity of the unrotated *emotion dimensions* as the central point of a 3D manifold described by all the $\pm 45^\circ$ explored rotations. We also mapped gradient intensity of all the rotated solutions as points in this space, color-coding the magnitude of the association between anatomical and functional distance. Rotations are expressed according to three cardinal trajectories originating from the central point (i.e., the unrotated *emotion dimensions*), each one determining the orthogonal rotation of two components while maintaining fixed the other one. Therefore, points lying on the red trajectory depict solutions in which the original unrotated version of *polarity* is present and *complexity* and *intensity* are actually rotated. The same applies also to the green and blue trajectories in which *complexity* and *intensity* respectively maintain their original unrotated form. All the other mapped solutions describe orthogonal rotations concurrently applied to the three *emotion dimensions*. The larger the geodesic distance in the solution space between axes origin and a specific point, the larger is the applied rotation to the original *emotion dimensions*. Lastly, the position of each solution with respect to the central point also defines the direction of the rotation (i.e., positive or negative).

Results for this procedure are depicted in Supplementary Figure 4B and show that the original unrotated version of the *polarity*, *complexity* and *intensity* dimensions is the optimal solution to explain the gradient-like organization of right TPJ. Indeed, within the space defined by PC rotations, no solutions retained p coefficients (i.e., gradient magnitude) larger than those associated with the unrotated components for all the three *emotion dimensions*.

In addition, rotations in which the gradient magnitude is similar across the three *emotion dimensions* are arranged close to the unrotated solution (i.e., white areas in Supplementary Figure 4B), whereas moving away from axes origin at least one of the three dimensions is not represented as a gradient in right TPJ (i.e., yellow and cyan areas in Supplementary Figure 4B). Of note, considering all the explored solutions, very few rotations produce gradients encoding combined *polarity* and *intensity*, but not *complexity* (i.e., lack of magenta areas in Supplementary Figure 4B).

As the original unrotated solution was the best among $\sim 70k$ explored rotations, we assessed the probability of occurrence of such behavior using a Monte Carlo simulation. Therefore, we created 1,000 PC models by selecting 100 consecutive timepoints from the *emotion dimension* timeseries to predict randomly sampled right TPJ activity ($N = 100$ consecutive timepoints). For each iteration, we then mapped the results of the multiple linear regression analysis (i.e., β coefficients) on a 3-D grid of 25 voxels and computed the correspondence between the anatomical and functional distance obtained using the unrotated and rotated ($\pm 45^\circ$ with 5° step; $\sim 7k$ explored solutions) predictors. Lastly, we counted the number of iterations in which the gradient magnitude of the rotated predictors was higher with respect to the original unrotated solution.

Results of the Monte Carlo simulation confirm the peculiarity of real data. Indeed, while the unrotated version of *emotion dimensions* represents the optimal solution in explaining the right TPJ gradient-like organization, rotated components produce stronger gradients in the vast majority of simulated cases (96.2%; $p < 0.05$). Of note, we tested the reliability of the results obtained from the Monte Carlo simulation by also varying the length of the timeseries (50, 100 and 200 timepoints), the number of voxels ($N = 25, 100$) and by generating synthetic PC models and fMRI signal using Gaussian noise. Results for all these procedures were consistent with the original simulation (data not shown).

We agree with the Reviewer that our approach may be of interest to other researchers and we made available the code for computing, exploring and rendering PC rotations (please see the script *gradient_explorer.m* at <https://osf.io/tzpdf>).

These aspects are briefly discussed in the *Results* section of the revised manuscript and thoroughly detailed in the *Supplementary Materials*.

2) The region the authors consider as the “TPJ” is very large – much larger than this region typically appears in the literature. It includes substantial portions of parietal and occipital cortex well outside of what would usually be called the TPJ (e.g., as defined using a false-belief localizer). I do not think this is necessarily problematic from an analytic point of view, but I do think it may give casual readers the wrong impression of the spatial extent of the observed patterns. I think the authors should acknowledge this discrepancy more explicitly, and make it clear from the beginning (i.e. in the title or the abstract) just how extensive these gradients appear. However, they can emphasize at the same time that this result generalizes across a range of spatial scales (as demonstrated in the results reported in supplementary table 2).

Response: We regret for not having better clarified the size of the region of interest encoding *emotion dimension* gradients. First, the patterns reported in the original version of Figure 2A,C and D (now Figure 3A,C and D) do not represent the optimal solution for the identification of emotion topography. In fact, while panel A shows regions significantly encoding the full *emotion rating model*, panels C and D depict the largest patch of cortex (i.e., maximum radius is 27mm, volume: 44,658mm³) employed to measure the association between cortical distance and brain activity. As correctly pointed out by the Reviewer, the existence of *emotion dimension* gradients generalizes across several definition of the ROI size, yet it is important to note that the optimal solution is represented by a 15mm radius sphere (11,556 mm³ volume). In fact, although *emotion dimension* gradients are significantly represented also considering a 27mm ROI (Supplementary Table 2), the effect size decreases for radii larger than 15mm.

Also, we agree that the paper would benefit from a quantitative comparison of the size of our ROI with the definition of right TPJ based on the neuroimaging literature.

To do so, we considered the right TPJ region obtained from the Neurosynth database (<http://old.neurosynth.org/analyses/terms/tpj/>). This meta-analytic definition is based

on brain activations elicited by classic Theory of Mind and affective processing tasks, such as false-belief (Aichhorn et al., 2009; Döhnel et al., 2012), emotion perception (Garrett & Maddock, 2006) or reappraisal tasks (Silvers et al., 2014). Therefore, this map would represent a reliable estimate of the right TPJ size, against which one could compare the volume of our spherical ROI. We have summarized this aspect in Supplementary Figure 15 of the revised submission.

Considering the Neurosynth *TPJ reverse inference map* - $p(F|A)$ -, the volume of the largest cluster was 8,127 mm³ (coordinates: $x = +58, y = -50, z = +16$), whereas the volume of the spherical ROI that better represents emotion topography in our study (i.e., 15mm radius) was 11,556 mm³. Yet, considering the *TPJ forward inference map* - $p(A|F)$ -, the volume of the largest cluster was 16,929 mm³ (coordinates: $x = +58, y = -50, z = +16$). Altogether, these results indicate that the optimal description of *emotion dimension* gradients is represented in a patch of cortex that approximates the definition of right TPJ based on brain activation studies (i.e., ~42% larger in volume as compared to the *reverse inference* map, but also ~32% smaller than the *forward inference* definition). However, as we agree with the Reviewer that these gradients extend well beyond (i.e., 27mm radius sphere) the size of right TPJ reported in many studies, we are now using the term "*temporo-parietal territories*" in the *Abstract* section and throughout the revised manuscript.

In addition, we are now detailing the similarities and discrepancies between our ROI and the canonical definition of R TPJ in the revised version of the *Discussion* section and Supplementary Text.

3) The TPJ is also a region which is more typically implicated in understanding others' thoughts and feelings (i.e., theory of mind) than in the actual experience of emotion. As the authors point out, one way in which movies elicit emotions in people is through empathy with the characters. However, these facts together suggest a possible confound: the emotions that participants rated may be highly associated with the emotions they perceive the characters to experience. Such a confound would complicate the interpretation of the present results: do the TPJ emotion gradients encode one's own emotional experience, or the perceived emotional experience of others? Either result would be interesting, but it is important to know which account is better supported. One straightforward way to address this would be to ask additional movie viewers to rate the characters' emotions, rather than their own. This

would allow the authors to measure the extent of this potential confound, and potentially statistically control for it.

Response: We thank the Reviewer for this comment, which has been also raised by Reviewer #2. Although we agree that the relevance of *emotionotopy* would not depend on whether *perceived* rather than *portrayed emotions* are mapped in temporoparietal territories, we followed the Reviewer's suggestion and tested which model better explained right TPJ gradients. To this aim, we used tagging data of Forrest Gump *portrayed emotions* provided by Labs and colleagues (2015).

In brief, we found that while our *subjective ratings* positively correlated with emotion attribution data, a significant amount of variance was not shared between models (i.e., ~65%). This allowed us to measure the association between emotion attribution data and brain activity, and to compare the goodness of this fitting with the one obtained from subjective ratings. The results of this analysis showed that only our model and the one representing embedded affective states of movie characters (i.e., *other-directed emotion attribution model*) significantly explain right TPJ activity ($p < 0.05$). As also pointed out by the Reviewer, this is in line with the well-known role of this region in mentalizing and perspective-taking processes (Saxe & Kanwisher, 2003; Van Overwalle, 2009; Schurz et al., 2014).

However, differently from our *subjective emotion rating model*, none of the first six components obtained from the *emotion attribution model* was topographically encoded in right TPJ.

These pieces of evidence clearly indicate that while there is a correspondence between *portrayed* and *perceived* emotions, the subjective experience of our raters cannot be merely reduced to a process of emotion attribution. In addition, while both *first-* and *third-person* descriptions of emotions explain activity of our ROI, the topographic organization of right TPJ exclusively reflects the inner affective experience.

Please note that all the details relative to these additional analyses are also specified in the response to point 1 of Reviewer #2 and throughout the revised version of the manuscript.

4) The authors tested Italian rater's emotional experiences in German speaking participants' brains, while each watched an American movie. The success of the encoding model across these linguistic and cultural boundaries is impressive and might be emphasized further. It might be interesting to discuss how other/larger cross-cultural differences might qualify the conclusions of this investigation. Ample evidence demonstrates that emotional experience and expression differ substantially across cultures – how might such differences potentially be reflected in the organization of cortex?

Response: We thank the Reviewer for having highlighted the cross-cultural relevance of our encoding analysis. As a matter of fact, subjects participating in the behavioral and the fMRI experiment watched the Forrest Gump movie in their own native language (i.e., Italian and German dubbed version, respectively). As pointed out by the Reviewer, the successful encoding of emotion ratings in brain activity of independent subjects indicates that linguistic features, related to the translation of movie dialogues, did not considerably affect the unfolding of the emotional experience. This is not a trivial aspect, as even within the Western macro-culture specific terms identify affective states that are difficult to translate into other languages (e.g., *schadenfreude* in German, *saudades* in Portuguese or *hygge* in Danish). Furthermore, in movies as in the real life, language is not the only way through which one can express emotions, as gestures, facial expressions and bodily postures play a fundamental role as well.

In line with our results, previous behavioral investigations showed the high consistency of emotional responses of European subjects to American, Italian, French and Belgian movies (Philippot, 1993; Schaefer et al., 2010). Likewise, a large sample study highlighted the relevance of culture over ethnicity in shaping the emotional response to movies (Gross & Levenson, 1995). These findings suggest that the emotional experience elicited by specific stimuli is similar across micro-cultures (e.g., Italian, French, German), though pertaining to the same macro-culture (e.g., Western). Indeed, our results further extend this evidence to the cortical representation of emotions, as we have been able to encode the affective experience of subjects across two distinct micro-cultures, the German and the Italian one. Despite this, it would be of major interest to test whether the topographic representation of emotions in

temporo-parietal territories is preserved when subjects come from different macro-cultures (e.g., Eastern vs Western).

Indeed, previous behavioral investigations reported differences between subjects with far-off cultural backgrounds in the display of emotional expressions (Marsh et al., 2003; Immordino-Yang et al., 2016), recognition accuracy (Elfenbein & Ambady, 2002) and emotion regulation (Butler et al., 2007; Matsumoto et al., 2008). Therefore, it is very likely that macro-cultures are actually able to influence social behaviors and affective processing. From an evolutionary perspective, this might strengthen the in-group/out-group distinction and would provide individuals with a specific code to behave appropriately in social contexts.

In addition, over the past years neuroimaging studies have investigated brain differences in the processing of social and emotional stimuli between macro-cultures. A meta-analysis by Han and Ma (2014) summarizes such findings by showing that regions involved in Theory of Mind, empathy and emotion recognition are differentially recruited in Westerners and Asians. Similarly, when in-/out-group stimuli (e.g., facial expressions) are employed, macro-cultural biases influence brain activations associated with emotion processing (Markham & Wang, 1996; Elfenbein & Ambady, 2002; Chiao et al., 2008; Adams et al., 2010).

Overall, these studies point toward the fact that macro-cultural differences influence the brain processing of emotions and affective states. Our hypothesis in this regard is that *topography* is a biologically advantageous *neural code* for the representation of affective states onto the cortical mantle. Such a code, defined by the three axes of our gradients, would exist in the human brain regardless of macro-cultural differences.

What would depend on the cultural background of each individual is instead the mapping of distinct emotional states within these gradients. For instance, ruminative thinking, sadness and apathy characterize *melancholy* in the Western culture and we speculate that such an emotion would be mapped in the brain as a negative state having high *complexity*. However, if different levels of *polarity*, *complexity* and *intensity* characterize *melancholy* in other macro-cultures, this emotion would be mapped differently with respect to the three right TPJ *emotion dimension* gradients. This would also imply that the *cross-encoding* we demonstrated for the Italian and German micro-cultures may not be generalized to distinct macro-cultures. The testing of these hypotheses is beyond the scope of this work. Yet, further studies addressing

such questions would add a relevant contribution to the existing literature on differences and similarities in the neural representation of emotions across cultures. As requested by the Reviewer, we added these considerations in the revised version of the *Discussion* section.

5) The authors raise the low R2 of their model as a potential limitation. Given that they have data from multiple participants watching/rating the same movie, it seems as if they have the necessary data to compute the reliability of both emotion dimensions and neural activity. These reliabilities could then be used to perform a noise-ceiling/disattenuation analysis. Knowing how much reliable variance is out there to explain would help to contextualize whether the observed variance-explained is really low or high.

Response: Following the Reviewer's suggestion, we conducted a noise-ceiling analysis for right TPJ data, similarly to what has been done by Ejaz and colleagues (Ejaz et al., 2015). For each voxel, we calculated the average association (i.e., R^2 value) between single-subject timeseries and group-level activity. This procedure considers group-level fMRI data as the ground-truth model. However, this averaged signal is biased as it includes single-subject information from all the enrolled participants, ultimately producing an overestimate of the actual noise-ceiling level (i.e., the upper bound). Therefore, to obtain an estimate of the lower-bound of noise-ceiling we iteratively measured the association between each individual timeseries and the group-level average signal obtained from all the other participants (i.e., leave-one-subject-out procedure). Considering the peak of association between the *emotion dimension model* and brain activity (CoG $x = 61$, $y = -40$, $z = 19$), lower and upper bounds were 0.13 and 0.23. These numbers suggest that, within right TPJ, our *emotion dimension model* ($R^2 = 0.07$) explains between 30% (i.e., upper bound) and 54% (i.e., lower bound) of the variance. Even though *emotion dimensions* explain at least one-third of the variance, other stimulus features are likely to be encoded within right TPJ. This is also in line with the large number of tasks activating this region (Young et al., 2010; Kim, 2011; Nardo et al., 2011; Bzdok et al., 2013; Krall et al., 2015). The results for the noise-ceiling procedure are reported in the revised version of the Results and detailed in Supplementary Materials.

References

1. Adams Jr, R. B., Rule, N. O., Franklin Jr, R. G., Wang, E., Stevenson, M. T., Yoshikawa, S., ... & Ambady, N. (2010). Cross-cultural reading the mind in the eyes: an fMRI investigation. *Journal of cognitive neuroscience*, 22(1), 97-108.
2. Aichhorn, M., Perner, J., Weiss, B., Kronbichler, M., Staffen, W., & Ladurner, G. (2009). Temporo-parietal junction activity in theory-of-mind tasks: falseness, beliefs, or attention. *Journal of Cognitive Neuroscience*, 21(6), 1179-1192.
3. Bilenko, N. Y., & Gallant, J. L. (2016). Pyrcca: regularized kernel canonical correlation analysis in python and its applications to neuroimaging. *Frontiers in neuroinformatics*, 10, 49.
4. Burnett, S., & Blakemore, S. J. (2009). Functional connectivity during a social emotion task in adolescents and in adults. *European Journal of Neuroscience*, 29(6), 1294-1301.
5. Butler, E. A., Lee, T. L., & Gross, J. J. (2007). Emotion regulation and culture: Are the social consequences of emotion suppression culture-specific?. *Emotion*, 7(1), 30.
6. Bzdok, D., Langner, R., Schilbach, L., Jakobs, O., Roski, C., Caspers, S., ... & Eickhoff, S. B. (2013). Characterization of the temporo-parietal junction by combining data-driven parcellation, complementary connectivity analyses, and functional decoding. *Neuroimage*, 81, 381-392.
7. Cheng, K., Waggoner, R. A., & Tanaka, K. (2001). Human ocular dominance columns as revealed by high-field functional magnetic resonance imaging. *Neuron*, 32(2), 359-374.
8. Chiao, J. Y., Iidaka, T., Gordon, H. L., Nogawa, J., Bar, M., Aminoff, E., ... & Ambady, N. (2008). Cultural specificity in amygdala response to fear faces. *Journal of Cognitive Neuroscience*, 20(12), 2167-2174.
9. Connolly, A.C., Sha, L., Guntupalli, J.S., Oosterhof, N., Halchenko, Y.O., Nastase, S.A., ... & Haxby, J.V. (2016). How the human brain represents perceived dangerousness or “predacity” of animals. *Journal of neuroscience*, 36(19), 5373-5384.
10. Cowen, A. S., & Keltner, D. (2017). Self-report captures 27 distinct categories of emotion bridged by continuous gradients. *Proceedings of the National Academy of Sciences*, 114(38), E7900-E7909.
11. Döhnell, K., Schuwerk, T., Meinhardt, J., Sodian, B., Hajak, G., & Sommer, M. (2012). Functional activity of the right temporo-parietal junction and of the medial prefrontal cortex associated with true and false belief reasoning. *Neuroimage*, 60(3), 1652-1661.
12. Dumoulin, S.O., & Wandell, B.A. (2008). Population receptive field estimates in human visual cortex. *Neuroimage*, 39(2), 647-660.
13. Ejaz, N., Hamada, M., & Diedrichsen, J. (2015). Hand use predicts the structure of representations in sensorimotor cortex. *Nature neuroscience*, 18(7), 1034.
14. Elfenbein, H. A., & Ambady, N. (2002). On the universality and cultural specificity of emotion recognition: a meta-analysis. *Psychological bulletin*, 128(2), 203.
15. Fischl, B., Sereno, M. I., & Dale, A. M. (1999). Cortical surface-based analysis: II: inflation, flattening, and a surface-based coordinate system. *Neuroimage*, 9(2), 195-207.
16. Freeman, J., Brouwer, G. J., Heeger, D. J., & Merriam, E. P. (2011). Orientation decoding depends on maps, not columns. *Journal of Neuroscience*, 31(13), 4792-4804.
17. Fonov, V. S., Evans, A. C., McKinstry, R. C., Almlí, C. R., & Collins, D. L. (2009). Unbiased nonlinear average age-appropriate brain templates from birth to adulthood. *NeuroImage*, (47), S102.
18. Gardumi, A., Ivanov, D., Hausfeld, L., Valente, G., Formisano, E., & Uludağ, K. (2016). The effect of spatial resolution on decoding accuracy in fMRI multivariate pattern analysis. *Neuroimage*, 132, 32-42.
19. Garrett, A. S., & Maddock, R. J. (2006). Separating subjective emotion from the perception of emotion-inducing stimuli: an fMRI study. *Neuroimage*, 33(1), 263-274.
20. Gross, J. J., & Levenson, R. W. (1995). Emotion elicitation using films. *Cognition & emotion*, 9(1), 87-108.

21. Han, S., & Ma, Y. (2014). Cultural differences in human brain activity: a quantitative meta-analysis. *Neuroimage*, *99*, 293-300.
22. Hanke, M., Adelhöfer, N., Kottke, D., Iacovella, V., Sengupta, A., Kaule, F. R., ... & Stadler, J. (2016). A studyforrest extension, simultaneous fMRI and eye gaze recordings during prolonged natural stimulation. *Scientific data*, *3*, 160092.
23. Harvey, B. M., Klein, B. P., Petridou, N., & Dumoulin, S. O. (2013). Topographic representation of numerosity in the human parietal cortex. *Science*, *341*(6150), 1123-1126.
24. Haxby, J.V., Guntupalli, J.S., Connolly, A.C., Halchenko, Y.O., Conroy, B.R., Gobbini, M.I., ... & Ramadge, P.J. (2011). A common, high-dimensional model of the representational space in human ventral temporal cortex. *Neuron*, *72*(2), 404-416.
25. Hervé, P.Y., Razafimandimby, A., Jobard, G., & Tzourio-Mazoyer, N. (2013). A shared neural substrate for mentalizing and the affective component of sentence comprehension. *PloS one*, *8*(1), e54400.
26. Hervé, P.Y., Razafimandimby, A., Vigneau, M., Mazoyer, B., & Tzourio-Mazoyer, N. (2012). Disentangling the brain networks supporting affective speech comprehension. *Neuroimage*, *61*(4), 1255-1267.
27. Huntenburg, J. M., Bazin, P. L., & Margulies, D. S. (2018). Large-scale gradients in human cortical organization. *Trends in cognitive sciences*, *22*(1), 21-31.
28. Huth, A.G., Nishimoto, S., Vu, A.T., & Gallant, J.L. (2012). A continuous semantic space describes the representation of thousands of object and action categories across the human brain. *Neuron*, *76*(6), 1210-1224.
29. Huth, A. G., de Heer, W. A., Griffiths, T. L., Theunissen, F. E., & Gallant, J. L. (2016). Natural speech reveals the semantic maps that tile human cerebral cortex. *Nature*, *532*(7600), 453.
30. Immordino-Yang, M. H., Yang, X. F., & Damasio, H. (2016). Cultural modes of expressing emotions influence how emotions are experienced. *Emotion*, *16*(7), 1033.
31. Khaligh-Razavi, S. M., & Kriegeskorte, N. (2014). Deep supervised, but not unsupervised, models may explain IT cortical representation. *PLoS computational biology*, *10*(11), e1003915.
32. Kim, H. (2011). Neural activity that predicts subsequent memory and forgetting: a meta-analysis of 74 fMRI studies. *Neuroimage*, *54*(3), 2446-2461.
33. Kober, H., Barrett, L.F., Joseph, J., Bliss-Moreau, E., Lindquist, K., & Wager, T.D. (2008). Functional grouping and cortical-subcortical interactions in emotion: a meta-analysis of neuroimaging studies. *Neuroimage*, *42*(2), 998-1031.
34. Krall, S. C., Rottschy, C., Oberwelland, E., Bzdok, D., Fox, P. T., Eickhoff, S. B., ... & Konrad, K. (2015). The role of the right temporoparietal junction in attention and social interaction as revealed by ALE meta-analysis. *Brain Structure and Function*, *220*(2), 587-604.
35. Labs, A., Reich, T., Schulenburg, H., Boennen, M., Mareike, G., Golz, M., ... & Peukmann, A. K. (2015). Portrayed emotions in the movie "Forrest Gump". *F1000Research*, *4*.
36. Lee, T.W., Dolan, R.J., & Critchley, H.D. (2007). Controlling emotional expression: behavioral and neural correlates of nonimitative emotional responses. *Cerebral Cortex*, *18*(1), 104-113.
37. Lombardo, M. V., Chakrabarti, B., Bullmore, E. T., Wheelwright, S. J., Sadek, S. A., Suckling, J., ... & Baron-Cohen, S. (2010). Shared neural circuits for mentalizing about the self and others. *Journal of cognitive neuroscience*, *22*(7), 1623-1635.
38. Long, B., Yu, C.P., & Konkle, T. (2018). Mid-level visual features underlie the high-level categorical organization of the ventral stream. *Proceedings of the National Academy of Sciences*, *115*(38), E9015-E9024.
39. Mano, Y., Harada, T., Sugiura, M., Saito, D.N., & Sadato, N. (2009). Perspective-taking as part of narrative comprehension: a functional MRI study. *Neuropsychologia*, *47*(3), 813-824.

40. Margulies, D. S., Ghosh, S. S., Goulas, A., Falkiewicz, M., Huntenburg, J. M., Langs, G., ... & Jefferies, E. (2016). Situating the default-mode network along a principal gradient of macroscale cortical organization. *Proceedings of the National Academy of Sciences*, *113*(44), 12574-12579.
41. Markham, R., & Wang, L. (1996). Recognition of emotion by Chinese and Australian children. *Journal of Cross-Cultural Psychology*, *27*(5), 616-643.
42. Marsh, A. A., Elfenbein, H. A., & Ambady, N. (2003). Nonverbal “accents” cultural differences in facial expressions of emotion. *Psychological Science*, *14*(4), 373-376.
43. Matsumoto, D., Yoo, S. H., & Fontaine, J. (2008). Mapping expressive differences around the world: The relationship between emotional display rules and individualism versus collectivism. *Journal of cross-cultural psychology*, *39*(1), 55-74.
44. Morelli, S.A., & Lieberman, M.D. (2013). The role of automaticity and attention in neural processes underlying empathy for happiness, sadness, and anxiety. *Frontiers in human neuroscience*, *7*, 160.
45. Morelli, S.A., Rameson, L.T., & Lieberman, M.D. (2012). The neural components of empathy: predicting daily prosocial behavior. *Social cognitive and affective neuroscience*, *9*(1), 39-47.
46. Nardo, D., Santangelo, V., & Macaluso, E. (2011). Stimulus-driven orienting of visuo-spatial attention in complex dynamic environments. *Neuron*, *69*(5), 1015-1028.
47. Nummenmaa, L., Glerean, E., Viinikainen, M., Jääskeläinen, I. P., Hari, R., & Sams, M. (2012). Emotions promote social interaction by synchronizing brain activity across individuals. *Proceedings of the National Academy of Sciences*, *109*(24), 9599-9604.
48. Nummenmaa, L., Hirvonen, J., Parkkola, R., & Hietanen, J.K. (2008). Is emotional contagion special? An fMRI study on neural systems for affective and cognitive empathy. *Neuroimage*, *43*(3), 571-580.
49. Ortony, A., Clore, G. L., & Collins, A. (1990). *The cognitive structure of emotions*. Cambridge university press.
50. Philippot, P. (1993). Inducing and assessing differentiated emotion-feeling states in the laboratory. *Cognition and emotion*, *7*(2), 171-193.
51. Raz, G., Jacob, Y., Gonen, T., Winetraub, Y., Flash, T., Soreq, E., & Hendler, T. (2013). Cry for her or cry with her: context-dependent dissociation of two modes of cinematic empathy reflected in network cohesion dynamics. *Social cognitive and affective neuroscience*, *9*(1), 30-38.
52. Raz, G., Touroutoglou, A., Wilson-Mendenhall, C., Gilam, G., Lin, T., Gonen, T., ... & Maron-Katz, A. (2016). Functional connectivity dynamics during film viewing reveal common networks for different emotional experiences. *Cognitive, Affective & Behavioral Neuroscience*, *16*(4), 709-723.
53. Reuter, M., Schmansky, N. J., Rosas, H. D., & Fischl, B. (2012). Within-subject template estimation for unbiased longitudinal image analysis. *Neuroimage*, *61*(4), 1402-1418.
54. Roth, Z. N., Heeger, D. J., & Merriam, E. P. (2018). Stimulus vignetting and orientation selectivity in human visual cortex. *eLife*, *7*, e37241.
55. Rousseeuw, P. J., & Kaufman, L. (1990). Finding groups in data. *Hoboken: Wiley Online Library*.
56. Saxe, R., & Kanwisher, N. (2003). People thinking about thinking people: the role of the temporo-parietal junction in “theory of mind”. *Neuroimage*, *19*(4), 1835-1842.
57. Schaefer, A., Nils, F., Sanchez, X., & Philippot, P. (2010). Assessing the effectiveness of a large database of emotion-eliciting films: A new tool for emotion researchers. *Cognition and Emotion*, *24*(7), 1153-1172.
58. Schurz, M., Radua, J., Aichhorn, M., Richlan, F., & Perner, J. (2014). Fractionating theory of mind: a meta-analysis of functional brain imaging studies. *Neuroscience & Biobehavioral Reviews*, *42*, 9-34.
59. Sereno, M. I., Dale, A. M., Reppas, J. B., Kwong, K. K., Belliveau, J. W., Brady, T. J., ... & Tootell, R. B. (1995). Borders of multiple visual areas in humans revealed by functional magnetic resonance imaging. *Science*, *268*(5212), 889-893.

60. Shamay-Tsoory, S. G., & Aharon-Peretz, J. (2007). Dissociable prefrontal networks for cognitive and affective theory of mind: a lesion study. *Neuropsychologia*, *45*(13), 3054-3067.
61. Silvers, J. A., Weber, J., Wager, T. D., & Ochsner, K. N. (2014). Bad and worse: neural systems underlying reappraisal of high-and low-intensity negative emotions. *Social Cognitive and Affective Neuroscience*, *10*(2), 172-179.
62. Skerry, A. E., & Saxe, R. (2015). Neural representations of emotion are organized around abstract event features. *Current biology*, *25*(15), 1945-1954.
63. Smith, M. (1995). *Engaging characters: Fiction, emotion and the cinema*. Oxford: Clarendon Press.
64. Spunt, R. P., & Adolphs, R. (2017). The neuroscience of understanding the emotions of others. *Neuroscience letters*.
65. Srinivasan, R., Golomb, J. D., & Martinez, A. M. (2016). A neural basis of facial action recognition in humans. *Journal of Neuroscience*, *36*(16), 4434-4442.
66. Thornton, M. A., & Tamir, D. I. (2017). Mental models accurately predict emotion transitions. *Proceedings of the National Academy of Sciences*, *114*(23), 5982-5987.
67. Van Essen, D. C., Glasser, M. F., Dierker, D. L., Harwell, J., & Coalson, T. (2011). Parcellations and hemispheric asymmetries of human cerebral cortex analyzed on surface-based atlases. *Cerebral cortex*, *22*(10), 2241-2262.
68. Van Overwalle, F. (2009). Social cognition and the brain: a meta-analysis. *Human brain mapping*, *30*(3), 829-858.
69. Yacoub, E., Shmuel, A., Logothetis, N., & Uğurbil, K. (2007). Robust detection of ocular dominance columns in humans using Hahn Spin Echo BOLD functional MRI at 7 Tesla. *Neuroimage*, *37*(4), 1161-1177.
70. Young, L., Dodell-Feder, D., & Saxe, R. (2010). What gets the attention of the temporo-parietal junction? An fMRI investigation of attention and theory of mind. *Neuropsychologia*, *48*(9), 2658-2664.

Reviewers' Comments:

Reviewer #1:

Remarks to the Author:

I would like to thank the authors for addressing my prior concerns.

Reviewer #2:

Remarks to the Author:

I've read the revised manuscript, and the response to reviewers. I think this research is interesting and potentially important, but the interpretation has substantial limits.

First, the authors claim that their results reflect the first-person subjective experience of the subjects, but I continue to doubt it. For a start, there have been literally hundreds, maybe thousands, of studies of the neuroscience of first person emotion, using a whole range of direct induction techniques (IAPs photos, music, memories, game experiences, etc etc etc). From that literature, it would be profoundly surprising to conclude that first person emotional experience is 3-dimensional, that these are the dimensions, or that they are primarily represented in TPJ. You could argue: no other study has ever measured the difference between positive and negative first person experiences as well as this study. But I don't agree.

(Also FWIW, a recent study from Luke Chang's group, using natural movie watching, and similar types of analyses, concluded that first person positive vs negative experiences during movie watching is represented in medial prefrontal cortex, not TPJ.)

Plus, there are probably hundreds of studies by now showing that TPJ is strongly modulated by the states we infer and attribute to others; and a few studies showing that the dimensions of others mental states can be decoded from patterns in TPJ (and related areas).

So: are the control analyses included here strong enough to override all that prior information, and support the conclusion that this study measured first-person experience? I don't think so. The evidence is:

— third person attributions to characters, taken from another study that measured (discrete, AFC) emotion attributions characters in short disconnected scenes, explain almost as much variance, voxel wise, as the continuous first person ratings (with which they share substantial variance — maybe almost as much as possible given the reliability of the first person emotions — this was not clear). (FWIW: this comparison was not a fair test of the hypothesis that the TPJ dimensions reflect 3rd P attributions — because lots of emotion and mental state attributions may only be possible when the scene is viewed in narrative context. Chopping the movie into short segments disrupts not only first person emotion, but also the third person attributions on which those first person emotions are based).

— BUT, the dimensions derived from the third person model are not mapped continuously across cortex.

By itself that is not enough of an argument to convince me that you have found the neural basis of first person emotional experience.

Here's my most deep concern. Like many papers that claim to be about first person emotion, I suspect that this study has discovered dimensions that are highly context specific and dependent. If the

authors could use their 3 PCs as an encoding model, and predict patterns of activation in a completely different task, I'd be much more impressed. They could try any of the emotion induction datasets — photos, movies, or whatever. They might also try one of Mark Thornton's datasets, and see if this model generalizes to another 3rd party context. Either way, a generalization to a new stimulus context is the only way to check the power of this model as an actual encoding model of emotion understanding / experience.

That doesn't seem to be the authors intention. So, I think the main claim of this paper honestly should not be about first vs third person emotions (which are anyway very highly correlated during movie viewing), but about the mapping of continuous, abstract dimensions of mental life onto cortex. Indeed, the main claim of this paper is not that emotion-relevant information is spatially organized in RTPJ (others have shown this); but a proposal for a specific spatial layout of the first three dimensions, and their content.

Given that claim, I think there are still some issues with this current analysis.

— I actually find it disconcerting that the analysis gives the same results in the volume vs on the cortical sheet. This paper is making a claim about geometry. Those are very different geometries of voxels. Finding the same results both ways is not really reassuring.

— I am not convinced that the low-level feature models have sufficiently explained the stimulus-feature confounds of the high-level dimensions. From the description, it seems that they used the vector of volume energy and contrast energy as the only low level features. (Were these convolved with an HRF? I couldn't tell). But the variance explained by these models is very low (max R^2 of 0.05, in A1 — less variance explained than by the emotion model in TPJ!) That suggests that these low level feature models aren't capturing much low-level variance — and therefore that just regressing them out of the rest of the neural signal is not sufficient to ensure that there are not stimulus confounds.

Reviewer #3:

Remarks to the Author:

The authors have provided a thorough and rigorous response to the issues I raised in my initial review. The results of their additional analyses substantially strengthen the paper in a number of ways. First, comparing the temporal dynamics of emotions in the movie to those in real-life (as measured by experiencing sampling) suggests a strong correspondence between the two, despite the differing time scales involved. This strengthens the generalizability of the observed results. Second, the factor rotation analysis provides convergent evidence that the dimensions specified by the PCA are indeed those canonically represented by the brain. Moreover, this analysis is a valuable methodological innovation in itself. Third, although the examination of character's emotions replicates previous findings implicating the TPJ in theory of mind, the authors also find that the emotions of others are not topographically mapped across the cortex, unlike participants' own subjective emotions. This helps to rule out the possibility that the present results are entirely the result considering others' emotions. Finally, the noise ceiling analysis suggests that the emotion dimensions model the authors test explains considerably more of the (reliable) variance in brain activity than examination of raw R^2 would suggest. This results indicates that their model accounts for a considerable fraction of activity in the TPJ. Together these secondary analyses and the other changes the authors have made in response to my and the other reviewers' concerns considerably increase my confidence in the soundness and generalizability of their conclusions. As such, I am happy to recommend this research for publication.

Mark Thornton

Reviewer #1 (Remarks to the Author):

I would like to thank the authors for addressing my prior concerns.

We thank the Reviewer for his/her comment.

Reviewer #2 (Remarks to the Author):

I've read the revised manuscript, and the response to reviewers. I think this research is interesting and potentially important, but the interpretation has substantial limits.

We thank the Reviewer for considering our research of interest and potentially important. We acknowledge that not using a completely independent dataset to test the generalizability of the obtained results may represent a limitation of our study. Therefore, we have further emphasized this aspect in the *Discussion* section.

At the same time, we have given serious consideration to the issues raised by the Reviewer about the interpretation of the current findings and we believe that, in this reply, we address these concerns and clarify why we are confident about its appropriateness.

Moreover, given the novelty of our findings, we agree that additional independent studies are needed to corroborate our conclusions, thus, we have reframed the manuscript along the lines suggested by the Reviewer.

Hereby, we provide a point-by-point reply to all the issues raised.

First, the authors claim that their results reflect the first-person subjective experience of the subjects, but I continue to doubt it.

Although in theatrical performances and movies, actions and dialogues are not directed toward the observer, the evocative events represented on stage do elicit first person emotional experiences. Indeed, since the very early days, the aim of theatrical performances has been to induce strong emotional responses in the audience. For instance, as concerns the tragedy, Aristotle wrote (Poetics [1453b]): *"Fear and pity sometimes result from the spectacle and are sometimes aroused by the actual arrangement of the incidents, which is preferable and the mark of a better poet. The plot should be so constructed that even without seeing the play anyone hearing of the incidents*

happening thrills with fear and pity as a result of what occurs. So would anyone feel who heard the story of Oedipus".

In line with this, in our behavioral experiment we explicitly asked individuals to detail their own subjective emotional experience, during the watching of an emotionally-charged movie. As in the vast majority of human studies, we presume that subjects are aware of, honest about, and able to describe their percept and, based on this, we do not see any reason to doubt of their reports about experienced emotions. In addition to the above considerations, one may ask, is not crying because of characters' misfortunes a subjective experience that spontaneously emerges? Of course, the sadness we perceive is mediated by emotional contagion and empathy, yet this is not sufficient to question that what one is feeling is not subjective. Which other objective reasons could motivate our crying? As humans, we are able to control our emotional responses (Ochsner & Gross, 2005; Gross & Thompson, 2007; Kohn et al., 2014), even though in most of the cases it requires a reason (e.g., social pressure), time and effort.

We can imagine that this view does not perfectly match the perspective of the Reviewer, but we also believe that there is no evidence that the subjective affective states mediated by emotional contagion and empathy are substantially different from those induced by actions directed toward oneself. In conclusion, it is very likely that when subjects are asked to report their own emotional experience, they are actually describing their feelings and not merely labeling portrayed emotions.

For a start, there have been literally hundreds, maybe thousands, of studies of the neuroscience of first person emotion, using a whole range of direct induction techniques (IAPS photos, music, memories, game experiences, etc etc etc).

We agree with the reviewer that IAPS stimuli have been successfully employed in thousands of researches and represent a robust method to induce specific emotional states in the observer. In this regard, we would like to emphasize that the IAPS collection includes images showing actions directed toward the observer, but also pictures of events that are just witnessed (as happens when subjects are watching a movie). Importantly, when subjects are exposed to these stimuli their subjective report is identical. To prove this, we are attaching below two IAPS images: in the first image (A), a gun is pointed toward who is watching and, despite one can discuss the ecological validity of the stimulus, it is clear that such a negative action is directed toward him/her. In contrast, in the second image (B), a person is threatening another one using a gun with no involvement of the observer in the scene (similarly to what is represented in a movie scene). Although these two pictures are completely different with respect to the number of actors, the underlying mentalization,

and several other aspects, the reported subjective experience does not significantly differ. This is testified by the nearly indistinguishable valence and arousal scores (Lang et al., 2005): valence image A = 2.83 ± 1.79 , arousal image A = 6.54 ± 2.61 ; valence image B = 2.16 ± 1.41 , arousal image B = 6.53 ± 2.42 .

In light of this, one could argue that the concerns raised by the Reviewer about the ability of our paradigm to elicit first-person subjective experiences apply to "direct induction techniques" (e.g., IAPS) as well. Indeed, as the action shown in picture B is neither adverse nor favorable to us we could just ignore it the same way we could ignore touching scenes represented in a movie. Yet we are not able to do so, unless we put some effort into it. Therefore, rather than a matter of subjective experience *versus* portrayed emotions, we believe that the issue raised by the Reviewer may be more related to whether first-person subjective emotional experiences are induced by stimuli directed toward the observer or by something that he/she is witnessing. However, based on the data reported above (i.e., identical valence and arousal scores for image A and B) this appears to be a point of no more than modest importance. Also, as this point does not challenge the validity of reports obtained by hundreds of studies based on the IAPS collection, it should not be considered an issue with respect to the interpretation of our results.

From that literature, it would be profoundly surprising to conclude that first person emotional experience is 3-dimensional, that these are the dimensions, or that they are primarily represented in TPJ. You could argue: no other study has ever measured the difference between positive and negative first person experiences as well as this study. But I don't agree.

(Also FWIW, a recent study from Luke Chang's group, using natural movie watching, and similar types of analyses, concluded that first person positive vs negative experiences during movie watching is represented in medial prefrontal cortex, not TPJ.)

As far as the number of emotion dimensions is concerned, a large body of literature tried to establish how many fundamental dimensions are required to describe affective states (see *Introduction* section of the manuscript). It is important to note that, regardless of the adopted taxonomy, early seminal studies reported that a small number is required: for instance, 2 or 3 in the circumplex model of affect (i.e., valence, arousal and dominance; Russell & Mehrabian, 1977; Russell, 1980) and 4 in the model proposed by Fontaine and colleagues (2007; evaluation/pleasantness, potency-control, activation-arousal, and unpredictability). However, in accordance with recent works, the affective space seems to have much higher dimensionality, as compared to these 2-, 3- or 4-dimensional descriptions. By asking individuals to watch a large number of brief, yet evocative, movies and to report their subjective emotional experience, Cowen and Keltner (2017; hereinafter C&K) discovered 27 "*categorical dimensions*" (e.g., awe, horror, romance). In this regard, what is crucial for the correct interpretation of our work is that the 27-dimensional C&K affective space should be compared with the 15-dimensional affective space represented in *Forrest Gump* (see t-SNE methods), and not with the 3-dimensional organization of *emotion dimensions*. In fact, in C&K (2018), the authors clarify that "*dimensionality*" of the affective space is obtained by "finding [...] linearly separable patterns of emotion judgments", which is 15 in our work (and 27 in C&K, 2017), whereas its "*conceptualization*" is "modeling whether domain-general concepts drawn from theories of emotional appraisal/construction (valence, arousal, dominance, etc.) explain reported emotion categories" (i.e., "*affective dimensions*" in C&K, 2017). In our study, *conceptualization* refers to the three emotion dimensions of *polarity*, *complexity* and *intensity*. Hence, by adopting the C&K terminology, we can affirm that the affective space elicited by *Forrest Gump* is described by 15 *categorical dimensions* and that the *conceptualization* of these categories is three-dimensional and topographically encoded in the brain. In addition, in the original paper, the authors report that 14 *affective dimensions* explain less variance than the 27 *categorical dimensions* (i.e., Figure 3 in C&K, 2017), and one could use this evidence to indirectly question the existence of *emotion dimensions* in the brain. It should be noted, though, that in our study *emotion dimensions* are derived from the same behavioral reports that led to the 15 *distinct affective states*, while in C&K (2017), ratings of *affective dimensions* and *categorical dimensions* were collected in separate surveys. Thus, the difference between the two models (i.e., *affective dimensions* versus *categorical dimensions*) in explaining subjective reports

could be interpreted as the fact that it is harder for naive subjects to describe their own emotional experience through questions based on psychological constructs, rather than employing common labels that share the same meaning across individuals (Izard, 2007). In other words, when presented with a short movie of a baby vomiting (see <https://s3-us-west-1.amazonaws.com/emogifs/map.html>) it would be harder to describe our own emotional experience by answering questions such as "To what extent does this make you feel like a sense of commitment to an individual or creature?" for the *commitment* dimension (for the complete list of questions see Supplementary Table 2 of C&K, 2017), as compared to just report that we are "disgusted" (for the complete list of categories see Supplementary Table 1 of C&K, 2017). Please note that we mentioned this aspect in the *Discussion* section of the originally submitted manuscript (pag.15 of the first submission and pag.21 of this revision: "*Moreover, while the definition of basic emotions is common across individuals, ratings based on emotion dimensions require participants to be acquainted with the meaning of psychological constructs*").

As far as the characteristics of *emotion dimensions* are concerned, the Reviewer argues that *polarity*, *complexity* and *intensity* are not adequate to describe the emotional experience and hypothesizes that they strongly depend on the context (i.e., from the stimulus we used; see below: "*[...]this study has discovered dimensions that are highly context specific and dependent [...]*"). Nonetheless, we already discussed in the manuscript (i.e., *Discussion* section; *Polarity, Complexity and Intensity of the Emotional Experience* paragraph, pag.22) the correspondence between *polarity* and valence, and between *intensity* and arousal. Both valence and arousal have been widely adopted to represent subjective affective states in the emotion dimension literature (see for instance Barrett & Russell, 1999; Russell, 2003) and, thus, it would not be surprising that cardinal dimensions, mapping important and general characteristics of the affective experience (i.e., pleasantness and relevance/strength), are actually represented in the brain (see for instance Lane et al., 1999; Anders et al., 2004; Lewis et al., 2006). Therefore, what depends on the context is the relationship between the action represented on the screen and how we personally feel with respect to that action. What does not depend on the context, instead, is our ability to describe whether that action makes us feel happy or sad. Regarding *complexity*, even though the literature is not as rich as for valence and arousal, similar dimensions have been reported in recent studies on mental models of emotion. For instance, the work by Thornton and Tamir 2017 (previously hypothesized also in Tamir et al., 2016) describes the *human mind* dimension as one of the four cardinals and their definition is very similar to what we intend here for *complexity*. In accordance with T&T interpretation, this dimension maps states "*[...] purely mental and human specific vs. bodily and shared with animals [...]*" (pag.5982 in

T&T 2017). Please note that other than *valence* and *social impact* (the latter representing "[...] *high arousal, social vs. low arousal, asocial* [...]" states; pag.5982 in T&T 2017), their fourth dimension is the one that distinguishes emotions from cognitions. Therefore, the number and characteristics of the dimensions they propose as relevant for mental models of affective states match those revealed here.

Regarding the Reviewer statement that *emotion dimensions "are primarily represented in TPJ"*, this does not reflect what we have reported in the manuscript. Indeed, we do not claim that *polarity, complexity* and *intensity* are primarily represented in right TPJ, but that exclusively within this region all the three *emotion dimensions* are topographically mapped (as also clearly represented in Supplementary Figure 6). That emotion dimensions fit brain activity in right TPJ is in line with previous findings, as we indicated in the *Introduction* section of the manuscript (see Nummenmaa et al., 2012). Instead, the novelty of our study is that within a region involved in processing of emotions, empathy and, of course, mentalization, subjectively reported affective states are mapped following the same principle that guides the arrangement of visual and auditory stimuli in sensory cortical areas.

As for the statement "*You could argue: no other study has ever measured the difference between positive and negative first person experiences as well as this study. But I don't agree*", we would like to emphasize that we never affirmed that our behavioral paradigm is the best possible method to record the subjective emotional experience. As a matter of fact, we do not have data to support this statement. At the same time, there is no evidence that our method is inferior to any other paradigm. Therefore any further discussion about this issue is, indeed, just a matter of opinion unsupported by data.

As far as the Chang preprint (<https://doi.org/10.1101/487892>; bioRxiv) is concerned, we very much appreciated the elegant paradigm and the sophisticated methods. Specifically, Chang and collaborators recorded brain activity while subjects were viewing a 45-minute television drama. An independent group of participants was also asked to watch the same drama and periodically pause the movie to report their subjective experience. Ratings were given based on 16 emotion categories (e.g., disgust, joy, fear; Figure 6D of Chang et al., 2018) and dimensionality reduction revealed 2 *emotion-rating components*: one mapping positive and the other negative emotions. The timecourse of these two principal dimensions was associated with inter-subject correlation in vmPFC.

In light of this, we must admit that we have some difficulties in fully comprehending the Reviewer argument in citing the results of Chang preprint, to question the interpretation of our results. First, as in both Chang and our study participants were asked to describe their own subjective experience during movie watching, why should one consider reports collected by Chang as first-person subjective experience and those in our study not? For instance, in which way the scene portraying the star quarterback suffering an injury (Figure 6A of Chang et al., 2018) is substantially different from the Forrest Gump scene in which Bubba dies? Second, as both studies found correspondence between affective components and brain activity, for which reason one should consider the 2 dimensions reported by Chang an adequate model to explain the subjective emotional experience and the 3 dimensions reported here not? Moreover, to cross-validate their results, Chang and colleagues recruited a second group of subjects, who watched the exact same drama in the fMRI setting. Thus, for which reason one should believe that their (but not our) findings generalize when using direct induction techniques and completely different stimuli?

In summary, the concerns raised by the Reviewer about our experimental paradigm, which appears to motivate his/her skepticism with respect to the interpretation of our results, do apply to the Chang *et al.* study as well. As pointed out by the Reviewer, the main difference between the two studies is that first person experience maps in vmPFC for Chang and in TPJ in our case. This is, however, a *post hoc ergo propter hoc* sophism: Chang paradigm maps subjective experience because vmPFC is engaged, whereas the interpretation of the current findings is not convincing because TPJ is involved. Of note, though, vmPFC was the only brain region tested (other than V1) in the Chang preprint (please note that here we tested *emotion dimension* gradients also using the searchlight approach).

Moreover, other than the points raised above, we would like to emphasize that both vmPFC and TPJ are integrative transmodal areas (Mesulam, 1998; Margulies et al., 2016) and are thus involved in a large number of distinct tasks. For instance, a very recent meta-analysis (Lieberman et al., 2019) reported that only 6%, 12% and 11% of dmPFC, amPFC and vmPFC voxels were specifically associated to the subjective experience of emotions (Figure 4 of Lieberman et al., 2019 and also reported as limitation in the *Discussion* section of Chang et al., 2018).

For all these reasons, we do not see any motivation to use the Chang preprint as an argument to support the fallacy of our interpretation.

Plus, there are probably hundreds of studies by now showing that TPJ Is strongly modulated by the states we infer and attribute to others; and a few studies showing that the dimensions of others mental states can be decoded from patterns in TPJ (and related areas).

The role of right TPJ activity in inferring and attributing mental states to others is well documented. Indeed, many fMRI studies demonstrated higher BOLD activity in this region for the "others > self" mentalization contrast. The question, however, is whether right TPJ is exclusively involved in the attribution of mental states to others. A recent electrocorticography (ECoG) study from Matthew Lieberman's group shows that right TPJ is similarly engaged during mentalization about oneself and others. Specifically, no differences in onset, peak or offset latency were found, and responses of right TPJ anticipated those in dmPFC and vmPFC (for further details, see: https://kevmtn.github.io/files/KTan_iEEGmentalizing_poster_SANS2019.pdf). These results suggest that this brain area is not only involved in the attribution of mental states to others, but also in the mentalization of self, as also discussed by the authors: "*Self & Other mentalizing appear to rely on common neural mechanisms –BOLD differences may arise from computational load (knowing oneself better than others)*". These novel, though preliminary, results are in line and support what we report here: the description of subjective emotional experience requires mentalization about oneself and this process, carried out by our subjects during the behavioral experiment, produces emotion features that map onto the temporo-parietal cortex of independent individuals, who were not explicitly asked to mentalize, but simply to watch and "enjoy" the movie.

So: are the control analyses included here strong enough to override all that prior information, and support the conclusion that this study measured first-person experience? I don't think so. The evidence is:

— third person attributions to characters, taken from another study that measured (discrete, AFC) emotion attributions characters in short disconnected scenes, explain almost as much variance, voxel wise, as the continuous first person ratings (with which they share substantial variance — maybe almost as much as possible given the reliability of the first person emotions — this was not clear). (FWIW: this comparison was not a fair test of the hypothesis that the TPJ dimensions reflect 3rd P attributions — because lots of emotion and mental state attributions may only be possible when the scene is viewed in narrative context. Chopping the movie into short segments disrupts not only first person emotion, but also the third person attributions on which those first person emotions are based).

We thank the Reviewer for raising this issue, which allows us to clarify an important methodological aspect that we missed to detail in the previous round of revision. As a matter of

fact, Labs and colleagues (2015) divided the entire movie (120 min) in 205 segments, each having the average duration of ~35 seconds. Importantly, in their editing the authors respected the narrative of movie scenes, so that the raters could have a clear understanding of what was shown on the screen. For instance, the segment in which Forrest is in front of Jenny's grave is a single cut with a duration of ~131s (for a complete list see: https://f1000researchdata.s3.amazonaws.com/datasets/6230/94134be1-dbac-4ff6-9ce9-22c8a0fecfe2_gump_emotions.zip). Indeed, seeing a man in tears in front of a grave is more than sufficient to understand that he is experiencing *sadness*. Similarly, seeing the two main characters running toward each other while the crowd is applauding is enough to recognize that they are *happy*. We have now added this important information in the revised version of the manuscript (*Methods* section, *Right Temporo-Parietal Gradients and Portrayed Emotions* paragraph; pag.33). In addition, 50% of movie scenes lasted longer (i.e., 35s) than the 20/30s average task period used in block-design fMRI paradigms mapping ToM (i.e., 20/30s task period in block-design paradigms), as well as emotion attribution (see for instance Skerry & Saxe 2015, in which written stories were presented for 13s). Therefore, we are confident that the descriptions obtained from Labs and colleagues (2015) represent an appropriate control model of third-person emotion attribution to movie characters.

— BUT, the dimensions derived from the third person model are not mapped continuously across cortex.

By itself that is not enough of an argument to convince me that you have found the neural basis of first person emotional experience.

Indeed, our results show that (1) descriptions based on third-person emotion attribution are not topographically encoded within right TPJ, whereas (2) subjective reports map along three *emotion dimension* gradients within the exact same region. After thorough consideration of the issues raised by the Reviewer, we respectfully believe that none of those may significantly affect the interpretation of the results that we are proposing in this manuscript. At the same time, we do recognize that given the novelty of our findings, additional and independent studies are required to corroborate the current interpretation. For the above consideration, we reframed the manuscript to incorporate the indications provided by the Reviewer.

Here's my most deep concern. Like many papers that claim to be about first person emotion, I suspect that this study has discovered dimensions that are highly context specific and dependent. If

the authors could use their 3 PCs as an encoding model, and predict patterns of activation in a completely different task, I'd be much more impressed. They could try any of the emotion induction datasets — photos, movies, or whatever. They might also try one of Mark Thornton's datasets, and see if this model generalizes to another 3rd party context. Either way, a generalization to a new stimulus context is the only way to check the power of this model as an actual encoding model of emotion understanding / experience.

As described in the manuscript (*Discussion* section; *Polarity, Complexity and Intensity of the Emotional Experience* paragraph, pag.22) and reported above in this letter (pag.5), *polarity* and *intensity* relate to the valence and arousal of the emotional experience, respectively. Therefore, these two dimensions map general characteristics of affective states, which have been reliably used in the emotion literature (Barrett & Russell, 1999; Russell, 2003). Further, previous fMRI researches already demonstrated that valence and arousal are actually represented as cardinal dimensions in the brain (Lane et al., 1999; Anders et al., 2004; Lewis et al., 2006), including within right TPJ (Kensinger & Schacter, 2006; Nummenmaa et al., 2012). Altogether, there is no indication that these two affective dimensions depend on the context (i.e., the movie employed in the current study) and could not be retrieved using different stimuli. As far as *complexity* is concerned, as already highlighted above (pag.5 of this reply letter), independent relevant studies (Tamir et al., 2016; Thornton & Tamir, 2017) considered a similar dimension as crucial for the mental representation of emotions. We now report in the revised version of the manuscript the parallel between *complexity* and the *human mind* dimension described in T&T 2017 (*Discussion* section, *Polarity, Complexity and Intensity of the Emotional Experience*, pag.22-23).

All that considered, however, we certainly agree with the Reviewer that independent replications are desirable and important. For this purpose, we provided the code to estimate the significance of gradients, so that independent groups may test whether the topographic organization of right TPJ can be revealed in other datasets and with different stimuli. This aspect has been further emphasized in the *Limitations* paragraph of the revised *Discussion* section (pag.24).

That doesn't seem to be the authors intention. So, I think the main claim of this paper honestly should not be about first vs third person emotions (which are anyway very highly correlated during movie viewing), but about the mapping of continuous, abstract dimensions of mental life onto cortex. Indeed, the main claim of this paper is not that emotion-relevant information is spatially organized in RTPJ (others have shown this); but a proposal for a specific spatial layout of the first three dimensions, and their content.

The present study demonstrates that affective states reported by our subjects, as well as emotions of movie characters, are associated with changes in right TPJ activity. Yet, only subjective reports are topographically mapped within this region along three cardinal *emotion dimensions*. We believe that one convincing interpretation of these findings is that portrayed emotions fostered empathy and emotional contagion in our subjects, which likely explains the positive correlation between subjectively experienced and portrayed emotions. These mechanisms may produce a remapping of events not directed toward the observer within a subjective framework that has its anatomic-functional counterpart in the three-dimensional topographic organization of right TPJ. We have now better detailed this aspect in the revised version of the manuscript (*Discussion* section, *Right Temporo-Parietal Gradients Do Not Simply Encode Portrayed Emotions* paragraph; pag.23). In support of the above interpretation, we have provided additional arguments detailed in this reply letter.

In summary:

- 1) Over 2,500 years of philosophical reasoning on theatrical representations, as well as consolidated assumptions of subjects' compliance, awareness and honesty support the claim that when individuals are asked to watch a movie and report their own subjective emotional experience, they actually do so.
- 2) Witnessing evocative events produces first-person affective responses, independently from the direct/indirect involvement of the observer in the scene (see IAPS A and B example above). This assumption is not exclusive of our paradigm as many of the so-called direct-induction techniques (e.g., IAPS) employ the same mechanism to elicit emotions.
- 3) The lower-dimensional cortical representation of affective states does not imply that only a small number of distinct emotion categories are mapped in the brain. *Forrest Gump* elicited 15 distinct states and their conceptualization (i.e., *polarity*, *complexity* and *intensity*) is topographically represented within right TPJ.
- 4) Although we did not use other stimuli to assess the generalizability of the three *emotion dimensions*, they do reflect general emotion properties that are very likely independent from the context. *Polarity* and *intensity* represent the pleasantness and strength/relevance of the experience,

respectively. For *complexity*, recent works on mental models of emotion (Thornton & Tamir, 2017) included an analogous component.

5) Although the involvement of right TPJ in the attribution of mental states to others is well documented, whether this region is exclusively involved in this process has still to be determined. As a matter of fact, preliminary data by the Lieberman's group using intracranial recordings demonstrate that this region is similarly involved in mentalization of oneself and others.

6) In splitting the drama, Labs and colleagues respected the narrative of each movie scene. This allowed participants to clearly understand what was presented on the screen. Also, the average duration of movie segments is longer than task periods of block-design paradigms commonly used in theory of mind and emotion attribution experiments.

Given that claim, I think there are still some issues with this current analysis.

— I actually find it disconcerting that the analysis gives the same results in the volume vs on the cortical sheet. This paper is making a claim about geometry. Those are very different geometries of voxels. Finding the same results both ways is not really reassuring.

We would like to point out that the results are not identical, as there is an improvement both in effect size and significance for all the three *emotion dimension* gradients when accounting for cortical folding. Indeed, gradient estimation of *polarity* was $\rho = 0.241$; $p = 0.041$ using the volumetric pipeline and $\rho = 0.248$; $p = 0.026$ using the surface method. *Complexity* gradient yielded $\rho = 0.271$; $p = 0.013$ using Euclidean distance and $\rho = 0.314$; $p = 0.001$ adopting the Dijkstra metric; similarly, *intensity* went from $\rho = 0.229$; $p = 0.049$ to $\rho = 0.249$; $p = 0.013$.

Overall, the impact of surface-based analysis on gradient estimation is positive and, intuitively, as the difference between the two methods increases with the number of cortical folds. Therefore, the magnitude of the effect is likely proportional to the size of the region of interest. One can argue that with relatively small ROIs (as in our study) the Euclidean and Dijkstra metrics converge, whereas for longer distances (as in Margulies et al., 2016) the two methods would produce appreciably different results. To prove this, we have measured the relationship between the volume- and the surface-based methods as function of anatomical distance. Thus, we sampled all voxel pairings having specific anatomical distance based on the Euclidean metric. We then measured the Dijkstra distance of the same voxels and computed Spearman correlation and root mean squared error (RMSE) of the two estimates. This procedure was repeated for anatomical distances ranging from 3

(i.e., minimum voxel size) to ~150mm (i.e., the maximum possible distance within an hemisphere). The results reported in the figure below demonstrate that for our definition of right TPJ (15mm radius sphere) the average correlation between the two metrics was $\rho \approx 0.70$ and the average error in the Euclidean estimate is $RMSE \approx 6mm$. As can be appreciated from the attached figure, the correlation drops and RMSE rapidly increases for longer anatomical distances, reaching a plateau at ~65mm distance. This further evidence clarifies that Euclidean and geodesic geometries are very different for long distances, but similar for shorter ones and demonstrates how this property relates to our findings.

— I am not convinced that the low-level feature models have sufficiently explained the stimulus-feature confounds of the high-level dimensions. From the description, it seems that they used the vector of volume energy and contrast energy as the only low level features. (Were these convolved with an HRF? I couldn't tell). But the variance explained by these models is very low (max R^2 of 0.05, in AI — less variance explained than by the emotion model in TPJ!) That suggests that these low level feature models aren't capturing much low-level variance — and therefore that just regressing them out of the rest of the neural signal is not sufficient to ensure that there are not stimulus confounds.

In this study, we regressed out low-level acoustic (i.e., volume energy) and visual (i.e., Gabor contrast energy) features of the movie from brain hemodynamic activity. These regressors of no interest were not convolved using the HRF, as they were lagged and smoothed in time, similarly to emotion ratings. This approach is in line with the pipeline proposed by the group of Jack Gallant (i.e., Huth et al., 2016).

To further address the Reviewer concern regarding the impact of low-level features on the existence of right TPJ *emotion dimension* gradients, we first built more complex descriptions of low-level visual and acoustic features of Forrest Gump. We then regressed out this information from BOLD

signal, similarly to what we did in the previous version of the manuscript, and tested the significance of the *polarity*, *complexity* and *intensity* gradients.

We selected spectral power density as a model of low-level acoustic information (de Heer et al. 2017), and GIST descriptors for visual features (Oliva & Torralba, 2001; Rice et al., 2014; Handjaras et al., 2017). We obtained the power spectrum for each 2 s segment of the audio track and calculated the power in dB units. The procedure we used is identical to the one described in de Heer and colleagues (2017): Welch method, Gaussian window with SD of 5 ms, length 30 ms, 1 ms spacing between windows. The resulting model comprised 449 columns and described the power spectrum of the acoustic signal ranging from 0 Hz to 15 kHz in steps of 33.5 Hz.

For the visual model, we segmented each movie frame into a 4x4 grid and sampled the responses to Gabor filters having four different sizes and four possible orientations. This procedure generated a vector of 256 elements, which described each video frame in terms of spatial frequencies, Gabor filter orientations and positions in the visual field. All the GIST descriptors were averaged within a 2 s time window.

Timeseries of 449 acoustic and 256 visual features were lagged by 2s and temporally smoothed using a 10s window, similarly to the emotion ratings model.

As all our procedures rely on multiple linear regression, which advocate for the use of orthogonal predictors, we performed a PCA on the acoustic and visual models separately and isolated the first PCs explaining more than 90% of the total variance. Components were then aggregated in a single model and fitted into brain activity to ensure that they explain more variance than the models we employed in the previous submission.

As a result of this procedure, we fitted 21 PCs describing the low-level acoustic and visual features of the movie and more than doubled the explained variance in early sensory cortical areas: 12% in Heschl's gyrus and 9% in pericalcarine cortex. Of note, we also performed noise ceiling estimation for the highest R^2 voxels, obtaining 0.268 and 0.172 as upper and lower bounds in auditory cortex and 0.412 and 0.330 in early visual cortex. These numbers suggest that our new model explains up to 70% and 27% of brain activity in early auditory and visual areas, respectively. Please note that the definition of a computational model of low-level visual features able to explain the vast majority of early visual cortex activity during naturalistic stimulation (e.g., no fixation, continuous stimulation) is still recognized as one of the most challenging task in vision studies (Raz et al., 2017).

Most importantly, when we used the new model to regress out low-level stimulus features from brain activity, we confirmed the existence of the three *emotion dimension* gradients in right TPJ: *polarity* ($\rho = 0.258$, p-value = 0.031, 95% CI: 0.252 to 0.264), *complexity* ($\rho = 0.261$, p-value =

0.013, 95% CI: 0.254 to 0.267) and *intensity* ($p = 0.270$, p -value = 0.016, 95% CI: 0.264 to 0.277). Overall, this evidence indicates that the topographic organization of affective states in right TPJ cannot be explained by low-level sensory information confounds.

This procedure and relative findings are now detailed in Supplementary Materials.

Reviewer #3 (Remarks to the Author):

The authors have provided a thorough and rigorous response to the issues I raised in my initial review. The results of their additional analyses substantially strengthen the paper in a number of ways. First, comparing the temporal dynamics of emotions in the movie to those in real-life (as measured by experiencing sampling) suggests a strong correspondence between the two, despite the differing time scales involved. This strengthens the generalizability of the observed results. Second, the factor rotation analysis provides convergent evidence that the dimensions specified by the PCA are indeed those canonically represented by the brain. Moreover, this analysis is a valuable methodological innovation in itself. Third, although the examination of character's emotions replicates previous findings implicating the TPJ in theory of mind, the authors also find that the emotions of others are not topographically mapped across the cortex, unlike participants' own subjective emotions. This helps to rule out the possibility that the present results are entirely the

result considering others' emotions. Finally, the noise ceiling analysis suggests that the emotion dimensions model the authors test explains considerably more of the (reliable) variance in brain activity than examination of raw R^2 would suggest. This results indicates that their model accounts for a considerable fraction of activity in the TPJ. Together these secondary analyses and the other changes the authors have made in response to my and the other reviewers' concerns considerably increase my confidence in the soundness and generalizability of their conclusions. As such, I am happy to recommend this research for publication.

Mark Thornton

We thank Dr. Mark Thornton for the positive evaluation of our work.

References:

- Anders, S., Lotze, M., Erb, M., Grodd, W., & Birbaumer, N. (2004). Brain activity underlying emotional valence and arousal: A response-related fMRI study. *Human brain mapping*, 23(4), 200-209.
- Barrett, L. F., & Russell, J. A. (1999). The structure of current affect: Controversies and emerging consensus. *Current directions in psychological science*, 8(1), 10-14.
- Chang, L. J., Jolly, E., Cheong, J. H., Rapuano, K., Greenstein, N., Chen, P. H. A., & Manning, J. R. (2018). Endogenous variation in ventromedial prefrontal cortex state dynamics during naturalistic viewing reflects affective experience. *bioRxiv*, 487892.
- Cowen, A. S., & Keltner, D. (2017). Self-report captures 27 distinct categories of emotion bridged by continuous gradients. *Proceedings of the National Academy of Sciences*, 114(38), E7900-E7909.
- Cowen, A. S., & Keltner, D. (2018). Clarifying the conceptualization, dimensionality, and structure of emotion: Response to Barrett and colleagues. *Trends in cognitive sciences*, 22(4), 274-276
- de Heer, W. A., Huth, A. G., Griffiths, T. L., Gallant, J. L., & Theunissen, F. E. (2017). The hierarchical cortical organization of human speech processing. *Journal of Neuroscience*, 37(27), 6539-6557.
- Fontaine, J. R., Scherer, K. R., Roesch, E. B., & Ellsworth, P. C. (2007). The world of emotions is not two-dimensional. *Psychological science*, 18(12), 1050-1057.
- Gross, J. J., & Thompson, R. A. (2007). Emotion regulation: Conceptual foundations.
- Handjaras, G., Leo, A., Cecchetti, L., Papale, P., Lenci, A., Marotta, G., ... & Ricciardi, E. (2017). Modality-independent encoding of individual concepts in the left parietal cortex. *Neuropsychologia*, 105, 39-49.
- Huth, A. G., de Heer, W. A., Griffiths, T. L., Theunissen, F. E., & Gallant, J. L. (2016). Natural speech reveals the semantic maps that tile human cerebral cortex. *Nature*, 532(7600), 453.
- Izard, C. E. (2007). Basic emotions, natural kinds, emotion schemas, and a new paradigm. *Perspectives on psychological science*, 2(3), 260-280.
- Kensinger, E. A., & Schacter, D. L. (2006). Processing emotional pictures and words: Effects of valence and arousal. *Cognitive, Affective, & Behavioral Neuroscience*, 6(2), 110-126.
- Kohn, N., Eickhoff, S. B., Scheller, M., Laird, A. R., Fox, P. T., & Habel, U. (2014). Neural network of cognitive emotion regulation—an ALE meta-analysis and MACM analysis. *Neuroimage*, 87, 345-355.
- Labs, A., Reich, T., Schulenburg, H., Boennen, M., Mareike, G., Golz, M., ... & Peukmann, A. K. (2015). Portrayed emotions in the movie "Forrest Gump". *F1000Research*, 4.
- Lane, R. D., Chua, P. M., & Dolan, R. J. (1999). Common effects of emotional valence, arousal and attention on neural activation during visual processing of pictures. *Neuropsychologia*, 37(9), 989-997.
- Lang, P. J. (2005). International affective picture system (IAPS): Affective ratings of pictures and instruction manual. Technical report.
- Lewis, P. A., Critchley, H. D., Rotshtein, P., & Dolan, R. J. (2006). Neural correlates of processing valence and arousal in affective words. *Cerebral cortex*, 17(3), 742-748.
- Lieberman, M. D., Straccia, M. A., Meyer, M. L., Du, M., & Tan, K. M. (2019). Social, self,(situational), and affective processes in medial prefrontal cortex (MPFC): causal, multivariate, and reverse inference evidence. *Neuroscience & Biobehavioral Reviews* (doi: 10.1016/j.neubiorev.2018.12.021).

- Margulies, D. S., Ghosh, S. S., Goulas, A., Falkiewicz, M., Huntenburg, J. M., Langs, G., ... & Jefferies, E. (2016). Situating the default-mode network along a principal gradient of macroscale cortical organization. *Proceedings of the National Academy of Sciences*, 113(44), 12574-12579.
- Mesulam, M. M. (1998). From sensation to cognition. *Brain: a journal of neurology*, 121(6), 1013-1052.
- Nummenmaa, L., Glerean, E., Viinikainen, M., Jääskeläinen, I. P., Hari, R., & Sams, M. (2012). Emotions promote social interaction by synchronizing brain activity across individuals. *Proceedings of the National Academy of Sciences*, 109(24), 9599-9604.
- Ochsner, K. N., & Gross, J. J. (2005). The cognitive control of emotion. *Trends in cognitive sciences*, 9(5), 242-249.
- Oliva, A., & Torralba, A. (2001). Modeling the shape of the scene: A holistic representation of the spatial envelope. *International journal of computer vision*, 42(3), 145-175.
- Raz, G., Svanera, M., Singer, N., Gilam, G., Cohen, M. B., Lin, T., ... & Goebel, R. (2017). Robust inter-subject audiovisual decoding in functional magnetic resonance imaging using high-dimensional regression. *Neuroimage*, 163, 244-263.
- Rice, G. E., Watson, D. M., Hartley, T., & Andrews, T. J. (2014). Low-level image properties of visual objects predict patterns of neural response across category-selective regions of the ventral visual pathway. *Journal of Neuroscience*, 34(26), 8837-8844.
- Russell, J. A. (1980). A circumplex model of affect. *Journal of personality and social psychology*, 39(6), 1161.
- Russell, J. A. (2003). Core affect and the psychological construction of emotion. *Psychological review*, 110(1), 145.
- Russell, J. A., & Mehrabian, A. (1977). Evidence for a three-factor theory of emotions. *Journal of research in Personality*, 11(3), 273-294.
- Skerry, A. E., & Saxe, R. (2015). Neural representations of emotion are organized around abstract event features. *Current biology*, 25(15), 1945-1954.
- Tamir, D. I., Thornton, M. A., Contreras, J. M., & Mitchell, J. P. (2016). Neural evidence that three dimensions organize mental state representation: Rationality, social impact, and valence. *Proceedings of the National Academy of Sciences*, 113(1), 194-199.
- Thornton, M. A., & Tamir, D. I. (2017). Mental models accurately predict emotion transitions. *Proceedings of the National Academy of Sciences*, 114(23), 5982-5987.

Reviewers' Comments:

Reviewer #2:

Remarks to the Author:

I remain unconvinced that the authors have fairly tested the alternative hypothesis I proposed. Instead, they have either misunderstood the hypothesis, or deemed it to be a priori untenable. They cite Aristotle (!) as evidence that first and third person emotions are fundamentally and profoundly confounded in responses to drama -- if so, then it seems strange to me to use fMRI to "prove" that RTPJ represents the former and not the latter. If by contrast the distinction between first person emotions and third person emotion attributions is of high theoretical relevance (as suggested by the rhetoric of this paper), then I do not think the authors have shown that their results reflect first person emotions, for the same reasons I said last time.

I am pleased that the authors have put effort into making their analysis pipeline easily accessible to others.

Reviewer #2 (Remarks to the Author):

I remain unconvinced that the authors have fairly tested the alternative hypothesis I proposed. Instead, they have either misunderstood the hypothesis, or deemed it to be a priori untenable. They cite Aristotle (!) as evidence that first and third person emotions are fundamentally and profoundly confounded in responses to drama -- if so, then it seems strange to me to use fMRI to "prove" that RTPJ represents the former and not the latter. If by contrast the distinction between first person emotions and third person emotion attributions is of high theoretical relevance (as suggested by the rhetoric of this paper), then I do not think the authors have shown that their results reflect first person emotions, for the same reasons I said last time.

We thank the Reviewer for this further comment on our manuscript. The reason for citing Aristotle was to emphasize that the aim of theatrical performances, and therefore of movies, has always been to induce strong emotional responses in the audience. At the same time, we recognize how difficult is to disentangle first-person experience from portrayed emotions, as many of the paradigms employed in psychological research require subjects to attend stimuli with no explicit involvement of the observer (e.g., a picture of a woman threatened by a burglar as in IAPS or Forrest kissing Jenny for the first time as in our movie).

In our view, when subjects are presented with such stimuli and asked to describe their own emotions, what they report is a subjective experience.

Nonetheless, we acknowledge that the association between subjective reports and characters' emotions, as well as differences in experimental setup may have affected the results obtained in the current study. Therefore, we agree that our findings may not provide the clearest support to the first person interpretation and we have thus rephrased the manuscript to highlight such a limitation.

Thus, any clear first- vs third-person conclusion has been removed and explicit statements on the ambiguity around this interpretation have been added throughout the manuscript. Specifically, we have emphasized that first- and third-person experiences are tightly linked, by adding the following sentence to the Introduction section:

"To understand our own emotions, as well as those of others, is crucial for human social interactions. Also, witnessing facts and events of others' life sometimes prompts inner reactions related to the beliefs, intentions and desires of actors. Through years, the relevance and pervasiveness of these aspects motivated the quest for models that optimally associate behavioral responses to emotional experiences".

In addition, we have clearly indicated in the Discussion section that further studies are needed to disentangle the first- vs third-person interpretation, by adding the following statement:

"Our findings suggest that emotion dimension gradients are better explained considering subjective reports of the affective experience, rather than by portrayed emotions. However, the significant association between subjective ratings and characters' emotions, as well as differences in rating scales and choice of emotion categories, limit the possibility to draw clear conclusions about the encoding of subjective experiences, rather than emotion attribution processes, in right TPJ topography".

Further, in the Results section, we have changed the title and toned down the paragraph: "Do emotion dimension gradients simply encode portrayed emotions?". Specifically, we have added the following sentences to highlight caveats relevant for the interpretation of the results:

"Overall, these results suggest that right TPJ topography is better explained by subjective reports, rather than by information coded in portrayed emotions. At the same time, they may not provide the clearest support for the interpretation that emotion dimension gradients exclusively map first-person experiences. First, in social interactions, one's affective state is often influenced by facts and events of others' life. In our study, we observe a positive correlation between first-person reports and portrayed emotions (e.g., highest sadness score when Forrest holds dying Bubba) and the lack of complete orthogonality between models prevents the precise distinction of the two. Second, real-time subjective ratings and accurate descriptions of characters' emotions are better captured using different experimental paradigms. Indeed, our emotion ratings were continuously recorded during movie watching, whereas for portrayed emotions, individuals tagged movie scenes in a random order choosing among a wide array of labels and were allowed to watch each excerpt more than once. In light of all this, further studies are needed to clarify whether emotion dimension gradients exclusively encode first-person experience".

I am pleased that the authors have put effort into making their analysis pipeline easily accessible to others.

We thank the Reviewer for this comment.